# Self-organization of kinetochore-fibers in human mitotic spindles

William Conway[1]*, Robert Kiewisz[2†], Gunar Fabig[2], Colm P Kelleher[3], Hai-Yin Wu[1], Maya Anjur-Dietrich[4], Thomas Müller-Reichert[2], Daniel J Needleman[3,4,5]

[1]Department of Physics, Harvard University, Cambridge, United States; [2]Experimental Center, Faculty of Medicine Carl Gustav Carus, Technische Universität Dresden, Dresden, Germany; [3]Department of Molecular and Cellular Biology, Harvard University, Cambridge, United States; [4]John A Paulson School of Engineering and Applied Sciences, Harvard University, Cambridge, United States; [5]Center for Computational Biology, Flatiron Institute, New York, United States

**Abstract** During eukaryotic cell division, chromosomes are linked to microtubules (MTs) in the spindle by a macromolecular complex called the kinetochore. The bound kinetochore microtubules (KMTs) are crucial to ensuring accurate chromosome segregation. Recent reconstructions by electron tomography (Kiewisz et al., 2022) captured the positions and configurations of every MT in human mitotic spindles, revealing that roughly half the KMTs in these spindles do not reach the pole. Here, we investigate the processes that give rise to this distribution of KMTs using a combination of analysis of large-scale electron tomography, photoconversion experiments, quantitative polarized light microscopy, and biophysical modeling. Our results indicate that in metaphase, KMTs grow away from the kinetochores along well-defined trajectories, with the speed of the KMT minus ends continually decreasing as the minus ends approach the pole, implying that longer KMTs grow more slowly than shorter KMTs. The locations of KMT minus ends, and the turnover and movements of tubulin in KMTs, are consistent with models in which KMTs predominately nucleate de novo at kinetochores in metaphase and are inconsistent with substantial numbers of non-KMTs being recruited to the kinetochore in metaphase. Taken together, this work leads to a mathematical model of the self-organization of kinetochore-fibers in human mitotic spindles.

**\*For correspondence:**
wconway@g.harvard.edu

**Present address:** †Simons Machine Learning Center, New York Structural Biology Center, New York, United States

**Competing interest:** The authors declare that no competing interests exist.

## Editor's evaluation

Conway and colleagues use a combination of experiments and theory to test models for the dynamics of kinetochore-fibers during metaphase in mammalian mitotic spindles. Their work is consistent with a model where kinetochore-fiber turnover is due primarily to the nucleation of microtubules at kinetochores, rather than from a "search-and-capture" of microtubules initiated elsewhere. This work should be of interest to experimentalists and theorists broadly interested in the control of form and in cell division.

## Introduction

When eukaryotic cells divide, a spindle composed of microtubules (MTs) and associated proteins assembles and segregates the chromosomes to the daughter cells (*Strasburger, 1880*; *McIntosh et al., 2012*; *Heald and Khodjakov, 2015*; *Petry, 2016*; *Prosser and Pelletier, 2017*, *Oriola et al., 2018*, *O'Toole et al., 2020*, *Anjur-Dietrich et al., 2021*). A macromolecular protein complex called the kinetochore binds each sister chromatid to MTs in the spindle thereby bi-orienting the two sisters to ensure they segregate to opposite daughter cells (*McDonald et al., 1992*; *McEwen et al., 1997*;

*Maiato et al., 2004a*; *Yoo et al., 2018*, *Monda and Cheeseman, 2018*; *Rieder, 1982*; *Maiato et al., 2004b*, *Musacchio and Desai, 2017*, *Pesenti, 2018*; *Monda and Cheeseman, 2018*; *DeLuca et al., 2011*; *Redemann et al., 2017*; *Long et al., 2019*). Any MT whose plus end is embedded in the kinetochore is referred to as a kinetochore microtubule (KMT) and the collection of all KMTs associated with an individual kinetochore is called a kinetochore-fiber (K-Fiber). The kinetochore-microtubule interaction stabilizes KMTs and generates tension across the sister chromatid pair (*Brinkley and Cartwright, 1975*; *Gorbsky and Borisy, 1989*; *Nicklas and Ward, 1994*; *Bakhoum et al., 2009*; *DeLuca et al., 2006*; *Cheeseman et al., 2006*; *Tanaka and Desai, 2008*; *Akiyoshi et al., 2010*; *Kabeche and Compton, 2013*; *Cheerambathur et al., 2017*; *Monda and Cheeseman, 2018*; *Steblyanko et al., 2020*; *Warren et al., 2020*). Modulation of the kinetochore-MT interaction is thought to be important in correcting mitotic errors (*DeLuca et al., 2011*; *Godek et al., 2015*; *Funabiki, 2019*; *Long et al., 2019*). Kinetochore-MT binding is thus central to normal mitotic progression and correctly segregating sister chromatids to opposite daughter cells (*Cimini et al., 2001*; *Chiang et al., 2010*; *Auckland and McAinsh, 2015*; *Lampson and Grishchuk, 2017*; *Dudka et al., 2018*). Chromosome segregation errors are implicated in a host of diseases ranging from cancer to development disorders such as Downs' and Turners' Syndromes (*Touati and Wassmann, 2016*; *Compton, 2017*, *Jo et al., 2021*).

The lifecycle of a metaphase KMT consists of its recruitment to the kinetochore, its subsequent motion, polymerization and depolymerization, and its eventual detachment from the kinetochore. In metaphase, KMTs turnover with a half-life of ~2.5min, so the KMTs that originally attached during initial spindle assembly in early prometaphase have long since detached from the kinetochore and been replaced by freshly recruited KMTs over the ~25min from nuclear envelope breakdown to anaphase. The number of KMTs remains relatively constant over the course of mitosis (*McEwen et al., 1997*; *McEwen et al., 1998*), so new KMTs must be continually recruited to kinetochores throughout metaphase to replace the detaching KMTs. Prior experiments have established that kinetochores are capable of both nucleating KMTs de novo and capturing exiting non-KMTs (*Telzer et al., 1975*; ; *Mitchison and Kirschner, 1985a*; *Mitchison and Kirschner, 1985b*, *Mitchison and Kirschner, 1986*, *Huitorel and Kirschner, 1988*; *Heald and Khodjakov, 2015*; *LaFountain and Oldenbourg, 2014*; *Petry, 2016*; *Sikirzhytski et al., 2018*; *David et al., 2019*; *Renda and Khodjakov, 2021*). Either of these mechanisms could potentially be responsible for the KMT recruitment to kinetochores during metaphase. The de novo kinetochore nucleated MTs may in fact be nucleated in the vicinity of the kinetochore and then attach while they are still near zero length, though this process would be distinct from indiscriminate capture of non-KMTs of varied lengths from the spindle (*Sikirzhytski et al., 2018*). Once attached, the plus-ends of KMTs can polymerize and depolymerize while remaining attached to the kinetochore, leading to a net flux of tubulin through the K-Fiber from the kinetochore toward the spindle pole (*Mitchison and Kirschner, 1985a*, *Mitchison, 1989*; *Rieder and Alexander, 1990*; *Mitchison and Salmon, 1992*; *Zhai et al., 1995*; *Waters et al., 1996*; *Khodjakov et al., 2003*; *Gadde and Heald, 2004*; *McIntosh et al., 2012*; *Steblyanko et al., 2020*; *DeLuca et al., 2011*; *Elting et al., 2014*; *Elting et al., 2017*, *Neahring et al., 2021*; *Risteski et al., 2021*). For human cells in metaphase, it is unclear to what extent these motions are due to movement of entire K-Fibers, movement of individual KMTs within a K-Fiber, or movement of tubulin through individual KMTs. Finally, when KMTs detach from the kinetochore, they become non-KMTs by definition. The regulation of KMT detachments is thought to be important for correcting improper attachments and ensuring accurate chromosome segregation (*Tanaka et al., 2002*; *Bakhoum et al., 2009*, *DeLuca et al., 2011*; *Godek et al., 2015*; *Krenn and Musacchio, 2015*; *Lampson and Grishchuk, 2017*; *Funabiki, 2019*; *Long et al., 2019*). KMT detachments typically occur with a time scale of ~2.5min in metaphase in human mitotic cells (*Kabeche and Compton, 2013*). How these processes – KMT recruitment, motion, polymerization and depolymerization, and detachment – lead to the self-organization of metaphase K-Fibers remains incompletely understood.

In a companion paper, we used serial-section electron tomography to reconstruct the locations, lengths, and configurations of MTs in metaphase spindles in HeLa cells (*Kiewisz et al., 2022*). These whole spindle reconstructions can unambiguously identify which MTs are bound to the kinetochore and measure their lengths, providing a remarkable new tool for the study of KMTs. Here, we sought to combine the electron tomography spindle reconstructions with live-cell experiments and biophysical modeling to characterize the lifecycle of KMTs in metaphase spindles in HeLa cells.

The electron tomography reconstructions revealed that only ~50% of KMTs have their minus ends at spindle poles. We used photoconversion experiments to measure the dynamics of KMTs, which revealed that while their stability does not spatially vary, their speed is greatest in the middle of the spindle and continually decreases closer to poles. We next show that the orientations of MTs throughout the spindle, measured by electron tomography and polarized light microscopy, can be quantitively explained by an active liquid crystal theory in which the mutual interactions between MTs cause them to locally align with each other. This argues that KMTs tend to move along well-defined trajectories in the spindle. We show that the distribution of KMT minus ends along these trajectories (measured by electron tomography) is only consistent with the motion and turnover of KMTs (measured by photoconversion) if KMTs predominately nucleate at kinetochores. Taken together, these results lead us to construct a model in which metaphase KMTs nucleate at the kinetochore and grow towards the spindle pole along defined trajectories. The KMT minus ends slow down as they approach the pole. Since the flux of tubulin is constant throughout a single KMT at any given moment in time, the minus end slowdown is coupled to a decrease in the polymerization rate at the KMT plus end. KMTs detach from the kinetochore at a constant rate, independent of the minus end position. Such a model of K-Fiber self-organization can quantitively explain the lengths, locations, configurations, motions, and turnover of KMTs throughout metaphase spindles in HeLa cells.

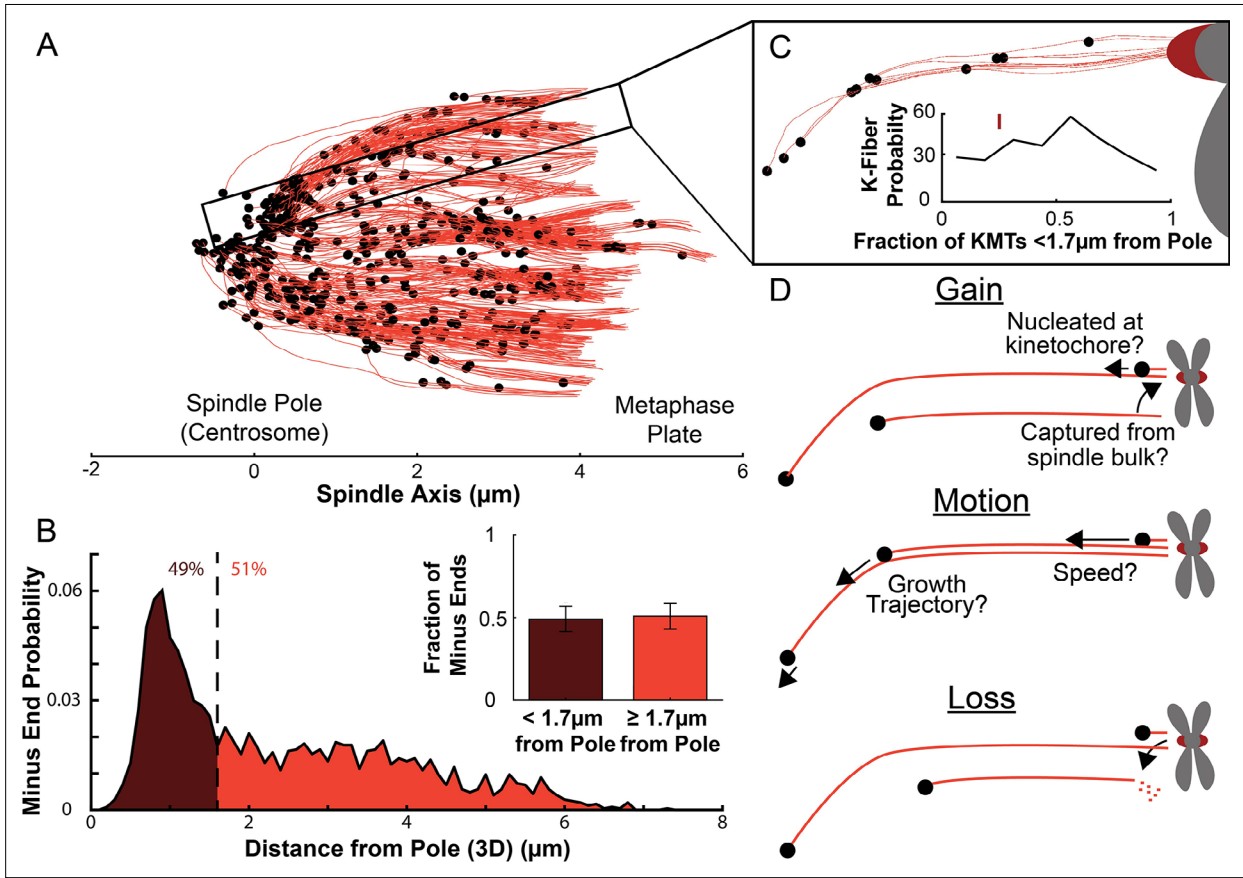

**Figure 1.** Many KMT minus ends are not in the vicinity of the pole. (**A**) A sample half spindle showing the KMTs from the EM ultrastructure. KMTs are shown in red while minus ends are marked in black. The spindle pole lies at 0μm on the spindle axis while the metaphase plate is between 4 and 6μm on the spindle axis. (**B**) The frequency of 3D minus end distance from the pole. Inset: the fraction of minus ends within 1.7μm of the pole (as shown in *Kiewisz et al., 2022*). (**C**) A sample k-Fiber. Again, KMTs are shown in red, minus ends are shown in black. The large red circle is the kinetochore. Inset: probability of k-Fiber with fraction of KMTs with their minus ends within 1.7μm of the pole. The mark shows the fraction of KMTs near the pole in the sample k-Fiber. (**D**) Schematic representation of models of KMT gain, motion, and loss. MTs could be recruited to the k-Fiber by de-novo nucleation at the kinetochore or by the capture and conversion of an pre-existing non-KMT to a KMT. The motion of KMTs is described by the trajectory and speed of the KMT minus ends. At some rate, KMTs detach from the kinetochore and become non-KMTs, by definition.

## Results

### Many KMT minus ends are not at the pole

We first analyzed a recent cellular tomography electron microscopy (EM) reconstruction data set which captured the trajectories of every MT in the mitotic spindle of three HeLa cells (*Kiewisz et al., 2022*). We defined KMTs as MTs with one end near a kinetochore in the reconstructions and assigned the plus end to the end at the kinetochore and the minus end to the opposite end of the MT (*Figure 1A*). KMT minus ends are located throughout the spindle, with approximately 51% of them more than 1.7μm away from the pole, as found in *Kiewisz et al., 2022* (*Figure 1B*). We defined the location of the pole as the center of the mother centriole. KMT minus ends are distributed throughout individual K-Fibers (*Figure 1C*), indicating that the processes that lead to a broad distribution of KMT minus end locations can occur at the level of individual kinetochores. We wanted to know how the observed distribution of KMT minus end locations results from the behaviors of KMTs. This requires understanding the life cycle of a metaphase KMT, namely (*Figure 1D*):

1. How are KMTs recruited to kinetochores in metaphase? To what extent are they nucleated de novo at the kinetochore vs. resulting from non-KMTs being captured from the bulk of the spindle?
2. How do KMTs move and grow? What are their growth trajectories and the minus end speeds?
3. How do KMTs detach from kinetochores?

We sought to answer these questions with a series of live-cell experiments, further analysis of the spindle reconstructions obtained from electron tomography, and mathematical modeling.

### The fraction of slow-turnover tubulin measured by photoactivation matches the fraction of tubulin in KMTs measured by electron tomography

To understand how the motion and turnover of KMTs results in the observed ultrastructure, we first sought to characterize the motion and stability of KMTs throughout the spindle. To that end, we constructed a HeLa line stably expressing SNAP:centrin to mark the spindle poles and PA-GFP:alpha-tubulin to mark tubulin. PA-GFP is a photoactivatable fluorophore that converts from a dark state to green fluorescence upon exposure to 750nm light using a two-photon photoactivation system. This photoactivation allows subsequent tracking of the tubulin that was in a photoactivated region at time t=0. After photoactivating a line of tubulin in the spindle, the converted tubulin moves poleward and fades over time (*Figure 2A*; *Mitchison, 1989*; *DeLuca, 2010*; *Kabeche and Compton, 2013*; *Fürthauer et al., 2019*; *Steblyanko et al., 2020*).

To measure the speed and turnover of MTs, we first projected the intensity of the photoconverted tubulin onto the spindle axis (*Figure 2B*; *Kabeche and Compton, 2013*). This projection will group together more bent KMTs near the spindle edge with less bent KMTs near the spindle center; however, the line of photoconverted tubulin remains coherent over the typical times that we tracked the photo-converted tubulin suggesting that such a projection is appropriate. The two-photon photoactivation produced a narrow line in the z-direction perpendicular to the imaging plane (σ=1.0 ± 0.1μm), so the contribution of out of focus photoactivated tubulin entering the imaging plane is minimal (*Figure 2—figure supplement 1*). We then fit the resulting peak to a Gaussian to track the motion of its center position and decay of its height over time (*Figure 2C*). We fit the position of the peak center over time to a line to determine the speed of tubulin movement in the spindle (*Figure 2D*). We then corrected the peak heights for bleaching by dividing by a bleaching reference (*Figure 2—figure supplement 2*) and fit the resulting time course to a dual-exponential decay to measure the tubulin turnover dynamics (*Figure 2E*; *DeLuca, 2010*).

Since the tubulin turnover is well-fit by a dual-exponential decay, it suggests that there are two subpopulations of MTs with different stabilities in the spindle, as previously argued for many model systems (*Brinkley and Cartwright, 1975*; *Salmon et al., 1976*; *Lambert and Bajer, 1977*, *Rieder and Bajer, 1977*; *Rieder, 1981*; *Cassimeris et al., 1990*; *DeLuca, 2010*). In prior studies, the slow-turnover subpopulation has typically been ascribed to the KMTs, while the fast-turnover subpopulation has typically been ascribed to the non-KMTs (*Zhai et al., 1995*; *DeLuca, 2010*; *Kabeche and Compton, 2013*). However, it is hypothetically possible that a portion of non-KMTs is also stabilized, due to bundling or some other mechanism (*Tipton and Gorbsky, 2022*). To gain insight into this issue, we

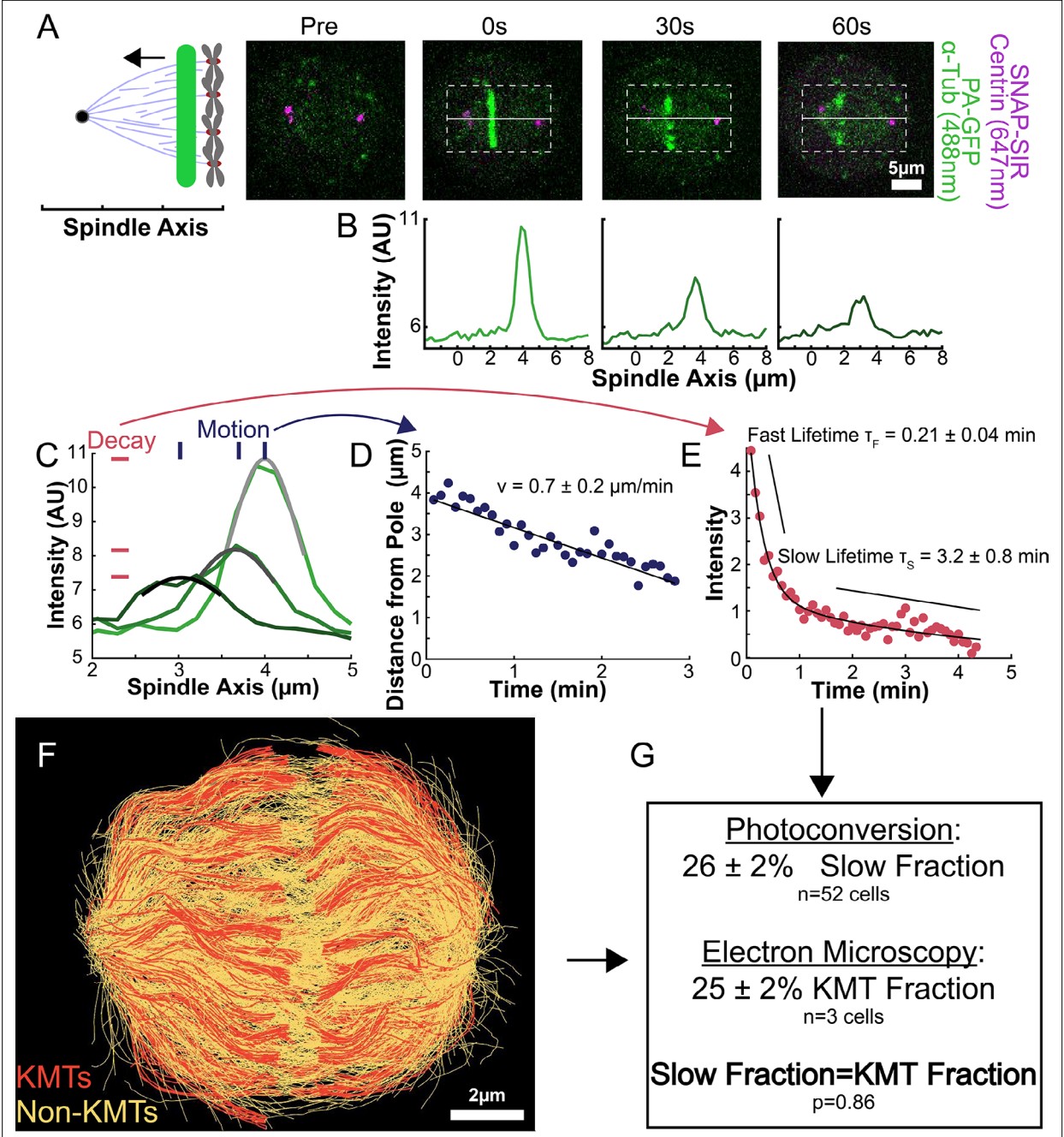

**Figure 2.** Photoactivation of spindle tubulin in live HeLa cells. (**A**) Photoactivation experiment showing PA-GFP:alpha-tubulin and SNAP-SIR:centrin immediately preceding photoactivation, 0 s, 30 s, and 60 s after photoactivation with a 750nm femtosecond pulsed laser; 500ms 488nm excitation, 514/30 bandpass emission filter; 300ms 647nm excitation, 647 longpass emission filter; 5s frame rate. (**B**) Line profile generated by averaging the intensity in 15 pixels on either side of the spindle axis in the dotted box shown in A. The intensity is corrected for background from the opposite side of the spindle (see methods). (**C**) Line profiles (shades of green) fit to Gaussian profiles (shades of grey) at 0s, 5s and 25s. Lighter shades are earlier times. The solid line on the fit represents the fit pixels. (**D**) Blue dots: fit position of the line profile peak from the sample cell shown in A, B, and C over time. Black line: linear fit to the central position of the fit peak over time. (**E**) Red dots: fit height of the line profile peak from the sample cell shown in A, B, and C over time. Black line: dual-exponential fit to the fit height of the peak over time. (**F**) Sample ultrastructure from a 3D spindle reconstructed by electron tomography (***Kiewisz et al., 2022***). KMTs are shown in red, non-KMTs yellow. (**G**) Comparison between the mean slow fraction from the photoconversion data (26% ± 2%, n=52 cells, error bars are standard error of the mean) and the fraction of KMTs (25% ± 2%, n=3 cells, error bars are standard error of the mean) from the EM data. The two means are statistically indistinguishable with *P*=0.86 on a Student's t-test.

The online version of this article includes the following figure supplement(s) for figure 2:

*Figure 2 continued on next page*

*Figure 2 continued*

**Figure supplement 1.** Z Perpendicular point spread function calibration.

**Figure supplement 2.** PA-GFP:Alpha Tubulin Bleaching Calibration.

generated a cell line with SNAP:centrin to mark the poles and mEOS3.2:alpha tubulin to mark MTs and performed photoconversion experiments on a total of 70 spindles. We compared the fraction of tubulin in KMTs, 25% ± 2% (n=3), measured by electron tomography (in which a KMT is defined morphologically as a MT with one end associated with a kinetochore; *Figure 2F*; *Kiewisz et al., 2022*) to the fraction of the slow-turnover subpopulation measured from photoconversion experiments, 26% ± 2% (n=52). Since these two fractions are statistically indistinguishable (*Figure 2G*, p=0.86 on a Students' t-test), we conclude that the slow-turnover subpopulation are indeed KMTs, and that there is not a significant number of stabilized non-KMTs.

## KMT speed is spatially varying while both KMT and non-KMT stability are uniform in the spindle bulk

We next explored the extent to which the speed and stability of MTs changed throughout the spindle (*Burbank et al., 2007*; *Yang et al., 2008*). To do this, we compared photoconversion results from lines drawn at different position along the spindle axis. After photoconverting close to the center of the spindle (~4.5 μm from the pole), the resulting line of marked tubulin migrated towards the pole (*Figure 3A*). This poleward motion was less evident when we photoconverted a line halfway between the kinetochores and the pole (*Figure 3B*), and barely visible when we photoconverted a line near the pole itself (*Figure 3C*). Tracking the subsequent motions of these photoconverted lines in different regions revealed clear differences in their speeds (*Figure 3D*), while their turnover appeared to be similar (*Figure 3E*). To quantitatively study this phenomenon, we photoconverted lines in 52 different spindles, at various distances from the pole and measured the speed and turnover times at each location. Combining data from these different spindles revealed that average speed of the photo-converted lines increased with increasing distance from the pole (*Figure 3F*; Slope = 0.25 ± 0.04(μm/min)/μm, p=4 × 10⁻⁸), while both the KMT (*Figure 3G*; Slope = −0.10 ± 0.15 (1/min)/μm, p=0.50) and non-KMT (*Figure 3H*; Slope = 0.01 ± 0.01 (1/min)/μm, p=0.13) turnover were independent of distance from the pole. Since the non-KMTs turnover roughly every 15–20s, the non-KMT contribution to the motion of the photoconverted line should be negligible roughly 1 minute after photoconversion. We typically track the photoconverted line for ~2.5min, so the line speed we measure is primarily the result of motion of tubulin in KMTs. The faster line speed further from the pole implies that tubulin in short KMTs, whose minus ends are near the kinetochore, move more quickly than tubulin in long KMTs that reach all the way from the kinetochore to the pole. The speeds we observed with the two-photon photoactivation were very similar to the speeds we observed with a traditional one-photon photoac-tivation system (*Figure 3—figure supplement 1*). The measured KMT and non-KMT lifetimes were indistinguishable between the one- and two-photon activation systems (KMT Lifetime: One-Photon: 2.7±0.2min, Two-Photon: 2.8±0.2min, p=0.71; Non-KMT Lifetime: One-Photon: 0.29±0.02min, Two-Photon: 0.26±0.01min, p=0.10). To test if the observed tubulin slowdown near the poles was a consequence of the increased curvature of MTs near the pole, we analyzed the motion of a thinner 2 μm section of the photoactivation line near the spindle axis where the MTs are relatively straight (*Figure 3—figure supplement 2*). We found that the motion of this central portion of the line moved at very similar speeds to the entire line binned together, suggesting that the observed slowdown was a not a consequence of increased curvature near the poles. These results therefore suggest that the speed of the KMTs is faster the further they are from the pole, and that the stability of KMTs and non-KMTs are constant throughout the spindle.

## KMTs and non-KMTs are well aligned in the spindle

To connect the static ultrastructure of KMTs (visualized by electron tomography) to the spatially varying KMT speeds (measured by photoconversion), we next sought to better characterize the orien-tation and alignment of MTs in the spindle. We started by separately analyzing the non-KMTs and KMTs (*Figure 4A*) in all three electron tomography reconstructions (*Figure 4—figure supplements 1 and 2*). We projected the MTs into a 2D XY plane and calculated the average orientation, $\langle \theta \rangle$ where

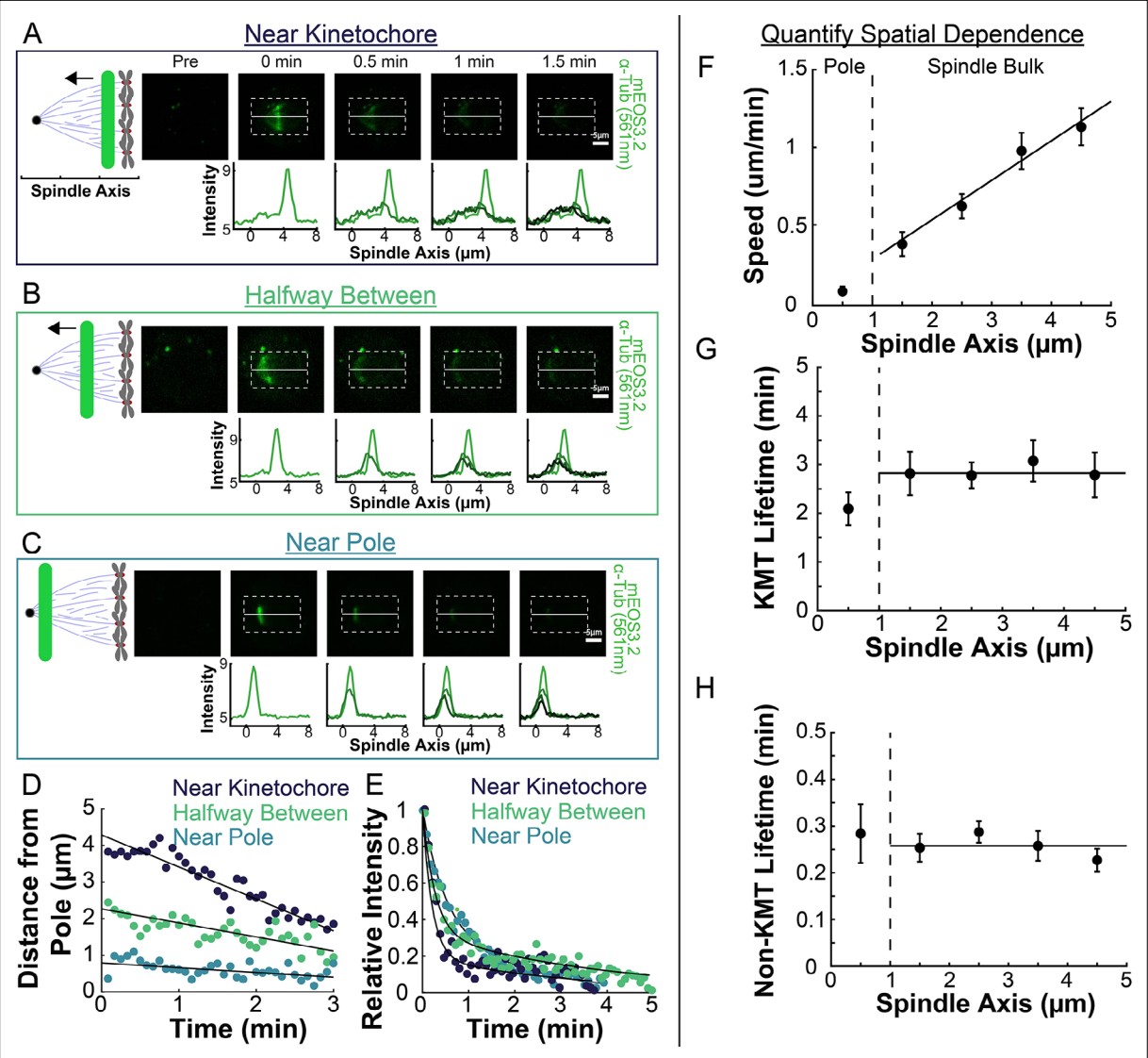

**Figure 3.** Spatial dependence of photoconversion parameters. (**A**) Sample photoactivated frames (488 nm, 500ms exposure, 5s frame rate) and line profiles from a line drawn near the kinetochore. (**B**) Sample photoconverted frames and line profiles from a line drawn halfway between the kinetochores and the pole. (**C**) Sample photoconverted frames and line profile from a line drawn near the pole. (**D**) Linear fits to the central position of the peaks from A, B and C to measure the line speeds (**E**) Dual-exponential fits to the intensity of the line in A, B, and C to measure the KMT and non-KMT lifetimes. (**F**) Line speed vs. initial position of the line drawn on the spindle axis. The area near the pole and in the spindle bulk are marked, divided by a dashed line at 1μm. Error bars are standard error of the mean. (0–1μm: n=5; 1–2μm: n=11; 2–3μm: n=15; 3–4μm: n=10; 4–5μm: n=10) (**G**) KMT lifetime vs. initial position of the line drawn on the spindle axis. (**H**) Non-KMT lifetime vs. initial position of the line drawn on the spindle axis.

The online version of this article includes the following figure supplement(s) for figure 3:

**Figure supplement 1.** Position dependence of one photon photoactivation in the spindle.

**Figure supplement 2.** Comparison of speed of tubulin in near the spindle axis and across the entire spindle width.

$\tan \theta = \frac{n_y}{n_x}$ , in the spindle for both non-KMTs (*Figure 4B*) and KMTs (*Figure 4C*). For each spindle, we averaged the spindle every $\frac{\pi}{10}$ radians to produce a uniform projection. The orientations of non-KMTs and KMTs were very similar to each other throughout the spindle, as can be seen by comparing the mean orientation of both sets of MTs along the spindle axis (*Figure 4D*). Thus, the non-KMTs and KMTs align along the same orientation field in the spindle.

The above analysis addresses how the average orientation of MTs varies throughout the spindle. We next sought to quantify the degree to which MTs are well aligned along these average orientations. This is conveniently achieved by calculating the scalar nematic order parameter, $S = \left\langle \frac{3}{2} \cos^2 \left( \theta - \langle \theta \rangle \right) - 1 \right\rangle$,

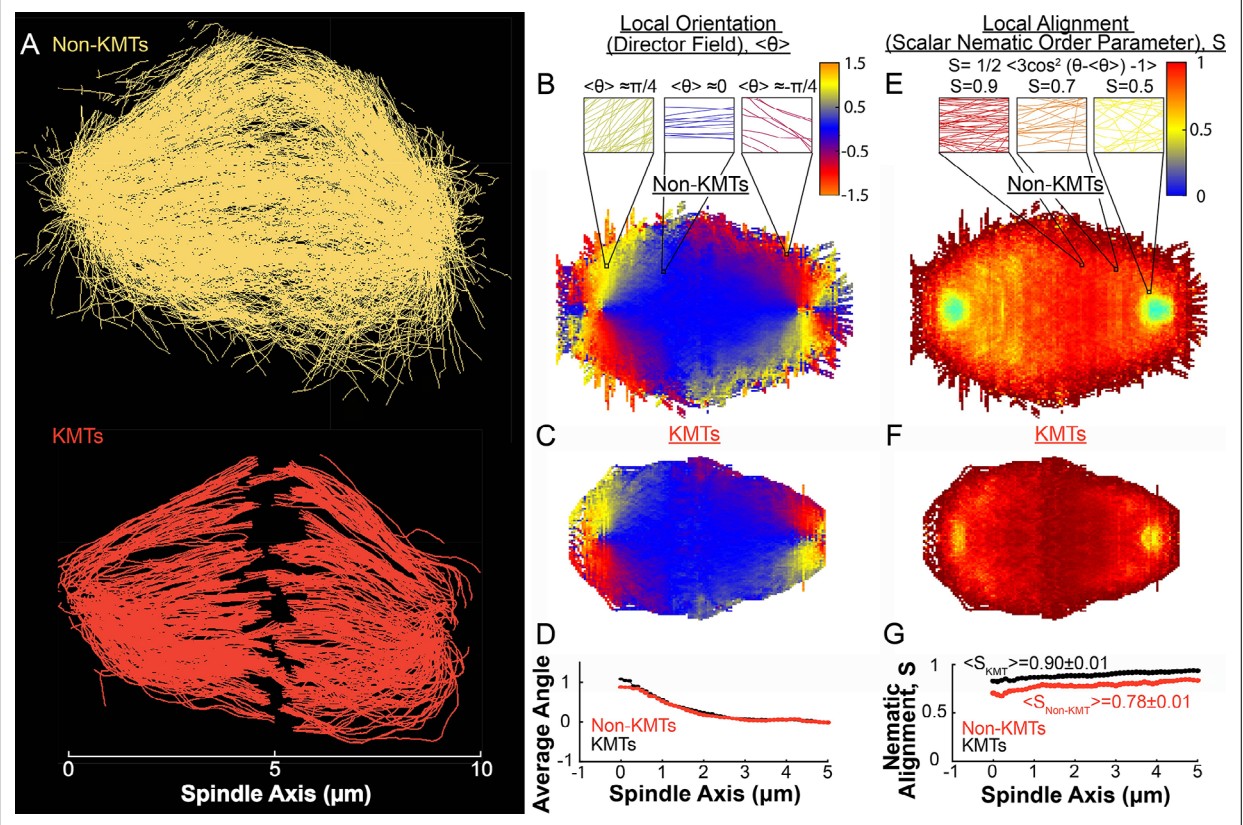

**Figure 4.** Measuring nematic alignment of non-KMTs and KMTs (3D reconstructed cell #1). (**A**) Sample from a 3D reconstruction of non-KMTs (yellow) and KMTS (red) from electron tomography (*Kiewisz et al., 2022*). (**B**) Mean local orientation of non-KMTs projected into a 2D XY plane averaged over the spindle rotated every $\frac{\pi}{10}$ radians along the spindle axis. Sample calculations of the local orientation in three representative pixels are shown above (yellow $\theta = \pi/4$, blue $\theta = 0$ red $\theta = -\pi/4$). (**C**) Mean local orientation of KMTs projected into a 2D XY plane averaged over the spindle rotated every $\frac{\pi}{10}$ radians along the spindle axis (**D**) Averaged orientation angle of KMTs (red) and non-KMTs (black) along the spindle axis. (**E**) Local alignment of the non-KMTs projected into a 2D XY plane and averaged over the spindle rotated every $\frac{\pi}{10}$ radians along the spindle axis. Sample calculation of the local alignment in three representative pixels are shown above (red S=0.9; orange S=0.7; yellow S=0.5). (**F**) Local alignment of the KMTs projected into a 2D XY plane and averaged over the spindle rotated every $\frac{\pi}{10}$ radians along the spindle axis. (**G**) Average alignment of the non-KMTs (black) and KMTs (red).

The online version of this article includes the following figure supplement(s) for figure 4:

**Figure supplement 1.** Measuring nematic alignment of non-KMTs and KMTs (3D reconstructed cell #2).

**Figure supplement 2.** Measuring nematic alignment of non-KMTs and KMTs (3D reconstructed cell #3).

which would be 1 for perfectly aligned MTs and 0 for randomly ordered MTs (*De Gennes and Post, 1993*). We calculated $S$ for both non-KMTs (*Figure 4E*) and KMTs (*Figure 4F*) throughout the spindle. Both sets of MTs are well aligned throughout the spindle (*Figure 4G*) with $\langle S \rangle = 0.90 \pm 0.01$ for KMTs and $\langle S \rangle = 0.78 \pm 0.01$ for non-KMTs. The strong alignment of MTs in the spindle along the (spatially varying) average orientation field suggests that MTs in the spindle tend to move and grow along this orientation field.

We next calculated the orientation field of MTs in HeLa spindles by averaging together data from both non-KMTs and KMTs from all three EM reconstructions by rescaling each spindle to have the same pole-pole distance and radial width (*Figure 5A*). We sought to test if the resulting orientation field was representative by obtaining data on additional HeLa spindles. Performing significantly more large-scale EM reconstructions is prohibitively time consuming, so we turned to an alternative technique: the LC-Polscope, a form of polarized light microscopy that can quantitively measure the optical slow axis (i.e. the average MT orientation) and the optical retardance (which is related to the integrated MT density over the image depth) with optical resolution (*Oldenbourg, 1998*). Due to our use of a low numerical aperture condenser (NA = 0.85), it is reasonable to approximate the Polscope

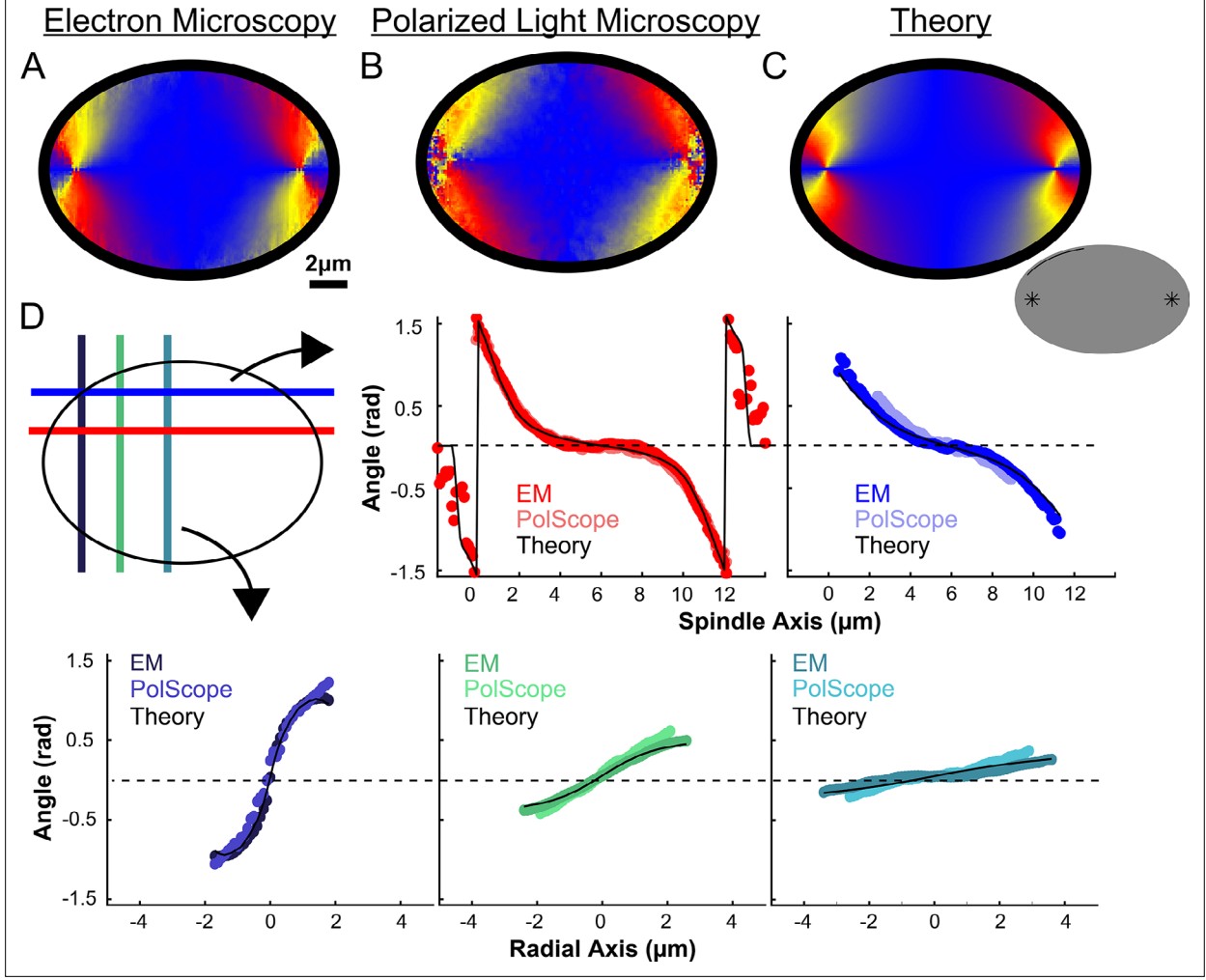

**Figure 5.** Experiment and theory of the orientation field of MTs in HeLa spindles. (**A**) Orientation field of MTs from averaging three spindle reconstructions from electron tomography. (**B**) Orientation field of MTs from averaging polarized light microscopy (LC-PolScope) data from 11 spindles. (**C**) A theoretical active liquid crystal model of the spindle geometry with tangential anchoring at the elliptical spindle boundary and point defects at the poles. Lower inset: graphic depicting the boundary conditions used in the model (tangential anchoring along the spindle boundary and radial anchoring at two point defects) (**D**) Average angle along narrow cuts parallel to the spindle and radial axis (red-lower spindle cut, blue-upper spindle cut purple-radial cut near pole, green-radial cut halfway between pole and kinetochore, teal-radial cut near kinetochore) shows close agreement between orientations from EM, polscope, and theory (black lines).

The online version of this article includes the following figure supplement(s) for figure 5:

**Figure supplement 1.** Comparison of PolScope and EM predicted measurements of the retardance area in HeLa spindles.

**Figure supplement 2.** Boundary conditions for the active liquid crystal models.

**Figure supplement 3.** Comparison of the predicted angles from the active liquid crystal theory.

**Figure supplement 4.** Experimentally measured orientation field of MTs in HeLa spindles compared to theoretical predictions with point defects localized on the spindle periphery.

measurements as projections over the z-depth of the spindle. Consistent with this expectation, the measured retardance from the PolScope agrees with the predicted retardance from projecting the entire z-depth of the EM reconstructions onto one plane (*Figure 5—figure supplement 1*). We next averaged together live-cell LC-Polscope data from eleven HeLa spindles and obtained an orientational field (*Figure 5B*) that looked remarkably similar to the projected orientations measured by EM (compare *Figure 5A and B*).

Previous work has shown that an active liquid crystal theory can quantitatively describe the morphology of *Xenopus* egg extract spindles, as well as the statistics of spontaneous fluctuations

of MT density, orientations, and stresses in those spindles (*Brugués and Needleman, 2014*; *Oriola et al., 2020*). Active liquid crystal theories are continuum theories of the collective behaviors of locally interacting, energy consuming molecules that spontaneously align (*Marchetti et al., 2013*; *Needleman and Dogic, 2017*). The central underlying assumptions in applying these theories to the spindle are that MTs align relative to each other due to their local, mutual interactions, that MTs in the spindle tend to move and grow along the direction defined by their axis, and that the phenomena of interest occur at sufficiently large length-scale such that the continuum approximation is appropriate. In this picture, the spindle is a finite size active liquid crystal 'droplet' made of interacting MTs and associated proteins. The shape of the spindle is then determined by a balance of forces in the droplet including the bending elasticity of MTs, a 'surface tension' at the spindle boundary arising from MT crosslinkers, and 'active "stresses" caused by molecular motors and MT polymerization/ depolymerization, along with a potential role of spatially regulated MT nucleation, polymerization, and depolymerization. Such active liquid crystal theories can be derived by explicitly coarse-graining equations describing the motions and interactions of MTs, molecular motors, and associated proteins (*Fürthauer et al., 2019*; *Fürthauer et al., 2021*). The resulting active liquid crystal theories can be complex and generally contain multiple coupled fields, as is the case for the previously validated active liquid crystal theory of *Xenopus* egg extract spindles. The theory, however, dramatically simplifies when there are no hydrodynamic flows, as previously observed in the spindle (*Brugués and Needleman, 2014*). We also use a single nematic elastic constant approximation, which often produces accurate results even when splay, bend, and twist deformation are associated with different moduli (as is presumably the case in the spindle). Taken together, the orientation of MTs in the spindle in the active liquid crystal theory at steady-state is given by the Laplace equation: $\nabla^2 \vec{n}(x, y, z) = 0$ where the nematic director field, $\vec{n}(x, y, z)$, are unit vectors indicating the average orientation of MTs at location $(x, y, z)$ in the spindle. Remarkably, the predicted steady-state orientation of MTs in the spindle is entirely determined by the system geometry – the spindle's boundary and the location of topological defects such as asters near the poles – and does not explicitly depend on any parameters of the theory.

We next tested if this same framework can accurately describe HeLa spindles. We fit the edge of the spindle to an elliptical boundary using a density threshold for the ET reconstructions and a retardance threshold for the PolScope images. We then calculated the orientation of MTs throughout the spindle by solving $\nabla^2 \vec{n}(x, y, z) = 0$ numerically with tangential anchoring at the spindle boundary and radial anchoring at the asters (*Figure 5C* insert, *Figure 5—figure supplement 2*). Intuitively, the 'boundary' is simply the edge of the spindle, that is, where the density of MTs in the spindle falls off very rapidly. The tangential anchoring condition encodes the tendency of MTs to orient along the edge of the spindle (which is expected because MTs crosslink each other along their lengths). The Poincaré–Hopf theorem requires that a vector field enclosed by a surface with tangential anchoring must contain topological defects (i.e., singular points where the field is discontinuous). We considered two +1 point defect discontinuities with radial anchoring to represent asters. We adjusted the location and size of the two asters, with a best fit from the projection of the 3D solution placing them near the centrosomes as expected (*Figure 5C*). The theoretically predicted orientation field is remarkably similar to the orientation fields experimentally measured with EM and LC-Polscope (*Figure 5D*). The predicted angles from a 2D approximate solution, the central slice of a 3D solution and the projection of the 3D solution are very similar, indicating that the predicted angles are robust to the exact method chosen to compare the EM reconstructions, the PolScope images, and the theory (*Figure 5—figure supplement 3*). Displacing the point aster defects to alternative locations, such as at the spindle periphery, results in substantially worse fits (*Figure 5—figure supplement 4*).

The observation that the active liquid crystal theory can accurately account for the orientation of MTs throughout HeLa spindles (a prediction which, as noted above, does not depend on any parameters of the theory) provides support for the utility of the theory and the validity of its underlying assumptions. This, in turn, suggests that the orientation of MTs in HeLa spindles are determined by their mutual, local interactions, which cause them to tend to grow and move along the direction set by the orientation field. The predicted trajectories of MT growth in the theory are streamlines that lie tangent to the MT orientation field.

# The distribution of KMT minus ends along streamlines constrains models of KMT behaviors

We next explored in more detail the implication that KMTs grow and move along the orientation field of the spindle. If the trajectories of KMTs are confined to lie along the orientation field then their minus ends will trace out paths on streamlines which lie tangent to the director field as they move towards the pole. These streamlines act like 'tracks' in the spindle that KMTs move along. We define a coordinate $s$ as the distance from the pole along the streamlines, with $s=0$ at the pole itself for all streamlines. We started by considering the locations of KMT minus ends on such streamlines. For each of the three individual reconstructed spindles, we fit the average MT orientations to the director field predicted by the active liquid crystal theory with two point defects and tangential anchoring along the spindle boundary (*Figure 6—figure supplement 1*). Then, for each KMT in each spindle, we integrated the fit director field from the KMT's minus end to the associated spindle pole to find the streamline trajectory and calculated the corresponding location as the arc length along that streamline (*Figure 6A*). We combined data from the three electron tomography reconstructions to construct the density distribution along streamlines of KMT minus ends whose plus ends were upstream of that position (*Figure 6B*, see modeling supplement). This distribution peaks roughly 1μm away from the pole and is flat in the spindle bulk.

The assumption that KMTs lie along streamlines suggests that this distribution of KMT minus ends results from the balance of three processes in metaphase (*Figure 6C*): (1) If a non-KMT whose minus end is at position $s$ along a streamline grows such that its plus end binds a kinetochore, then that non-KMT is recruited to become a KMT. This results in the addition of a new KMT minus end appearing at position $s$, which occurs with a rate $j(s)$ ; (2) KMT minus ends move towards the pole with a speed, $v(s)$, that may vary with position along the streamline. The speed of the KMT minus ends is coupled to the plus end polymerization speed because the flux of tubulin is constant throughout a single microtubule at a given point in time. Assuming that KMTs do not deviate from a single streamline trajectory, the minus end speed along the streamlines is equal to the plus end polymerization rate (in the absence of treadmilling); (3) When a KMT whose minus end is at position $s$ along a streamline detaches from the kinetochore it becomes a non-KMT (by definition). This results in the loss of a KMT minus end at position $s$, which occurs at a rate $r$. The observation that the turnover rates of KMTs, as measured by photoactivation, is uniform throughout the bulk of the spindle (*Figure 3G*) argues that the detachment rate, $r$ does not depend on the position along a streamline.

If the measured distribution of KMT minus ends (*Figure 6B*) is at steady-state in metaphase, then the fluxes from the three processes described above – gain, movement, and loss – must balance at all locations along streamlines (*Figure 6C*), leading to:

$$j(s) + v(s)\frac{dn}{ds} + \frac{dv}{ds}n(s) - \ rn(s) = 0 \qquad (1)$$

where $n(s)$ , is the density of KMT minus ends at position $s$, and $v(s)\frac{dn}{ds} + \frac{dv}{ds}n(s)$ is the flux that results from the difference between KMT minus ends moving in and out of position $s$.

Thus, *Equation 1* specifies a relationship between the distribution of KMT minus ends, $n(s)$, the spatially varying speed of KMT minus ends, $v(s)$, and rate at which KMTs are recruited, $j(s)$. This relationship suggests a means to experimentally test models of KMT recruitment: since we directly measured $n(s)$ in metaphase by electron tomography (i.e. *Figure 6B*), postulating a form $j(s)$ allows $v(s)$ to be calculated. The predicted $v(s)$ can then be compared with measured KMT movements (*Figure 3*) to determine the extent to which it, and thus the postulated $j(s)$, are consistent with both the electron microscopy and photoconversion data. This prediction requires specifying the rate of metaphase KMT detachment, which, based on our photoconversion measurements, we take to be $r=0.4$ min$^{-1}$ from the mean observed KMT lifetime in the spindle bulk (*Figure 3*) Presumably, the KMT minus end distribution $n(s)$ the KMT minus end velocity $v(s)$, the detachment rate r and the KMT recruitment rate $j(s)$ all vary over the course of mitosis, so here we only focus on metaphase when the spindle is in (approximate) steady-state.

We consider two possible models of recruitment of new KMTs to the kinetochore during metaphase: either that KMTs are nucleated at kinetochores or that KMTs arise from non-KMTs whose plus ends are captured by kinetochores. We base these two possibilities on prior experiments indicting that kinetochores are capable of both nucleating KMTs de novo (*Witt et al., 1980*; *Mitchinson and Kirschaner, 1984*; *Khodjakov et al., 2000*; *Khodjakov et al., 2003*; *Maiato et al., 2004b*; *Sikirzhytski*

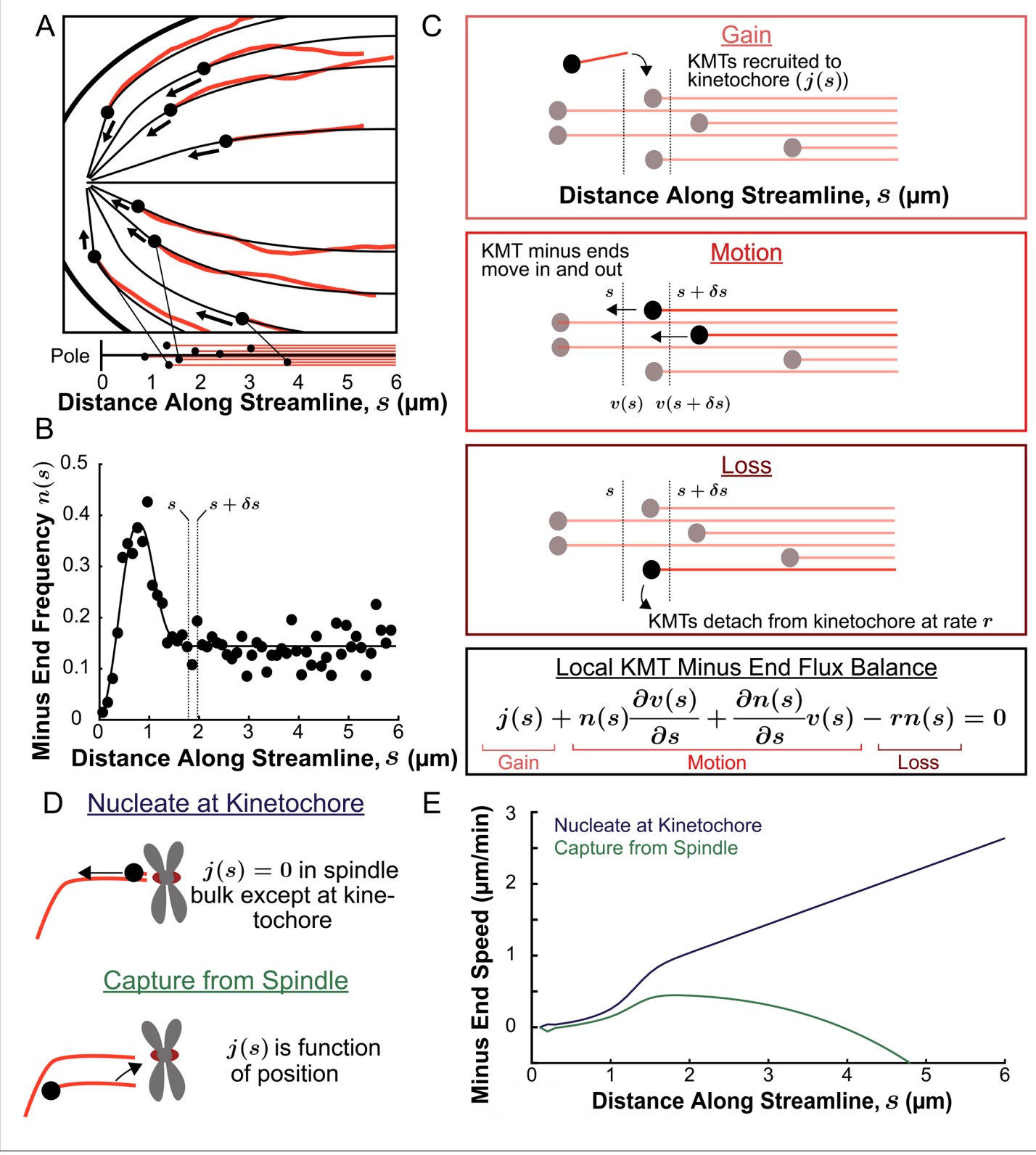

**Figure 6.** Predicting the KMT minus end speeds from the steady state distribution of minus ends along streamlines. (**A**) Eight representative KMTs from spindle reconstructions by electron tomography (red), with their minus ends (black dots) and the streamlines (thin black lines) these minus ends are located on. The distance of these minus ends along the streamlines, *x*, are depicted (lower). (**B**) Binned histogram, combining data from all three EM reconstructions, of the frequency along streamlines of KMT minus ends whose plus ends were upstream of that position. Histogram is fit to a Gaussian

*Figure 6 continued on next page*

*Figure 6 continued*

peaked near the pole and a constant in the spindle bulk (black line). (**C**) Schematic depicting cartoon representations of KMT recruitment, minus end position and KMT detachment. The three cartoons depict KMT gain, ($j(s)$), KMT minus end motion in ($n(s + \delta s)v(s + \delta s)$), KMT minus end motion out ($n(s)v(s)$) and MT loss ($r_k$). Balancing these fluxes gives the mass conservation equation $j(s) + v(s)\frac{dn}{ds} + \frac{dv}{ds}n(s) - rn(s) = 0$. (**D**) Cartoon showing two models of KMT nucleation 1. nucleate at the kinetochore where $j(s) = 0$ everywhere except at the kinetochore and 2. capture from spindle where $j(s)$ is a function of position in the spindle. (**E**) Comparison of the predictions of KMT minus end speeds in the nucleate at kinetochore and capture from spindle models.

The online version of this article includes the following figure supplement(s) for figure 6:

**Figure supplement 1.** Comparison of EM and fit liquid crystal theory for individual reconstructed spindles.

**Figure supplement 2.** Density distribution of non-KMT minus ends along streamlines.

**Figure supplement 3.** Simulated distribution of minus ends along streamlines using either a nucleate at kinetochore model (blue) or a capture from spindle recruitment model (green), compared to the experimentally measured minus distribution from electron microscopy reconstructions (black).

*et al., 2018*) and capturing existing non-KMTs (*Huitorel and Kirschner, 1988*; *Rieder and Alexander, 1990*; *Hayden et al., 1990*; *Kamasaki et al., 2013*; *David et al., 2019*). It has not been clear which of these possibilities is responsible for recruiting metaphase KMTs to kinetochores. If all metaphase KMTs were nucleated at kinetochores, then $j(s) = 0$ everywhere in the spindle bulk (*Figure 6D*, upper). These 'kinetochore-nucleated' KMTs could either be nucleated by the kinetochore itself or could be nucleated nearby and captured while still near zero length (*Sikirzhytski et al., 2018*). If instead all KMTs result from the capture of non-KMTs then $j(s)$ would be non-zero in the spindle bulk (*Figure 6D*, lower). In this latter case, $j(s)$ would be the rate that a non-KMT whose minus end is at a position $s$ along a streamline has its plus end captured by a kinetochore. We considered a model of non-KMT capture where any non-KMT can be captured provided that it reaches the kinetochore. We took the distribution of non-KMT minus ends along streamlines (*Figure 6—figure supplement 2*) as a proxy for the non-KMT nucleation rate, implying that $j(s)$ is proportional to non-KMT minus end density times that probability that a nucleated non-KMTs grows long enough to reach the kinetochore before undergoing catastrophe and depolymerizing (see supplement). These non-KMTs could be nucleated by a variety of mechanisms in the spindle bulk such as by the Augmin complex (*Goshima et al., 2008*; *David et al., 2019*). Alternatively, Augmin nucleated non-KMTs may increase the local density of aligned MTs around the k-Fiber and therefore reinforce the existing KMT trajectories (*Almeida et al., 2022*). The metaphase kinetochore nucleation model predicts that the minus end speed monotonically increases with distance away from the pole along streamlines (*Figure 6E*, see supplement). The non-KMT capture model predicts that the speed is near zero throughout the spindle. The two models thus offer qualitatively different predictions for KMT motions.

To understand why the two models offer qualitatively different predictions for the KMT minus end speeds, it is helpful to consider the contribution of each of the terms in the mass conservation, *Equation 1*, separately in the spindle bulk, where the minus end density distribution is roughly flat. In the nucleate at kinetochore model, the recruitment term, $j(s)$, is zero by definition. The first KMT minus end motion flux term $v(s)\frac{dn}{ds} = 0$ as well because the minus end density distribution is flat (i.e. $\frac{dn(s)}{ds} = 0$). This leaves only the second KMT minus end motion flux term, $\frac{dv}{ds}n(s)$, which describes changing KMT minus end speed and the detachment term $rn(s)$, giving $\frac{dv}{ds}n(s) - rn(s) = 0$, or equivalently $\frac{dv}{ds} = r$. Thus, a linear increase in the speed of the KMTs with distance from the pole balances the constant detachment term in the spindle bulk. In contrast, in the capture from spindle model, the $j(s)$ recruitment term is non-zero and can counteract the detachment terms in place of the changing speed term. The experimentally observed density of non-KMT minus ends is roughly the same as the density of KMT minus end along streamlines, so the newly nucleated KMTs roughly recapitulate the observed distribution, leaving a near-zero speed everywhere in the capture from spindle model. Therefore, the nucleate at kinetochore model predicts that the speed of KMT minus ends will increase with distance from the pole while the capture from spindle model predicts the KMT minus end speed is near-zero throughout the spindle.

## A simulation of the photoconversion experiment with nucleation at the kinetochore is consistent with the observed speed of tubulin

We next sought to determine whether the predictions from either the nucleate at kinetochore model or the capture from the spindle model were consistent with the motions of tubulin measured from the metaphase photoconversion experiments. To do so, we simulated the motion of a photoconverted line of tubulin in the metaphase spindle using the two different models for KMT recruitment with the dynamics inferred from the metaphase flux balance analysis (*Figure 6E*).

Our simulations used a discrete model of KMTs with recruitment, growth, and detachment along streamlines in the spindle. We used a 2D simulated spindle to model KMT motion. Since the imaging depth from the photoactivation experiments was narrow (~1μm), we used a central slice of the full 3D director field predicted by the active liquid crystal theory as a model for the KMT orientations. From these orientations, we generated a set of streamlines spaced 0.5μm apart at the center of the spindle (*Figure 7A*). At each timestep of the simulation, we generated newly recruited KMTs with Poisson statistics along each of these streamlines. The plus end position of these new KMTs was selected from the experimentally measured density distribution of kinetochores along streamlines (binned from all three reconstructed spindles) (*Figure 7—figure supplement 1*). The initial position of the minus ends of the new KMTs depended on the recruitment model: for the kinetochore nucleation model, the KMT minus end started at the position of kinetochores; in the capture from spindle model, the initial KMT minus end position was drawn from the (non-zero) distribution $j(s)$ (see supplement). Thus, in the kinetochore nucleation model, newly recruited KMTs start with zero length (since they are nucleated at kinetochores), while in the spindle-capture model KMTs begin with finite length (since they arise from non-KMTs whose plus ends bind kinetochores). After a lifetime drawn from an exponential distribution with a detachment rate $r$=0.4min⁻¹ (based on our photoconversion measurements), the KMT detaches from the kinetochore and is removed from the simulation.

Newly polymerized tubulin incorporates at stationary, kinetochore bound KMT plus ends, while their minus ends move backwards along the streamline towards the pole with the experimentally inferred speed $v(s)$, which varies based on the recruitment model (*Figure 6E*). In the absence of minus end depolymerization, all the tubulin in a KMT moves at the same speed as its minus end $v_{tub}(s) = v(s)$, for a KMT whose minus end is at position $s$. If, however, the minus end of a KMT depolymerizes with a speed $v_{tread}(s)$, then the tubulin in the KMT will move faster than its minus end, at speed $v_{tub}(s) = v(s) + v_{tread}(s)$. Based on a 'chipper-feeder' model of minus end depolymerization, we included minus end depolymerases only at the spindle pole (*Gadde and Heald, 2004*; *Dumont and Mitchison, 2004*, *Long et al., 2020*). KMT minus ends in the spindle bulk thus move along streamlines without minus end depolymerization. When KMT minus ends enter the pole region at position $s_p = 1.5$ μm along a streamline, the tubulin continues to incorporate at the plus end at the same speed as at the pole boundary, but minus end depolymerization begins, leading to tubulin to treadmill through the KMT at speed $v_{tread}(s) = [v(s_p) - v(s)]\theta(s_p - s)$, where $\theta(s)$ is the Heavyside step function.

Both the kinetochore nucleation model and the capture from spindle model reproduce the experimentally measured KMT minus end distribution along streamlines (*Figure 6—figure supplement 3*), as they must by construction. We next considered a 2D slice of a spindle (to replicate confocal imaging) and modeled photoconverting a line of tubulin in the spindle with a modified Cauchy profile, which fits the shape of the experimentally converted region well (*Figure 7—figure supplement 2*). We simulated the motion of tubulin in individual KMTs and summed the contributions of each KMT together to produce a final simulated spindle image. Such simulations of the kinetochore nucleation model showed a steady poleward motion of the photoconverted tubulin (*Figure 7A*). In contrast, simulations of photoconverted tubulin in the capture from spindle model exhibited substantially less motion (*Figure 7—figure supplement 3*). To facilitate comparison to experiments, we analyzed the simulations with the same approach we used for photoconversion data. First, we projected the simulated photoconverted tubulin intensity onto the spindle axis to find the photoconverted line profile over time (*Figure 7A*, lower). We then fit the simulated line profile to a Gaussian and tracked the position of the peak over time to determine the speed of tubulin at the location of photoconversion. We varied the position of the simulated photoconversion line and repeated this procedure, to measure the speed of tubulin throughout the spindle in the two recruitment models (*Figure 7B*). The predicted spatially varying speeds of tubulin in the kinetochore nucleation model are consistent with experimentally measured values (*Figure 3F*), while the predictions from the capture from spindle model are too

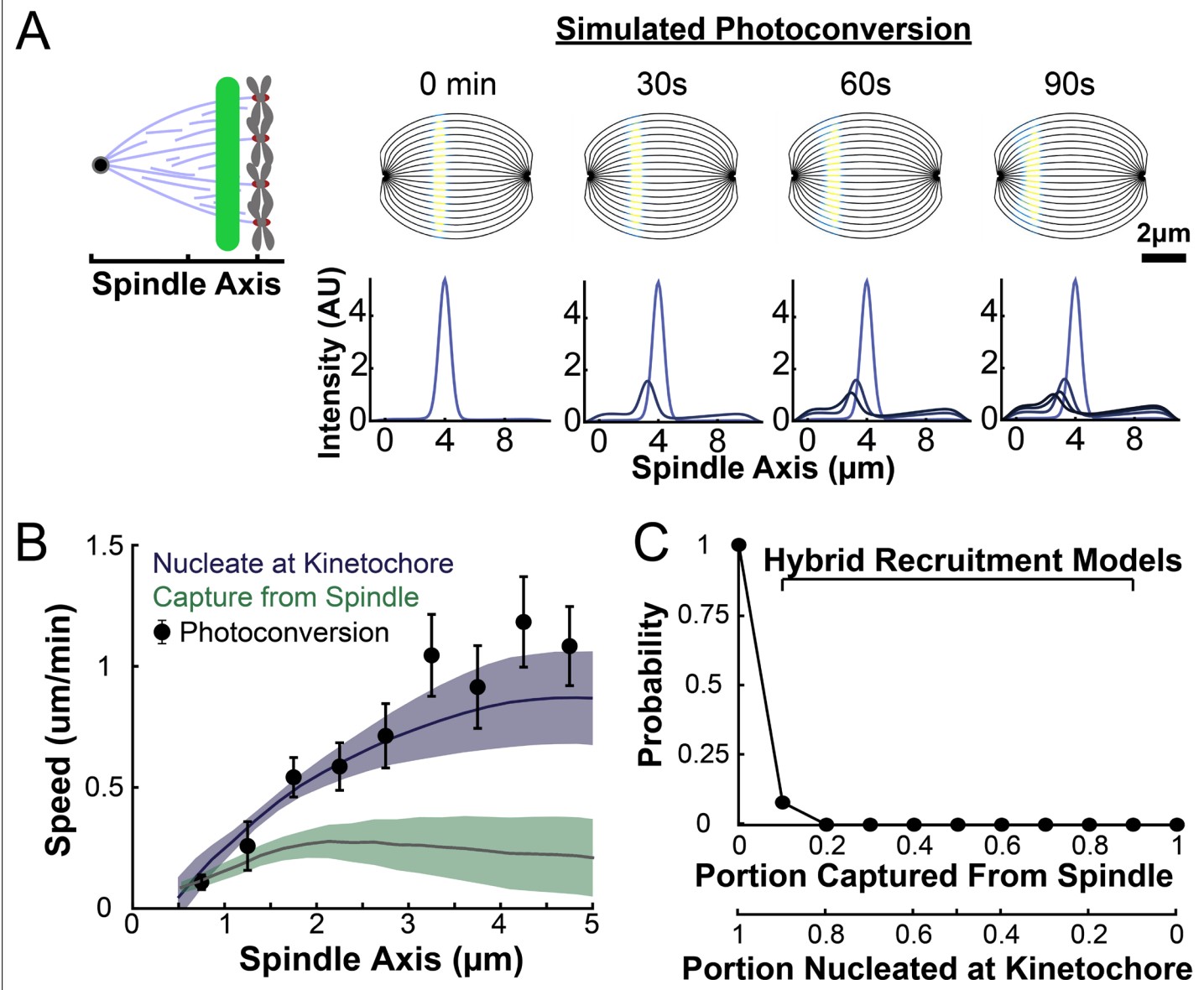

**Figure 7.** Model predicted tubulin flux compared to observed values. (**A**) Sample simulated images and line profiles from a photoconversion simulation using KMT minus end speeds in the nucleate at kinetochore model. (**B**) Comparison of the predicted spatial dependence tubulin flux speed in the nucleate at kinetochore and capture from spindle models. Error bars are standard error of the mean. (**C**) Relative probabilities of hybrid version of the two models.

The online version of this article includes the following figure supplement(s) for figure 7:

**Figure supplement 1.** Density distribution of kinetochores along streamlines.

**Figure supplement 2.** Sample experimental line profile from a photoconversion experiment and a fit modified Cauchy profile ($I(x) = \frac{1}{1+\left(\frac{x-x_0}{\sigma}\right)^2}$), The fit profile was generated by drawing a photoconverted line on the simulated spindle (*Figure 7A*) and projecting the calculated tubulin intensity onto the spindle axis with the modified Cauchy profile with various central positions $l_0$, widths w, and Cauchy exponent a.

**Figure supplement 3.** Sample simulated images from photoconversion in the capture from spindle model.

**Figure supplement 4.** Model predicted tubulin flux compared to observed values without minus end depolymerization at the pole.

**Figure supplement 5.** Model predicted tubulin flux compared to observed values with constant KMT minus end speed along streamlines in a narrow region near the spindle axis.

slow. If minus end depolymerization at the pole is turned off in the simulations, then the predicted speeds from both recruitment models become inconsistent with the experimental data (*Figure 7—figure supplement 4*). We compared the predictions from a model where KMT minus ends move at constant velocity in the spindle bulk and found that this model did not agree well with the speeds from the photoactivation experiment (*Figure 7—figure supplement 5*). This suggests that the KMT tubulin slowdown near the pole is not merely the result of increased MT curvature.

Our analysis showed that a model in which all KMTs nucleate at kinetochores is consistent with the observed speeds of tubulin throughout the spindle, while a model in which all KMTs are captured from the spindle bulk is inconsistent with this data. We next considered hybrid models which contained both KMT recruitment mechanisms. We simulated the motion of a line of photoconverted tubulin and varied the portion of KMTs nucleated at the kinetochore vs. captured from the spindle. We compare the feasibility of predictions from hybrid models with the data by calculating the Bayesian probability of observing the measured speeds with a uniform prior (*Figure 7C*). The model probability peaks at the edge where all KMTs are nucleated by the kinetochore. Thus, while the observed speeds are not inconsistent with a small fraction (less than 20%) of KMTs being captured from the spindle bulk, the data favors a model where KMTs are exclusively nucleated at kinetochores.

## A quantitative 3D model of metaphase KMT nucleation, minus end motion, and detachment

We therefore propose a model where metaphase KMTs nucleate at kinetochores and grow along streamlines (*Figure 8A*). As the KMTs grow and the KMT minus ends approach the pole, the KMT minus ends slow down. This decrease in the KMT minus end speed is coupled to a decrease in the KMT polymerization rate at the plus end. As a result of this minus end slow down, longer KMTs that reach all the way to the pole grow more slowly than short KMTs with minus ends near the kinetochore. When the KMT minus ends reach the pole, minus end depolymerization causes tubulin to treadmill through the KMT. The KMTs detach from the kinetochore at a constant rate, independent of their position in the spindle.

To test our model predictions against the full 3D reconstructed KMT ultrastructure of each spindle, we simulated the nucleation, growth, and detachment of KMTs in 3D for each spindle separately. In each spindle, we simulated KMT nucleation by placing newly formed, zero length KMTs at the reconstructed kinetochore positions with Poisson statistics. The KMT minus ends then move toward the pole at the experimentally minus end flux conservation inferred speed $v(s)$ undergo minus end depolymerization near the pole causing tubulin to treadmill at speed $v_{tread}(s) = [v(s_p) - v(s)]\theta(s_p - s)$, and detach with a constant rate r.

The agreement between the electron tomography reconstruction (*Figure 8B*) and the predicted model structure is striking (*Figure 8C*, Animation 1). We next compared the lengths of KMTs from the simulations with the experimentally measured length distribution. We found the lengths of KMTs in the simulated spindles by measuring the distance between the minus and plus end along the model KMT streamline trajectory; in the reconstructed spindles we traced the arclength of each KMT along its reconstructed trajectory. We binned the KMT lengths for each simulated and reconstructed spindle and averaged the spindles together to obtain the KMT length distributions. (*Figure 8D*). The observed length distribution of the KMTs from the reconstructed spindles is well predicted by the model. To compare the orientation of the simulated and reconstructed KMTs, we divided the MTs into short 100nm sections and projected these sub-segments onto the spindle axis to find what portion of the section lie on the spindle axis. We binned the projections from each spindle and averaged the three resulting distributions together to obtain the distribution of projected lengths along the spindle axis (*Figure 8E*). There is similarly good agreement between the simulation prediction and the reconstructed projected lengths along the spindle axis. Both the predicted lengths and orientations of the KMTs are thus consistent with the ultrastructure measured by electron tomography.

We finally tested whether the predicted tubulin motion was consistent with the photoconversion experiment. We simulated the motion of a photoconverted plane of tubulin (with a modified Cauchy intensity profile) as we did in the 2D confocal case, but now moved the tubulin along 3D nematic trajectories (Animation 1). To simulate confocal imaging, we projected a thin 1μm confocal z-slice centered at the poles onto the spindle axis over the course of the simulation to produce a line profile. The simulated line profile agrees well with the experimental profile even after 120s of simulation

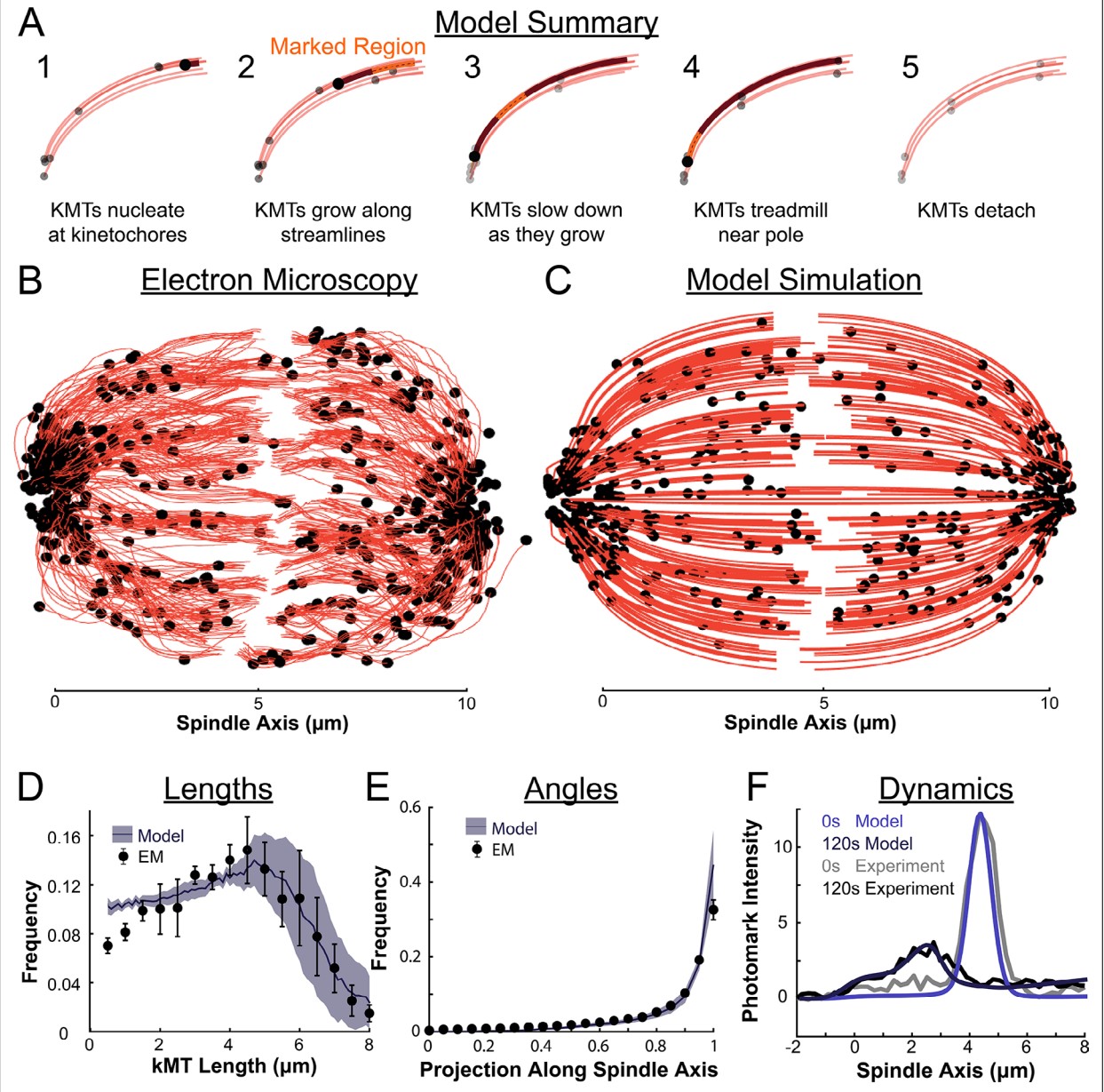

**Figure 8.** Summary of a nucleate at kinetochore model of KMT dynamics and structure in HeLa cells. (**A**) Summary of the steps of the model: 1. KMTs nucleate at kinetochores 2. KMTs grow along streamlines 3. KMTs slow down as they grow 4. KMTs treadmill near the pole; and 5. KMTs detach. (**B**) KMT structure from a sample EM reconstruction (*Kiewisz et al., 2022*; spindle #2). (**C**) Model simulation of the KMT structure given the spindle geometry and kinetochore positions. (**D**) Comparison of predicted and observed KMT lengths averaged over all three EM cells (purple-model prediction, black EM data; error bars are standard error of the mean). (**E**) Comparison of predicted and observed KMT angles averaged (purple-model prediction, black EM data; error bars are standard error of the mean). (**F**) Comparison of predicted and observed photoconverted line profiles (blue-model prediction, grey-experiment; lighter shades are 0s, darker shades are 120s).

The online version of this article includes the following figure supplement(s) for figure 8:

**Figure supplement 1.** Line spreading after photoactivation.

time (*Figure 8F*), indicating that the dynamics of the model are consistent with the experimentally measured tubulin motion and turnover. The model also accurately predicted a slight spread in the peak width over time (*Figure 8—figure supplement 1*). Taken together, these results favor a model where KMTs nucleate at the kinetochore, grow, and slow down along nematic streamlines, undergo minus end depolymerization near the pole and detach with a constant rate. Such a model is consistent

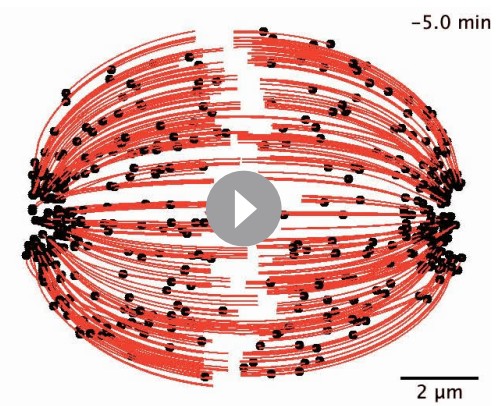

**Animation 1.** Simulated tubulin photoconversion in a 3D model spindle. Model simulation of the motion of KMTs in a nucleate at kinetochore model. KMTs are shown in red, KMT minus ends are shown in black, photoconverted tubulin is shown in yellow. The model runs for 5 minutes of simulation time before the photoconverted line is drawn.

with both measurements of KMT ultrastructure, from electron tomography, and measurements KMT dynamics, from photoconversion, in HeLa cells.

## Discussion

In this study, we leveraged recent electron tomography reconstructions that contain the positions, lengths, and configurations of MTs in metaphase spindles in HeLa cells (*Kiewisz et al., 2022*). We used these datasets, in combination with live cell microscopy measurements and biophysical modeling, to investigate the behaviors of KMTs. We found that roughly half of KMT minus ends were not located at the poles in metaphase (*Figure 1*). To better understand this metaphase KMT minus end distribution we performed a series of photoconversion experiments to measure the metaphase dynamics of KMTs. The fraction of slow turnover tubulin measured from photoconversion matched the fraction of tubulin in KMTs measured by electron tomography. This observation argues that KMTs are the only MTs in metaphase spindles that are appreciably stabilized. The photoconversion experiments also showed that tubulin in KMTs moves more slowly near the poles and that KMT turnover was uniform throughout the spindle (*Figures 2 and 3*). We found that both KMTs and non-KMTs were highly aligned in metaphase (*Figure 4*) and that the orientations of MTs throughout the spindle can be quantitively explained by an active liquid crustal theory in which MTs locally align with each other due to their mutual interactions (*Figure 5*). This suggests that KMTs tend to move along well-defined trajectories in the spindle, so we analyzed the distribution of KMT minus ends along these trajectories (*Figure 6*). From these distributions, we predicted the speed of KMT minus ends in metaphase using a mass conservation analysis. This analysis depends on the model of how metaphase KMTs are recruited to the kinetochore. We found that predictions from the nucleate at kinetochore model agreed well with the experimental metaphase measurements while the predictions from the capture from spindle model did not (*Figure 7*). We therefore propose a model where metaphase KMTs are nucleated at the kinetochore and polymerize from their plus ends as their minus ends move backwards along nematic streamline trajectories towards the pole. The KMT minus end speed decreases as the KMTs approach the pole. Because the flux of tubulin is constant throughout a single KMT at any given point in time, the decrease in the minus end speed must be accompanied by a decrease in the plus end polymerization rate (in the absence of treadmilling). KMTs detach from the kinetochore at a constant rate. This model accurately predicts the lengths, orientations, and dynamics of KMTs in mitotic spindles of metaphase HeLa cells (*Figure 8*).

Previous work has shown that the photoconversion of tubulin in the spindle implies that there are at least two population of MTs, one with fast and one with slow turnover (*Gorbsky and Borisy, 1989*; *DeLuca et al., 2016*, *Warren et al., 2020*). While the slow turnover fraction has often been ascribed to KMTs (*Zhai et al., 1995*; *DeLuca, 2010*; *Kabeche and Compton, 2013*), some work has suggested that substantial fractions of non-KMTs may be stabilized as well (*Tipton and Gorbsky, 2022*). We found that the fraction of tubulin in KMTs identified structural from electron tomography reconstructions (25% ± 2%) and the stable fraction from the photoconversion experiments (24% ± 2%) are statistically indistinguishable (*Figure 2*). This agreement argues that KMTs account for the overwhelming majority of stable MTs in the spindle. Thus, the slow decay rate can be interpreted as the rate of KMT turnover.

We observed that the speed of motion the photoconverted line was slower for lines drawn near the pole than in the center of the spindle (*Figure 3*), and our analysis of the position of the KMT minus ends indicates that the KMT minus ends themselves slow down as they approach the pole (*Figure 6*). The mechanism of KMT minus end transport to the pole is not clear. Tubulin polymerization at the

kinetochore might push the KMTs backwards towards the pole. Alternatively, the KMT minus ends could be transported to the pole by dynein (*Elting et al., 2014*; *Sikirzhytski et al., 2014*). Either way, since the flux of tubulin is constant throughout an individual KMT at any given point in time, the speed of the KMT minus end must be coupled to the speed of tubulin polymerization at the KMT plus end. The mechanism responsible for KMT minus ends slowing down as they approach the pole is also unclear. Longer KMTs are presumable subject to more drag, and more friction with the surrounding network of non-KMTs, which might lead to a reduction in the KMT polymerization rate and hence a slowdown of KMT minus ends. In *Xenopus* egg extract spindles, non-KMTs move more slowly near the spindle poles than near the spindle equator. Inhibiting dynein causes the non-KMT speed to become spatially uniform, suggesting that the non-KMT slowdown is the result of dynein-mediated MT clustering (*Burbank et al., 2007*; *Yang et al., 2008*). A similar mechanism might explain the slowdown of KMTs described in this work. Further experiments in dynein-inhibited HeLa spindles will be necessary to test this possibility.

We found that MTs in spindles in HeLa cells were well-aligned with a high scalar nematic order parameter along orientations that are consistent with the predictions of an active liquid crystal theory. This implies that the orientations of MTs in the spindle is dictated by their tendency to locally align with each other. The tendency of MTs in the spindle to locally align with each other could result from the activity of MT crosslinkers, such as dynein, kinesin-5, or PRC1 (*Kapitein et al., 2005*; *Tanenbaum et al., 2013*; *Wijeratne and Subramanian, 2018*), or simply from steric interactions between the densely packed rod-like MTs. The volume fraction of MTs in the reconstructed spindles is 0.052±0.05, which is slightly above the volume fraction where the nematic phase is expected to become more stable than the isotropic phase, assuming a mean non-KMT length of 2.0μm as measured in the EM reconstructions (~0.04) (*Doi and Edwards, 1988*. *Brugués and Needleman, 2014*). Steric interaction between the MTs could therefore be enough to explain the observed nematic behavior. Studying spindles with depleted crosslinking proteins, lower MT density and perturbed KMT dynamics would help to determine the origin of these aligning interactions.

Since the observed KMT turnover (~2.5min) is rapid and the number of KMTs remains roughly constant throughout the course of mitosis (~30min; *McEwen et al., 1997*; *McEwen et al., 1998*), new KMTs must be recruited to kinetochores during metaphase. Prior work has shown that kinetochores are capable of both nucleating new KMTs de-novo and capturing existing KMTs from solution; however, it has been unclear which of these possible mechanisms is responsible for recruiting new KMTs to the kinetochore during metaphase (*Telzer et al., 1975*; *Mitchinson and Kirschaner, 1984*; *Mitchison and Kirschner, 1985a*, *Huitorel and Kirschner, 1988*; *Heald and Khodjakov, 2015*; *LaFountain and Oldenbourg, 2014*; *Petry, 2016*; *Sikirzhytski et al., 2018*; *David et al., 2019*; *Renda and Khodjakov, 2021*). We show that a model where metaphase KMTs nucleate at the kinetochore is consistent with the KMT ultrastructure observed by electron tomography and the tubulin dynamics observed in the photoconversion experiments. Our results would also be consistent with a model in which specifically MTs nucleate very near the kinetochore and are rapidly captured while they are still near zero length (*Sikirzhytski et al., 2018*).

The present work combined large-scale EM reconstructions, light microscopy, and theory to study the behaviors of KMTs in metaphase spindles. The behaviors of KMTs may be dominated by other processes at those different times. In the future, it would be interesting to apply a similar methodology to investigate the behavior of KMTs during spindle assembly in prometaphase and chromosome segregation in anaphase. Another interesting direction would be to apply a similar methodology to the study of spindles in other organisms. Previous EM reconstructions in *C. elegans* mitotic spindles have found a similar distribution of KMT lengths in metaphase (*Redemann et al., 2017*). Acquiring electron tomography reconstructions and dynamics measurements in different model systems would help elucidate whether the proposed KMT lifecycle is conserved across metazoans.

One significant feature of the nematic-aligned, nucleate-at-kinetochore model is that it provides a simple hypothesis for the mechanism of chromosomes biorientation: A pair of sister kinetochores, with each extending KMTs, will naturally biorient as the KMTs locally align along nematic streamlines that are flat near the center of the spindle. This initial nematic alignment of KMTs into the existing nematic spindle network would be consistent with observations that interactions between nascent KMTs and existing non-KMTs in the spindle are important for proper chromosome biorientation in prometaphase (*Renda et al., 2022*) Once bioriented, newly nucleated KMTs from either sister will

naturally grow towards opposite poles. MTs attached to the incorrect pole will turnover over and be replaced by newly nucleated MTs that will integrate into the nematic network, growing towards the correct pole. Once all the incorrect MTs have been cleared, tension generated across the opposite sisters will stabilize the existing, correct attachments. The nematic aligned, kinetochore-nucleated picture thus provides a self-organized physical explanation for chromosome bi-orientation and the correction of mitotic errors. It will be an exciting challenge for future work to test the validity of this model.

# Materials and methods

## Key resources table

| Reagent type (species) or resource | Designation | Source or reference | Identifiers | Additional information |
|---|---|---|---|---|
| Cell line (*Homo sapiens*) | HeLa Kyoto | Gerlich Lab, IMBA, Vienna Austria | - | - |
| Transfected construct (*Homo sapiens*) | pBABE-puro CENP-A:GFP | *Yu et al., 2019* | - | CENP-A C-terminally labeled with sfGFP; in retroviral vector with puromycin selection marker |
| Transfected construct (*Homo sapiens*) | pBABE-hygro SNAP:Centrin | This paper (Needleman Lab, Harvard) | - | Centrin N-terminally labeled with a SNAP tag; in retroviral vector with hygromycin selection marker |
| Transfected construct (*Homo sapiens*) | pJAG98(pBABE-blast) mEOS3.2:alpha tubulin | *Yu et al., 2019* | - | Alpha tubulin N-terminally labeled with mEOS3.2; in retroviral vector with blastcidin selection marker |
| Transfected construct (*Homo sapiens*) | pIRESneo-PA-GFP-alpha Tubulin | *Tulu et al., 2003* | - | Alpha tubulin N-terminally labeled with PA-GFP in a vector with a neomycin marker |
| Commercial assay or kit | SNAP-Cell 647-SiR | New England Biolabs | - | Catalog number S9102S |
| Software algorithm | Interactive spindle photoconversion analysis GUI (MATLAB 2020b) | This paper (Dryad) | - | - |
| Software algorithm | Photoconversion simulation package | This paper (Dryad) | - | - |
| Software algorithm | Photoconversion control and imaging | *Wu et al., 2016* | - | Controls custom confocal photoconversion for arbitrary geometry |
| Software algorithm | Polarized light microscopy control software | https://openpolscope.org/ | - | - |

## HeLa cell culture and cell line generation

HeLa Kyoto cells were thawed from aliquots and cultured in DMEM (ThermoFisher) supplemented with 10% FBS (ThermoFisher) and Pen-Strep (ThermoFisher) at 37 °C in a humidified incubator with 5% $CO_2$. The HeLa Kyoto cell line was a gift from the Gerlich Lab, Vienna and was authenticated using a Multiplex Human Cell Line Authentication test (MCA). Cells were regularly tested for mycoplasma contamination (Southern Biotech).

Four stable HeLa cell lines were generated using a retroviral system. A stable HeLa Kyoto cell line expressing mEOS3.2:alpha tubulin and CENPA:GFP was generated and selected using puromycin and blasticidin (ThermoFisher) (*Fürthauer et al., 2019*). An additional mEOS3.2:alpha tubulin and SNAP:centrin cell line was generated and selected using puromycin, blasticidin and hygromycin. A PA-GFP:alpha-tubulin and SNAP:centrin cell line was generated and selected using G418 and hygromycin. A final cell line expressing CENPA:GFP and GFP:centrin was generated and selected using puromycin and hygromycin.

## Spinning disc confocal microscopy and photoconversion

All photoconversion experiments were performed on a home built spinning disc confocal microscope (Nikon Ti2000, Yokugawa CSU-X1) with 488 nm, 561nm, and 647nm lasers, an EMCCD camera (Hamamatsu) and a 60x oil immersion objective. Imaging was controlled using a custom Labview program (*Wu et al., 2016*). Two separate fluorescence channels were acquired every 5s with either 500ms exposure, 488nm excitation, 514/30 emission for the photoactivated PA-GFP channel or 300ms exposure 647nnm excitation, 647 longpass emission in a single z plane for both channels. The PA-GFP was photoconverted using an Insight X3 femtosecond pulsed laser tuned to 750nm (Spectra Physics) and a

PI-XYZnano piezo (P-545 PInano XYZ; Physik Instrumente) to draw the photoconverted line by moving a diffraction limited spot across the spindle. The line was moved at a speed of 5µm/s with a laser power of 3mW (measured at the objective).Cells were plated onto 25-mm-diameter, #1.5-thickness, round coverglass coated with poly-d-lysine (GG-25–1.5-pdl, neuVitro) the day before experiments. Cells were stained with 500nM SNAP-SIR (New England Biolabs) in standard DMEM media for 30 min and then recovered in standard DMEM media for at least 4hr. Before imaging, cells were pre-incubated in an imaging media containing Fluorobrite DMEM (ThermoFisher) supplemented with 10 mM HEPES for ~15min before being transferred to a custom-built cell-heater calibrated to 37 °C. In the heater, cells were covered with 750µL of imaging media and 2.5mL of mineral oil. Samples were used for roughly 1hr before being discarded. During imaging, the focus of the microscope was adjusted to keep both poles in the imaging plane for the entire image acquisition.

## Quantitative analysis of photoconversion data

All quantitative analysis was performed using a custom MATLAB GUI. We first fit the tracked both poles using the Kilfiol tracking algorithm (*Gao and Kilfoil, 2009*) and defined the spindle axis as the line passing between the two pole markers. We generated a line profile along the spindle axis by averaging the intensity in 15 pixels on either side of the spindle axis. The activated peak from each frame was fit to a Gaussian using only the central 5 pixels. If multiple peaks were identified, the peak closest to the peak from the previous frame was used. The position of the peak was defined to be the distance from the center of the peak to the pole marker. To determine the height of the peak, we subtracted the height of a gaussian fit on the opposite side of the spindle from the height of the main Gaussian peak to correct for background and divided by a bleaching calibration curve.

## Bleaching calibration

HeLa spindles were activated by drawing 3 lines along the spindle axis from pole to pole. We then waited 5min for the tubulin to equilibrate and began imaging using the same conditions as during the photoconversion measurement (561nm, 500ms exposure, 5s frames; 647nm, 300ms exposure, 5s frames). We calculated the mean intensity inside an ROI around the spindle (*Figure 2—figure supplement 1a*) and plotted the average of the relative intensity of 10 cells. We subtracted off a region outside of the cell to account for the dark noise of the camera. We then divide our intensity vs. time curve by the bleaching calibration curve to produce a bleaching-corrected intensity curve to fit to a dual-exponential model.

## Polarized light microscopy (PolScope)

We measured the orientation of spindle MTs in living cells using an LC-PolScope quantitative polarization microscope (*Oldenbourg, 1998*; *Oldenbourg, 2005*) The PolScope hardware (Cambridge Research Instruments) was mounted on a Nikon TE2000-E microscope equipped with a 100x NA 1.45 oil immersion objective lens. We controlled the PolScope hardware and analyze the images we obtained using the OpenPolScope software package. To ensure that the long axis of the spindle lies in or near the image plane, we labeled the poles with SNAP-Sir and imaged the poles using epifluorescence while we acquired the PolScope data. In all subsequent analysis, we use only data from cells where the poles lie within ~1µm of each other in the direction perpendicular to the image plane. To average the orientation fields from different spindles, we first determined the unique geometric transformation (rotation, translation, and rescaling) that aligns the poles. We then applied the same transformation to the orientation fields and took the average.

## Fitting average MT angles to nematic theory

For each 3D reconstructed spindle, the positions of the MTs were first projected into a 2D XY plane (averaging along the z axis coordinateinto a single plane). Local MT angles were then averaged ($<\Theta = \arg(<\exp(2i\Theta)>)/2$) in 0.1µm by 0.1µm bins in the spindle-radial plane. Each spindle was rotated and averaged along the spindle axis every $\frac{\pi}{10}$ radians to produce a uniform projection.

We registered the three spindles obtained from electron tomography by rescaling them along the spindle and radial axis. We rescaled the spindle axis of each spindle, so that all three spindles had the same pole-pole distance. We rescaled the radial axis so that the width of the spindles, measured by the width of an ellipse fit to the spindle density in the spindle axis-radial axis plane, was the same.

We then averaged the three EM spindles together to produce *Figure 5A*. We similarly registered the PolScope images by rescaling the spindle axis using the pole-pole distance and the radial axis using the width of an ellipse fit to the spindle retardance image before averaging the cells together to produce *Figure 5B*.

The angles predicted by the active liquid-crystal model were found by solving the Laplace equation in the spindle bulk using a 2D finite difference method subjected to the tangential anchoring and defect boundary conditions. We imposed the boundary conditions by setting the MT orientation at the elliptical boundary to be tangent to the ellipse and radially outward within the aster defect radius at every finite difference method iteration. The model's geometric parameters were determined by fitting the predicted angles to the averaged EM data by minimizing a $\chi^2$ statistic. We first fit the height, width, and center of the elliptical boundary with the +1 point defects fixed at the edge using the averaged EM spindles. The elliptical boundary parameters were then fixed, and the position of the +1 point defects along the spindle axis and the radius of the defects were fit. The fit angles at each position in the z-direction for each XY pixel were projected into the XY plane and averaged to produce *Figure 5C*. The fit angles were weighted by the density of microtubules in the EM reconstructions during the averaging. The individual spindles were similarly fit by first fitting the elliptical boundary with the +1 point defects on the edge and then fitting the position and radius of the defects to produce *Figure 6—figure supplement 1*.

## Acknowledgements

The authors thank Gloria Ha and Dr. Che-Hang Yu for technical assistance with experiments; Dr. Reza Farhadifar and Dr. Sebastian Fürthauer for helpful comments on the manuscript. Research in the Needleman Lab is supported by the NSF-Simons Center for Mathematical and Statistical Analysis of Biology at Harvard (award number #1764269), and the Harvard Quantitative Biology Initiative. Will Conway was support by an NSF GRFP fellowship and an NSF-Simons Harvard Quantitative Biology Initiative student fellowship. Research in the Müller-Reichert laboratory is supported by funds from the Deutsche Forschungsgemeinschaft (MU 1423/8–2). RK received funding from the European Union's Horizon 2020 research and innovation program under the Marie Skłodowska-Curie grant agreement No. 675,737 (grant to TMR).

## Additional information

### Funding

| Funder | Grant reference number | Author |
|---|---|---|
| NSF-Simons Foundation | Center for Mathematical and Statistical Analysis of Biology at Harvard | William Conway<br>Colm P Kelleher<br>Hai-Yin Wu<br>Maya Anjur-Dietrich<br>Daniel J Needleman |
| NSF | Graduate Research Fellowship Program | William Conway |
| Deutsche Forsschunggemeinshaft | MU 1423/8-2 | Robert Kiewisz<br>Gunar Fabig<br>Thomas Müller-Reichert |
| European Union Horizon | Marie Sklodowska-Curie Agreement | Robert Kiewisz<br>Thomas Müller-Reichert |
| European Union Horizon | 675737 | Robert Kiewisz<br>Thomas Müller-Reichert |
| NSF-Simons Foundation | 1764269 | William Conway<br>Colm P Kelleher<br>Hai-Yin Wu<br>Maya Anjur-Dietrich<br>Daniel J Needleman |

| Funder | Grant reference number | Author |
|---|---|---|

The funders had no role in study design, data collection and interpretation, or the decision to submit the work for publication.

## Author contributions

William Conway, Robert Kiewisz, Conceptualization, Data curation, Software, Formal analysis, Validation, Investigation, Visualization, Methodology, Writing – original draft, Writing – review and editing; Gunar Fabig, Data curation, Software, Formal analysis, Validation, Investigation, Visualization, Methodology; Colm P Kelleher, Data curation, Formal analysis; Hai-Yin Wu, Maya Anjur-Dietrich, Data curation; Thomas Müller-Reichert, Supervision, Funding acquisition, Validation, Investigation, Project administration, Writing – review and editing; Daniel J Needleman, Conceptualization, Supervision, Funding acquisition, Methodology, Writing – original draft, Project administration, Writing – review and editing

## Author ORCIDs

William Conway ⓘ http://orcid.org/0000-0001-7532-4331
Gunar Fabig ⓘ http://orcid.org/0000-0003-3017-0978
Thomas Müller-Reichert ⓘ http://orcid.org/0000-0003-0203-1436

## Decision letter and Author response

Decision letter https://doi.org/10.7554/eLife.75458.sa1
Author response https://doi.org/10.7554/eLife.75458.sa2

# Additional files

## Supplementary files

• Transparent reporting form

## Data availability

Source code and data for all figures is uploaded to Dryad under https://doi.org/10.5061/dryad.69p8cz948.

The following dataset was generated:

| Author(s) | Year | Dataset title | Dataset URL | Database and Identifier |
|---|---|---|---|---|
| Conway W, Kiewisz R, Fabig G, Kelleher C, Wu H, Anjur-Dietrich M, Müller-Reichert T, Needleman D | 2022 | Self-organization of kinetochore-fibers in human mitotic spindles | https://doi.org/10.5061/dryad.69p8cz948 | Dryad Digital Repository, 10.5061/dryad.69p8cz948 |

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

## Appendix 1

### Computational Modeling Supplement

Here, we describe the details of the analysis, biophysical modeling, and simulations we performed to connect the structure of individual microtubules measured by electron tomography to the dynamics we observed in the photoconversion experiment. We first define the geometry of the simulated spindles. We then describe the details of the minus end speed prediction calculation and the simulation.

### Simulation spindle geometry

To generate idealized versions of each of the three reconstructed spindles for the simulations, we first separately fit each of the three spindles that were reconstructed by electron microscopy (EM) to an ellipse (*Figure 6—figure supplement 1*). We then fit the position and size of + 1 point defects to the director fields of each spindle with tangential anchoring at the elliptical boundary (see Methods). In the simulations, we considered the motion of photoconverted tubulin along discrete nematic streamlines. We placed these streamlines 0.5 μm apart at the center of the spindle along the radial axis (*Appendix 1—figure 1*). We found the trajectories of the streamlines by integrating along the director field predicted by a nematic model with tangential anchoring along the elliptical boundary and + 1 point defects at the poles.

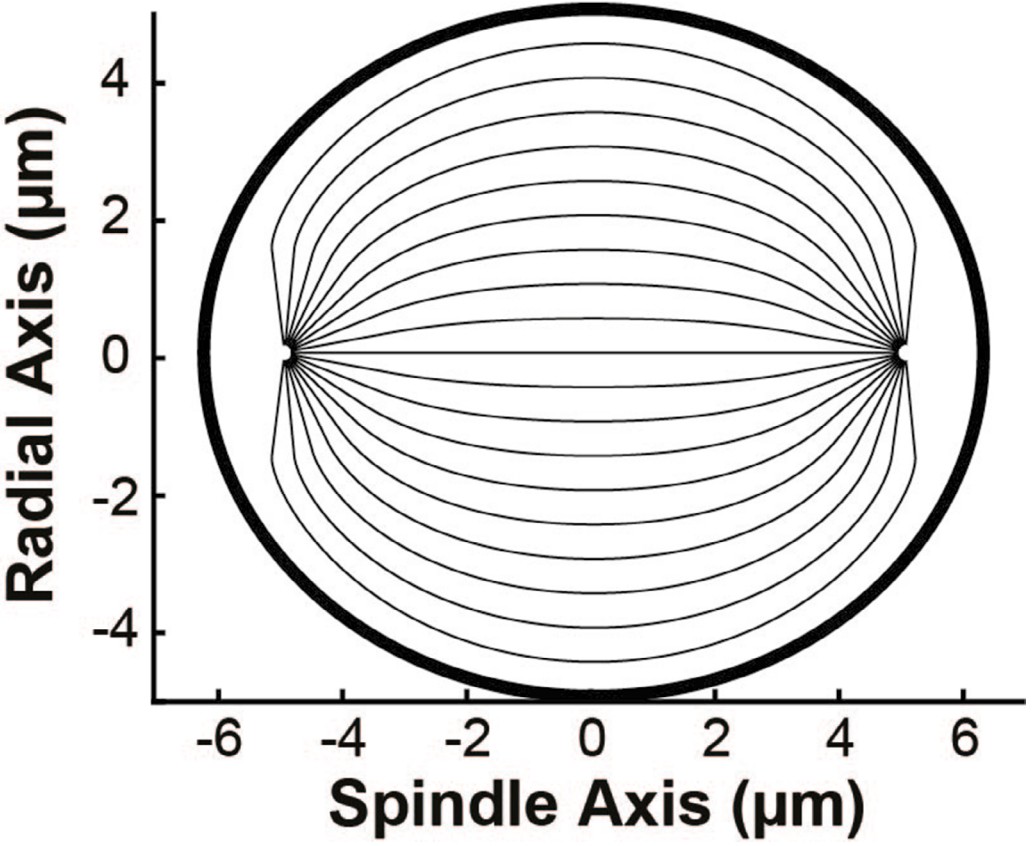

**Appendix 1—figure 1.** Sample geometry of spindle streamlines used in the simulation. Geometry of the spindle streamlines used in the simulations. The thin lines show the trajectories of nematic streamlines in the spindle bulk. The thick black line shows the elliptical boundary of the spindle.

### Measuring the minus end density distribution $n(s)$ from the EM reconstructed spindles

To measure the minus end density distribution $n(s)$ along streamlines, we first found the position of every kinetochore microtubule (KMT) minus end along the fit nematic streamlines in each of the three EM reconstructions. For each KMT minus end, we found the streamline it was on by integrating

along the fit nematic director field of that reconstructed spindle from the minus end's position to the pole (*Figure 6A*). We then calculated the distance $s$ between the KMT's minus end position and the pole along this streamline, with $s = 0$ for minus ends at the pole. A density histogram constructed by binning together all minus ends positions with respect to $s$ reflect two distinct effects: (1) variations of KMT minus end positions along $s$ within a k-fiber; (2) variations of the number of k-fibers along $s$. We wished to study the former, not the latter, so we focused on an alternative distribution: the density distribution of KMT minus ends along streamlines whose plus ends were upstream of that position. To construct that distribution, we first calculated the density of minus ends in a small bin within 0.1 µm of the pole along the streamline trajectories. To find the density of the minus ends in the next 0.1 µm bin upstream, we multiplied the KMT density in the first bin by the ratio of the number of KMT minus ends in the second bin with plus ends more than 500 nm upstream from the second bin to the number of KMT minus ends in the first bin with plus ends more than 500 nm upstream from the second bin. We then iterated this procedure along the streamline trajectory to produce the density distribution of KMT minus end along streamlines whose plus ends were upstream of that position (*Figure 6B*).

## Deriving mass conservation equation for KMT minus ends to calculate the KMT minus end speed $v(s)$

We performed a mass-conservation flux analysis on the KMT minus end density distribution (measured from the EM reconstructions) to predict the speed of the KMT minus ends throughout the spindle (*Figure 6C*). We assumed that the KMTs in metaphase are in steady state and move along streamlines, which implies that the fluxes associated with KMT gain, motion and loss must balance at every position along the streamlines. We considered a region along a streamline between positions $s$ and $s + ds$, and defined the fluxes associated with KMT minus end gain, motion, and loss in this region as:

### Gain
New KMTs join the fiber with their minus ends at position $s$ along streamlines at rate $j(s)$. The form of $j(s)$ depends on the choice of a model for how KMTs are recruited to the kinetochore and is discussed in more detail below.

### Motion
KMT minus ends move into the region with flux $v(s + ds)n(s + ds)$ and move out of the region with flux $v(s)n(s)$, where $n(s)$ is the density of KMT minus ends at position $s$, and $v(s)$ is the speed of KMT minus ends at position $s$. Subtracting these terms and taking the limit $ds \to 0$ gives the motion flux as $v(s)\frac{dn}{ds} + \frac{dv}{ds}n(s)$.

### Loss
KMTs detach from the kinetochore and depolymerize at rate $r$. Our photoconversion experiments revealed that the lifetime of KMTs was independent of their position in the spindle bulk (*Figure 3G*), so we took $r$ to be constant (i.e., independent of $s$). We set $r$ to be the inverse of the average lifetime of KMTs in the spindle bulk measured in the photoconversion experiments: i.e., $r = 0.4$ min⁻¹.

Since the KMT minus ends are in steady state, these three fluxes must sum to zero everywhere. This gives us a steady state mass conservation equation:

$$j(s) + v(s)\frac{dn}{ds} + \frac{dv}{ds}n(s) - rn(s) = 0 \tag{2}$$

## Defining the $j(s)$ gain flux term

The form of the $j(s)$ gain flux term depends on the KMT recruitment model (*Figure 6D*). If all KMTs result from de novo nucleation at kinetochores (i.e., the nucleate at kinetochore model), then, by assumption, $j(s) = 0$ at all locations in the spindle bulk. Alternatively, if KMTs result from non-KMTs that bind the kinetochore (i.e., the capture from spindle model), then $j(s) \neq 0$. For a non-KMT to bind a kinetochore, it must first be nucleated and then grow far enough to contact a kinetochore. Non-KMTs turnover in ~ 0.25 min and move at a speed of ~ 1 µm/min (*Figure 3*), so we estimate that they travel only ~ 0.25 µm before depolymerizing. Thus, since non-KMTs are not expected to significantly move over their lifetime, we take the inferred density of non-KMT minus ends along streamlines, $n_{NK}(s)$ (*Figure 6—figure supplement 2*), as an estimate of the non-KMT nucleation rate along streamlines. The length distribution of non-KMTs is observed to be exponential, with a

mean length of $l_{NK} = 2.0 \pm 0.05\ m$ (*Appendix 1—figure 2*). Thus, if a non-KMT nucleates at position $s$ along a streamline, the probabilities that it grows far enough to reach a kinetochore located at position $s_0$ is proportional to $e^{-\frac{(s_0-s)}{l_{NK}}}$. Taken together, this leads to $j(s) \propto n_{NK}(s)e^{\frac{s}{l_{NK}}}$ for the capture from spindle model, where the dependence on the position of the kinetochore is absorbed into the constant of proportionality.

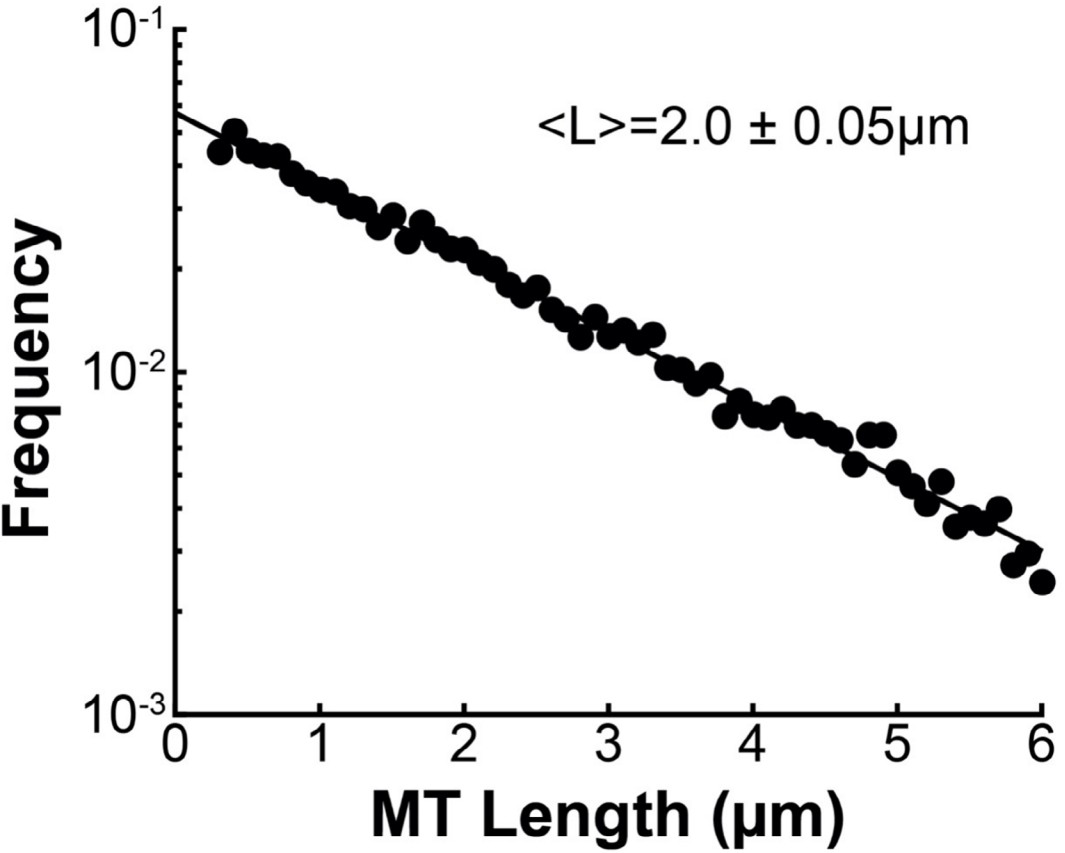

**Appendix 1—figure 2.** Length distribution of non-KMTs in the spindle. Binned histogram of the lengths of non-KMTs in three reconstructed mitotic HeLa spindle. Black dots: electron microscopy data; black line: exponential fit. Mean MT length is 2.0±0.05 μm.

## Integrating the mass conservation *Equation 2* to find minus end speed predictions

We set a no-flux boundary at the pole to integrate the mass conservation *Equation 2*. The no-flux condition at the pole implies that either $n(0) = 0$ or $v(0) = 0$, reducing the mass conservation equation to:

$$n(0 + ds)v(0 + ds) -\ r\,n(0) =\ 0 \qquad (3)$$

$n(0) = 0$ therefore requires that $n(0 + ds)v(0 + ds) = 0$ which reproduces the no-flux boundary condition at the position $ds$. Iterating this procedure produces a trivial solution that $n(s) = 0$ everywhere. Since we observed a non-zero minus end distribution, we used the $v(0) = 0$ condition instead. Using this $v(0) = 0$ condition we integrated *equation (1)* numerically to find the KMT minus end speed predictions from the nucleate at kinetochore and capture from spindle recruitment models (*Figure 6E*).

## Simulated 2D confocal imaging of a photoconverted line

We simulated the motion of tubulin after photoconversion in both KMTs and non-KMTs and the reincorporation of depolymerized tubulin in the simulation spindle for each of the three reconstructed cells (*Appendix 1—figure 1*). We assumed that the dynamics were the same along all streamlines

and simulated the motion of photoconverted tubulin in KMTs and in non-KMTs along a streamline. We calculated the tubulin profile along the spindle axis from each streamline and then combined the results from the different streamlines. We finally added a background profile from depolymerized photoconverted tubulin that reincorporated throughout the spindle to produce a final line profile for analysis.

## KMTs

We simulated the gain, motion, and loss of individual KMTs along streamlines at discrete timesteps. At each simulation timestep, we generated newly recruited KMTs with Poisson statistics. The KMT plus end positions were selected from the distribution of kinetochores along streamlines (***Figure 7— figure supplement 1***). For kinetochore nucleated KMTs, the minus ends started at the same location as the plus end. For spindle captured KMTs, the minus ends position was drawn from the probability that a microtubule would nucleate times the probably it would reach the kinetochore ($j(s) \propto n_{NK}(s)e^{\frac{s}{l_{NK}}}$). Newly encorporated tubulin polymerized at the KMT plus ends while the minus ends move backwards along the streamline towards the pole with an experimentally inferred speed $v(s)$. In the spindle bulk, the minus ends moved at the same speed that the tubulin incorporated at the plus end. When KMT minus ends entered the pole region, at $s_p = 1.5 \mu m$ upstream from the pole, tubulin continued to polymerize at the same rate as at the boundary, but the minus ends began to depolymerize at speed $v_{tread}(s) = [v(s_p) - v(s)]\theta(s_p - s)$, where $\theta(s)$ is the Heavyside step function. The tubulin in a KMT therefore moved at speed $v_{tub}(s) = v(s) + v_{tread}(s)$ while the minus ends moved at the experimentally inferred speed $v(s)$. After an exponential drawn lifetime with mean $1/r = 1/0.4$ min = 2.5 min, the KMTs detach from the kinetochore and are removed from the simulation.

To simulate the motion of photoconverted tubulin, we calculated the intensity of the photoconverted tubulin along streamlines with a Cauchy profile $I(x) = \frac{1}{1 + \left(\frac{x - x_0}{w}\right)^2}$. Based on fits to the experimental line profile immediately after photoconversion, we set $w = 150$ nm (***Figure 7—figure supplement 2***, ***Appendix 1—table 1***). The KMTs were pre-equilibrated for 20 minutes of simulation time before the simulation line was drawn to ensure the KMTs were in steady state. We projected the simulated photoconverted tubulin intensity along the spindle axis and summed the contribution of each KMTs along each of the spindle streamlines to produce a KMT line profile.

## Non-KMTs

For the non-KMTs, we calculated the initial intensity of photoconverted tubulin along a streamline by multiplying the density of non-KMTs along the streamline by a Cauchy intensity profile along the spindle axis. We then translated the entire profile along the streamline towards the pole at a uniform speed equal to the speed of KMT minus ends where the line was drawn $v(s)$. The profile height decayed at a rate $r_{NK} = 4$ min$^{-1}$ (***Appendix 1—table 1***) measured in the photoconversion experiment (***Figure 3H***). Changing the simulated speed of the non-KMTs did not significantly impact the measured speed of the KMTs after the final analysis (***Appendix 1—figure 3***). Like the KMTs, we simulated the motion of the peak along each streamline, projected onto the spindle axis and then summed the streamlines together to produce a line profile. We added the KMT and non-KMT profile together, normalizing the profiles so that the KMT to non-KMT intensity ratio was 4:1.

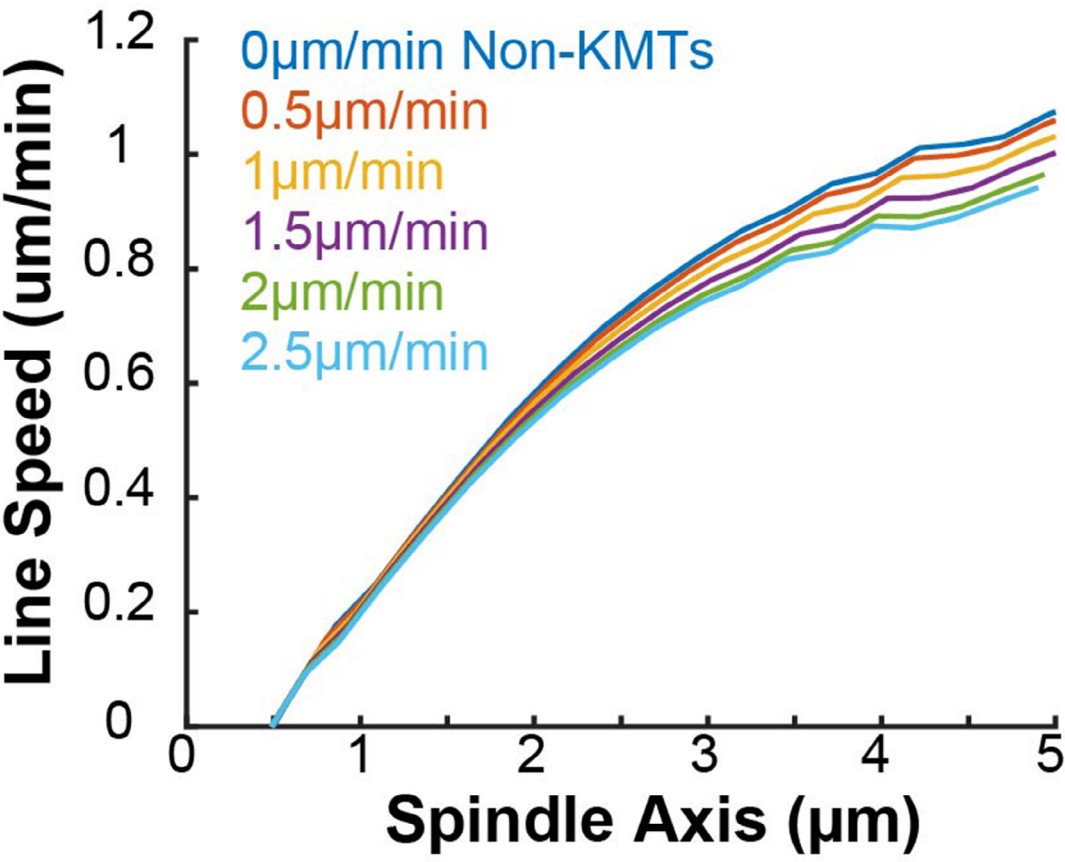

**Appendix 1—figure 3.** Predicted photoconverted line speed for various uniform non-KMT motion speeds. The speed of the non-KMTs was varied (assorted colors) in 0.5 µm/min increments in a 2D confocal imaging spindle simulation.

## Reincorporated Background

Finally, we included the contribution of reincorporated tubulin from photomarked microtubules that depolymerized. We modeled the background as a constant tubulin profile whose height exponentially approached a plateau value $h(t) = A\left[1 - e^{-\frac{t}{\tau_{bkgd}}}\right]$. We determined the profile of reincorporated tubulin background from the average profile of tubulin along the spindle axis in cells with an mCherry:alpha-tubulin marker. (*Appendix 1—figure 4*). The height and timescale of the background profile were found using the photoconverted tubulin signal at the opposite pole in the photoconversion experiments. We fit a Gaussian to the photoconverted tubulin profile at the opposite pole. We then fit the height of the peak over time to determine the height and timescale of the background profile (*Appendix 1—figure 5*). The background incorporation took $\tau_{bkgd} = 60s$ and leveled off to $A = 6\%$ of the height of the original peak (*Appendix 1—table 1*).

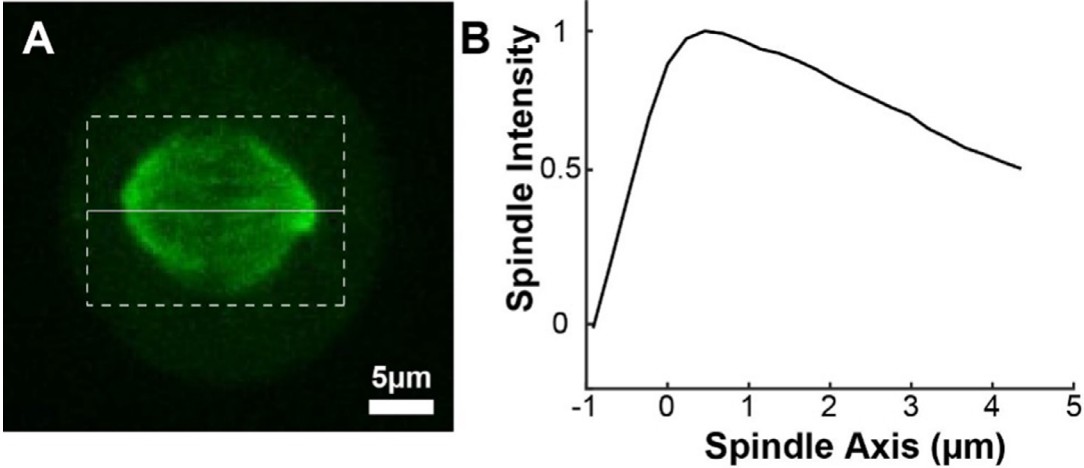

**Appendix 1—figure 4.** pindle background profile. (**A**) Sample representative spindle image (Green: mCherry:tubulin). (**B**) The intensity of the tubulin marker projected onto the spindle axis and then averaged for n=72 half spindles. The spindle axis x=0 is located at the pole.

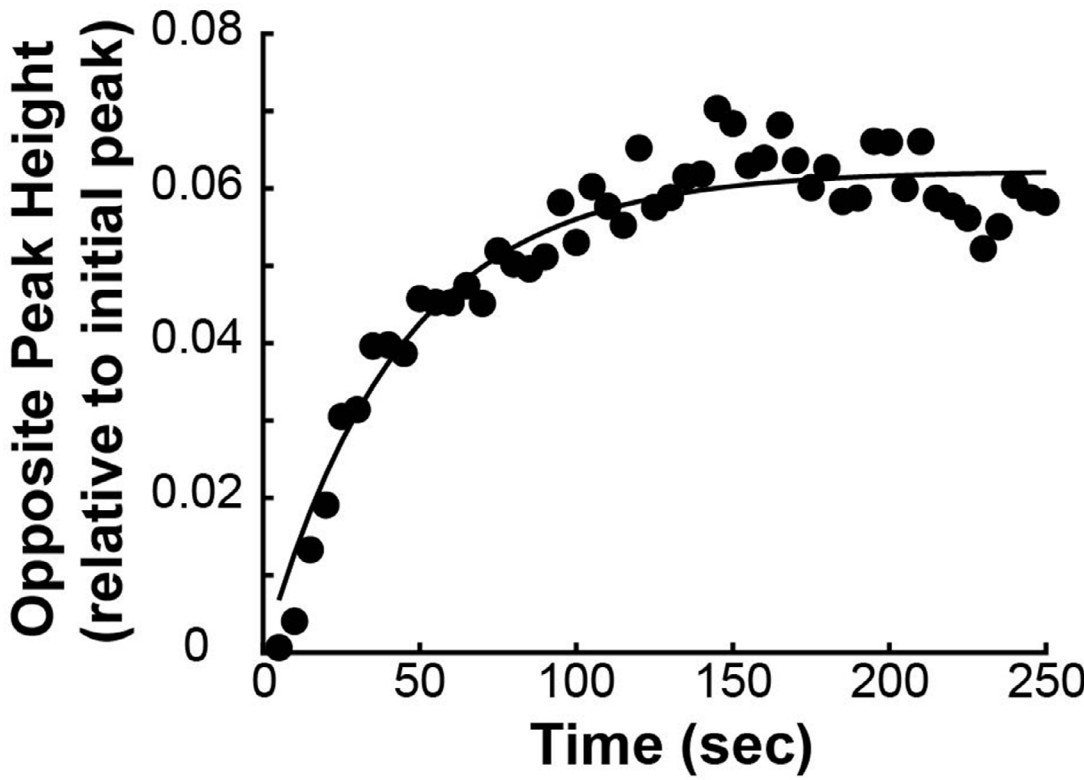

**Appendix 1—figure 5.** Height of the opposite pole over time. The peak height averaged from n=5 spindles displaying a clear opposite peak (black dots) is fit to an exponential (black line).

## Fitting the motion and decay of the simulated peak

We summed the contribution of the KMTs, non-KMTs and background together and then convolved the line profile with a Gaussian with width 250 nm to simulate the microscope point spread function. We then processed our simulated curves through the same algorithm we used to fit the experimental curves (see Methods): fit the pixels near the top of the peak to a Gaussian, fit the center of the Gaussian to a line to determine the velocity, fit the height of the Gaussian corrected for background to a dual-exponential to determine KMT and non-KMT stability.

## Error Analysis

We repeated the simulations for each of the three EM-reconstructions. We used the measured KMT minus end distribution and spindle geometry from each individual spindle. We took the mean of the predictions from the three cells to find the model predicted speed of the photoconverted line (*Figure 7B*). We then took the standard error of the mean for the speed predictions from all three spindles to find the error in our model predictions.

## 3D Spindle Simulations

We simulated the gain, motion, and loss of discrete KMTs in each of the three reconstructed cells in 3D. At each timestep, we nucleated new KMTs at kinetochores by placing both the plus and the minus end at the same position within 200 nm of the position of a kinetochore in the reconstruction. We then moved the minus ends of the existing KMTs towards the pole at the experimentally inferred speed $v(s)$ along nematic streamlines in 3D. The nematic streamline for each KMT were found by calculating the 2D nematic streamline from the plus end position in the spindle-radial axis plane and rotating the spindle-radial axis plane about the spindle axis to the kinetochore position. This procedure produced a 3D streamline that was flat in the theta direction. When the KMT minus ends cross the pole boundary at $s_p = 1.5m$ from the pole along a streamline, the minus ends begin to depolymerize causing tubulin to treadmill through the spindle at a speed $v_{tread}(s) = [v(s_p) - v(s)]\theta(s_p - s)$, as in the 2D case. The KMTs detach from the kinetochore at a rate $r = 0.4$ min$^{-1}$ and are removed from the simulation.

We compared the predicted lengths, orientations, and dynamics of the simulated and experimentally measured KMTs. We measured the lengths of the simulated KMTs from the distance between the plus and the minus end along the streamline trajectory. To compare the orientations of the simulated and reconstructed KMTs, we divided each KMT into short 100 nm subsections and projected the subsections onto the spindle axis. We compared the fraction of the 100 nm subsection lengths along the spindle axis in the simulation and experiment. We drew a plane of photoactivation tubulin perpendicular to the spindle axis with a Cauchy profile. We projected the tubulin intensity in a thin 1 µm confocal z-slice onto the spindle axis to produce a line profile. The center position, width, and exponent of the profile were fit to a sample photoconverted line profile at t=0 min. We then tracked the converted tubulin in the spindle over 60 s of simulated time and reprojected the confocal slice onto the spindle axis to compare the line profile with experimental converted line profile at t=60 s.

**Appendix 1—table 1.** Parameters values and sources.

| Simulation Parameter | Value | Source |
|---|---|---|
| KMT Trajectories, $t(s)$ | - | Nematic Theory (*Figure 5* and. 6 A) |
| KMT Velocity $v(s)$ | Varies | Mass Conservation Analysis (*Figure 6E*) |
| KMT Stability, $r$ | 0.4 min$^{-1}$ | Photoconversion (*Figure 3G*) |
| Non-KMT Mean Length, $l_{NK}$ | 2 µm | Electron Microscopy (*Appendix 1—figure 2*) |
| Photoconverted Line Width, $w$ | 150 nm | Converted Line Profile (*Figure 7—figure supplement 2*) |
| Background Height, $h_{bkgd}$ | 0.06 | Opposite Peak Height (*Appendix 1—figure 5*) |
| Background Rise Time, $\tau_{bkgd}$ | 60 s | Opposite Peak Height (*Appendix 1—figure 5*) |

