## [Editor Report]

Conway and colleagues use a combination of experiments and theory to test models for the dynamics of kinetochore-fibers during metaphase in mammalian mitotic spindles. Their work is consistent with a model where kinetochore-fiber turnover is due primarily to the nucleation of microtubules at kinetochores, rather than from a "search-and-capture" of microtubules initiated elsewhere. This work should be of interest to experimentalists and theorists broadly interested in the control of form and in cell division.

---

## [Decision Letter]

**Decision letter after peer review:**

Thank you for submitting your article "Self-organization of kinetochore-fibers in human mitotic spindles" for consideration by *eLife*. Your article has been reviewed by 3 peer reviewers, and the evaluation has been overseen by a Reviewing Editor and Anna Akhmanova as the Senior Editor. The following individuals involved in review of your submission have agreed to reveal their identity: J Richard McIntosh (Reviewer #1); Helder Maiato (Reviewer #3).

*Essential Revisions (for the authors):*

1) The authors need to consider in their analyses the almost spherical shape of the HeLa spindle and the impact of the resulting MT curvatures in 3D on the way photoactivated fluorescence will appear to change with time as KMTs flux when observed in a plane that contains the spindle axis. Clarity is required on whether the spindle is considered a 2 or a 3 dimensional object. SDCF imaging is essentially planar; photoactivation of fluorescence is not. The slowing of flux near the poles might be a result of this feature of experimentation and the shape of a human spindle: the apparent slowing may simply be a result of observing the fluorescence of a 3D object in 2D. Similar 2D/3D issues emerge in the projections of ET data to make to 2D color map that is compared with Pol optics and theory. Though better explanations are required to understand how this was done, the impression is that the theory is in 2D, (Laplassian not = 0 in 3D).

2) The authors also need to give proper consideration to the impact of this same spherical geometry on the distribution of birefringence observed with a PolScope, which uses a high numerical aperture objective.

3) The model for spindle structure based on "active liquid crystal theory", similar to work from this same group on spindles formed in frog egg extract has two major issues. (A) It does not seem to have the interpretive power attributed to it. (B) A more accessible description of the active liquid crystal theory work in Figures 4-5 is required. The authors summarize this in lines 220-223, and the logic there is not clear. This seems like an almost tautological conclusion the way it is presented.

4) The authors need to clear up confusion in terminology and apparently in thinking, about the difference between MTs whose growth is initiated at the kinetochore vs. those whose dynamics are affected by tubulin addition at the kinetochore. This confusion is compounded by a presentation that conflates what goes on during spindle formation and what happens in a metaphase spindle, where one is looking at the dynamics and turnover of an already formed K-fiber, vs. the formation of such a fiber as the spindle develops. Ibn other words, the manuscript analyses KMT turnover in metaphase cells and uses this to address the question of where KMTs come from. However, the source of KMTs may change with time in mitosis, so the origin of KMTs during prometaphase may be different from that in metaphase. This is important because it is the early KMTs that initiate chromosome attachment and guide most of metakinesis. However,the observations in this manuscript pertain to KMT turnover, not initial KMT formation.

5) The authors show regional variation of flux in human spindles, whereby KMTs slow down near poles. The authors should discuss this finding in the context of existing models showing slowed speeds near poles, such as in Burbank et al. 2007 slide-and-cluster model. Dscussion on possible mechanisms for slower flux closer to poles in human spindles. should also be added.*Reviewer #1 (Recommendations for the authors):*

Their model for spindle structure is based on the condensed matter physics of an "active liquid crystal". This model was initially developed in previous papers from this same group, but here it is revised to correspond to the structure of a human spindle and to interpret the structural and kinetic data mentioned above. In this paper the model is poorly described, so this part of the paper is hard to read with confidence. However, several aspects of the model appear to be misleading or incorrect. For example, the theory described resembles the familiar equilibrium Frank orientational elastic theory of nematic liquid crystals with prescribed boundary conditions (tangential anchoring at the spindle boundary) and with two point-defects, which the authors associate with the spindle poles. Although details are lacking, this model appears to be the same as that developed in Brugués and Needleman 2014, namely a one-constant approximation with equal splay and bend elastic constants, with the additional assumptions that the spindle is twist-free and axially symmetric. The authors claim that within this approximation the MT orientation field θ(r) in the spindle satisfies the Laplace equation in this case (∇^2 θ=0), but this is not correct. In cylindrical coordinates, the equilibrium MT orientation field θ(ρ,z) in the one-constant approximation is the solution of

∇^2 θ(ρ,z)=1/ρ^2 sinθcosθ

see, for example, M. Press and A. Arrott, "Elastic energies and director fields in liquid crystal droplets, I. Cylindrical symmetry", Journal de Physique Colloques 36, (C1), C1-177 – C1-184 (1975). Apparently, both here and in the earlier paper by Brugués and Needleman, the spindle was modeled as a two-dimensional object, in which case the director field is indeed the solution of the Laplace equation in the one-constant approximation. Two significant limitations of the current paper are: (1) that the discussion switches back and forth from 3D to 2D without adequate signals to the reader and (2) that details of the theory are not in the manuscript, for example details about how the boundary conditions were imposed.

It is noteworthy that the ambiguity about two- or three-dimensional treatments is a persistent problem in several aspects of the paper. The tomographic reconstructions, which are being related to the data from light microscopy, are unquestionably 3D. The polarization optical data are apparently 2D, because the PoleScope used has a high numerical aperture objective, and there is no mention of collecting data from serial optical sections. Likewise, the images for analysis of fluorescence redistribution after photo-activation are 2D, since they were collected with a spinning disk confocal microscope (SDCM). In the paper's section where the tomographic data are being projected, it appears (the writing is unclear) that all spindle MTs are projected onto a plane and this projection is then compared with the images from polarization microscopy. If this is in fact what was done, the two structures being compared are not the same thing, and the apparent similarity in distribution of angle-revealing colors is a coincidence.

Moreover, the activation of fluorescence was done by moving the photoconverting laser beam across the spindle, and this beam certainly activates fluorescence above and below the plane of observation. (The beam is presumably a diffraction limited spot, whose length along the optic axis is greater than its width in the plane of view. Moreover, although photo-converting intensity drops with the square of distance from the image plane, the time of irradiation by the moving beam increases linearly, so fluorescence at significant distances from the plane of focus will experience photo-conversion.) This leads to a key problem in data interpretation. The slowing of MT migration with proximity to the pole probably derives simply from the large number of MTs that are oblique to the optical section observed in the SDCM, particularly in the vicinity of the pole. From this point of view, KMTs flux away from kinetochores at a constant rate, but the apparent rate of KMT motion along the spindle axis changes as one nears the pole, simply because in that region most MTs are at high angles to that axis. This interpretation has the value that different parts of a given MT all move through space at the same rate, not different rates, as implied by the language of paper.

Another problem with the paper is imprecise and confusing use of language. For example, the name "theta" is given both to an axis and an angle; no distinction is made between them when the word is used in the text, making for very hard reading. The word "initiation" is used to describe KMT behavior in the vicinity of the kinetochore, when really elongation (or polymerization) is what the data report. The paper cites previous work that examined the issue of KMTs arising from capture of MTs initiated at the pole or initiation of MTs at the kinetochore, but these papers were looking at KMT formation during spindle development, not at metaphase. Flux of KMTs is a result of kinetochore-proximal tubulin addition to existing MTs and does not relate to issues of KMT initiation. This whole section of the paper is written in such a way that it is difficult to understand what the authors are trying to say.

Although the model, the data from electron tomography, and structural information from high quality polarization microscopy all appear to agree well in their descriptions of MT orientations, limitations in the paper's consideration of the full three-dimensional structure of the spindle and issues of depth of field cloud the interpretations given. The possible role of the spindle's 3D shape in offering a simple explanation for the slowing of fluorescence as it approaches the pole is not discussed, so the more complex model that is discussed does not have the intellectual force one might hope for. In sum, this paper does not contribute very much to our understanding of metaphase structure or spindle MT dynamics.

Detailed comments

Ln 18 The word "many" is used here and at several other places in the paper to describe the number of KMTs that do not extend all the way to the metaphase spindle pole. The numbers, both in this paper and in Kiewisz et al. show that half go that far, half do not. Since you have the data, why not state it?

Ln 56 The most important paper about initial capture of pole-initiated MTs by prometaphase kinetochores is work from the Rieder lab, which should be cited.

Ln 62 This long list of observations includes many that do not pertain to the statement made. However, the initial observations that support the statement (Data from Mitchison's observations with injected biotinylated tubulin, then with photo-activatable tubulin) are not mentioned.

Ln 85 suggest adding "here" at sentence beginning to mark the transition from discussion of other people's work to the description of this paper.

Ln 93 Nucleate seems to be the wrong word. This is a big issue as the paper goes forward. To me, and I think most people in the field, the initiation of a MT is the event that starts its growth, whether this involves a seed, like the γ tubulin complex, or simply the coming together of enough tubulin oligomers to get a MT started. This is distinct from MT growth (elongation by tubulin polymerization at one end or the other). KMT turnover could be accomplished by dissociation of KMT plus ends and then true MT initiation at the kinetochores or by the mechanism sketched here (spontaneous initiation near the kinetochore, followed by kinetochore binding of the MT plus end) or by kinetochore binding of the plus end of a fully formed non-KMT initiated elsewhere. These distinctions are not clearly made in this paper, and the relevance of MT initiation to KMT flux is not spelled out.

Ln 624 Legend to Figure 1: the minus ends are not "shown" in black, they are marked with a black circle.

Also, the term Pole is used here without specific definition. For these measurements, I presume you are using the middle of the mother centriole, or perhaps the center of a Gaussian spot from a fluorescent label, but you should be specific. Figure 1C. In the PDF, there is marked aliasing in each curve representing the KMTs. Find a better mode of display? Also, the representation of the metaphase kinetochore in the upper right doesn't look like any HeLa metaphase kinetochore that I have seen. Note also that the fiber shown in this diagram does not show any of the decrease in diameter toward the pole that is described in Kiewisz et al. Is this one really representative? Moreover, word usage in the insert is very cryptic. I presume that the probability you are talking about is that of finding a KMT plus end at that distance from the pole whose minus end lies within 1 μm of the Pole (to be defined). Please clarify. In D, the term, "captured from spindle bulk" again uses undefined terms. I expect you mean the capture of a non-KMT, turning it into a KMT. Please clarify.

Ln 128 Images in Figure 2. Why does the polar staining with the marker for CENP-A increase so dramatically as soon as your photo-converting irradiation is done? Has the camera gain been increased to accommodate bleaching? Note also that the "line" of photo-activated tubulin appears to spread rather markedly throughout the half-spindle within 5 sec. This is not so marked in the traces of fluorescent intensity as it is in the images. Have the display parameters for the images been altered between the 0s and 5s images? The spreading is even more marked at 25s suggesting this might be the case. If so, more realistic images are called for.

Ln 633 You have a very compact way of stating the wavelength for excitation and emission in your fluorescence observations. The first time you use it, I suggest stating explicitly what each number means, so the reader is not obliged to infer it from your words.

Ln 635 You refer to the light that stimulates fluorescence as 40 nm. Typo?

Ln 636 You say, "Line profile pulled from the dotted box shown in B." Are the graphs based on a scan simply down the line shown, or are you using a slit oriented perpendicular to the direction of scanning (the line) whose length is the vertical dimension of the box. Please be more specific.

This legend lacks periods at the ends of each unit of information.

Ln 171 This appears to be an intriguing observation, and to my knowledge novel. Differences in MT flux over time in mitosis are well characterized, but difference at one mitotic stage, depending on position could be very interesting. After a close look at Figure 3, however, I am concerned about both the robustness of the data supporting this observation and their interpretation. Finding the position of the fluorescence peaks looks non-trivial, because the shapes of the traces change so much, both with position and time. Since this phenomenon is really intriguing, your analysis of the raw data needs better support. Some supplementary figures here, showing exactly how you determined the position of the peak and the uncertainty in those positions is needed to make the claim convincing.

Ln 179 Here, I'm confused. You mention projecting all MTs on "a" spindle-axis, radial-axis plane. In Figure 4s3 we are supposed to learn what you mean by this. It is easy to see that there is one spindle-axis. However, given the rough cylindrical symmetry of the spindle about this axis, one must either think of one radial axis and define a single plane that contains both these axes or consider a large number of radial axes at different orientations (all perpendicular to the spindle axis), in which case one is going to project the MTs onto many different planes. The latter would preserve the shape of curving MTs quite well, the former very badly. It seems as if you projected the 3D data from ET onto a single plane and from here on think of the spindle as a 2D object. However, the text is not sufficiently explicit about this projection to know what you have done. If your average orientations are based on projecting all MTs onto a single plane, I can't see that the parameter has any value, because you have distorted the spindle fibers so dramatically in projection. Moreover, this planar projection now contains a large number of MTs that are not imaged by a single focal plane image in the PolScope. Thus, if these colored displays of MT orientation look the same, it demonstrates that they have no value.

Figure 4, legend. Your text (ln 681) theta is not defined until Figure 4s3. Moreover, when you say, "averaged over all theta", where theta is the name you have given to both an axis and an angle. This needs to be clarified. Moreover, the quantities in question obviously vary with z and ρ, so this doesn't make much sense. Figure 4D you give no values for the scalar nematic order parameter. Please be more specific. Most of these comments also pertain to Figure 4s2. In addition, note that the color bar that relates image hue to angle is so small it is hard to read, and it is missing from all these supplementary figures.

Ln 181 There is where θ is first introduced as a MT orientation parameter, but it is never defined.

Ln 188 In this statement of the equation for S you use θ for the angle. Previously you have called a standard Cartesian axis theta. Suggest renaming to avoid confusion. To understand the use of this equation in the context of the spindle, we need to know how you are measuring <θ>. Is <θ> determined at each position in the spindle volume and each specific θ compared with that? The writing doesn't spell out what you have done.

Ln 200 How does the LC-Polscope cope with birefringence that is outside the depth of field of the lens being used? I worry that the pleasing similarities between the EM data and the polescope images are not as informative as they appear, due to the ways you have projected the EM data and observed birefringence in the LM. (This point is expanded in comments on Methods.)

Ln 704 The words used here to describe the model are not sufficient to spell out what it is. Clarify?

Ln 204 These two paragraphs contain a brief description of the theory used to model the orientation field of MTs in Hela spindles. The same theory was applied to *Xenopus* spindles in earlier studies (Brugués and Needleman 2014, Oriola et al. 2020). As mentioned above, the development here lacks both specifics and clarity. It also reflects a general problem of this paper, the issue of whether the problem at hand is being considered in two dimensions or three. This issue must be clarified throughout the paper.

Ln 211 The model requires "a boundary" and two "defects". As stated, these are undefined. The relationship between the defects and the centrosomes is mentioned, but the nature of the boundary is left unspecified. As such, it sounds as if the model is set up to mimic spindle structure but without physically meaningful relations to the biological reality in the cell.

Ln 242 This paragraph describes phenomenology of MT behavior in the language you have introduced (streamlines), but each phenomenon mentioned is equally well described by a more conventional view of the spindle, based only on MTs whose plus ends may bind and dissociate from kinetochores, whose walls may be bridged to adjacent MTs though motors and non-motor MAPs, and whose plus-end growth may promote flux of the MT's center of mass toward the spindle pole. Your streamlines are beautifully represented by the trajectories of spindles MTs in the electron tomograms, so the value of the condensed matter terminology seems limited.

Ln 257 I think there is a fundamental problem with this formulation. The test of the model will be the way in which it fits the distributions of spindle fluorescence as a function of time after excitation of fluorescence. The model very reasonably talks about j(s) as the rate at which non-KMTs become KMTs as a result of kinetochore binding by their plus ends. However, the fluorescence of non-KMTs changes much faster than that of KMTs, because this is the population that displays fast fluorescence loss. Therefore, when a non-KMT binds to a kinetochore, it does not bring to the KMT population a fluorescence commensurate with a normal KMT. It has already exchanged many subunits and become less fluorescent. This is, of course, the phenomenon you used to calculate exchange rates. How is this aspect of the phenomenology accounted for in the analysis?

Ln 266 Here you state that the rate of KMC detachment based on photoconversion experiments is r = 0.4 min-1. Is it obvious how this number is obtained? I don't recall seeing this explained.

Ln 272 "If all KMTs were nucleated at kinetochores…" is a statement that reveals a complexity not considered in this development. All data come from metaphase spindles and all consideration is directed to such structures. However, every metaphase spindle has a history; it formed during prometaphase through the action of many processes and during that formation, it displayed shapes that are quite different from the spindle at metaphase. Moreover, spindles can form normal metaphase structures by a variety of pathways, depending in part on the degree of separation of the centrosomes at the time of nuclear envelope breakdown and rapid MT growth. MTs that become KMTs early may be initiated in ways that differ from MTs that become KMTs later in mitosis. Thus, the language in this paragraph must be changed to focus simply on the metaphase spindle.

Ln 280 I don't understand how the spatial distribution can be a proxy for the non-KMT nucleation rate. Do you mean the spatial variation of that rate? This, of course, assumes that minus ends of KMTs are always due to MT initiation at the kinetochore, rather than initiation in the body of the spindle. This point is a major weakness of the paper. Considerable data in the literature demonstrates that there is Augmin-dependent initiation of MTs, often in association with KMTs, and this process adds many minus ends to the distribution seen by electron tomography. This complexity is never mentioned in the current work.

Ln 727 Isn't the statement "whose plus ends were upstream from that position" unnecessary, because the way polarity is assigned to an end is by the fact that it is proximal to the pole.

Ln 283 The statement here and the graphs in Figure 6E are dependent on the model developed in the appendix, and this should be noted. As it is, they appear to come out of thin air.

Ln 735 I don't see how the kinetochore nucleation model leads the speed of KMT flux to decrease toward the spindle pole.

Ln 371 There is a persistent terminology problem in this paper. It is the difference between KMTs being nucleated at the kinetochore, vs. growing at the kinetochore by addition of tubulin to the KMT's plus-end. Can't MTs be captured by the kinetochore and then grow at the kinetochore, even if they were not nucleated there? In that scenario, I don't see how the models are so clearly discriminated as the paper suggests.

Ln 376 This line states quite clearly the enigma in this paper's data: KMTs are said to polymerize at their plus ends, and once their minus end has moved far enough poleward, it is captured by polar material. But if the KMT is continuous, how can different parts of the same MT move at different rates? Isn't it likely that the observed slowing is simply a result of the curved trajectory of most KMTs, so a constant speed along the stream lines leads to a perceived slowing of the fluorescence when viewed relative to a plane that contains the pole-to-pole axis?

Ln 383 Why do you call the observed speed an inferred speed? Is this because you are trying to relate the observations to the geometry of the K-fibers?

Ln 483 The novel observation in this paper is the slowing of the bar of fluorescence as it moves toward the pole. As suggested above, this result can be explained by spindle geometry with no reference to streamlines, nematic liquid crystals on anything but the shape of the spindle MTs as determined by ET. The apparent speed of the bar moving along the spindle axis will vary with position because most KMTs and others are in roughly elliptical arcs. The motion of a zone of activated fluorescence along those curving fibers will appear to slow, because the component of its velocity vector that is parallel to the spindle axis is less and less as the MTs get closer to the poles. I think this simple idea explains everything without any reference to liquid crystals.

Ln 457 This statement seems incorrect for several reasons. There is no evidence that non-KMTs in HeLa cells flux toward the pole in these cells. They turn over too fast for this motion to be easily seen. Thus, only KMT are being followed in the fluorescence observation, and these, by definition, have their plus ends on the kinetochores. Shorter KMTs simply have their minus end farther from the pole. These KMTs of different lengths cannot move at different speeds, because their speed of movement is defined by the rate of tubulin addition at the kinetochore.

Ln 463 The Onsager estimate for the I-N transition volume fraction (~0.04) depends on the mean aspect ratio of MTs. What aspect ratio was assumed in making this estimate?

Ln 529 There is a key point missing from this analysis of fluorescence intensity: the shape of the spindle in 3D. Since observations were made with spinning disk confocal optics, the fluorescence recorded will have been largely from a plane containing the spindle axis and perpendicular to the optic axis, including material from some half-micrometer of thickness. The line of photoactivation, on the other hand, probably includes tubulin at different heights along the optic axis. The actual method of photo-activation is not well spelled out, but from the use of the Physika photo-positioner, I infer that the 405 nm light was passed through the same objective used for imaging, forming a diffraction-limited spot. This was then moved perpendicular to the spindle axis to generate a line, perhaps with successive pulses from the laser. Although the high NA will have made the region of maximum intensity pretty short along the optic axis, the multiple sites of activation, pulsed or not, will mean that some MTs below and above the plane of focus will have been activated. Given the spindle's geometry, these MTs will curve in toward the plane of observation as one approaches the pole, so the fluorescence observed along the spindle axis will contain fluorescence from MTs that were out of the plane of view near the equator. If these regions of fluorescence move along the arcing KMTs, they will arrive at the poles later than the fluorescence that moved on straight KMTs. The same will of course be true for MTs curving within the plan of observation. This is probably a major cause of the observed smearing of the photo-activated line, and it is probably the explanation for the observed slowing of the fluorescent bar as it nears the pole. Note the very high curvature of most MTs in the vicinity of the pole. These geometrical issues are not mentioned, and they are probably the reason for the most interesting observation presented. In spindles like those formed in PtK cells, which are much flatter, so the MTs are far straighter in each half spindle, this curvature would be much less of an issue. This difference may explain why previous studies of spindle flux have not observed a pole-proximal slowing of flux during metaphase, even though flux rates were shown to vary with time in mitosis.

Ln 529 It is not obvious why the photoconverting line is moved at this rate. Do you mean that the spot of the photo-activating laser was moved to make a line? Please clarify.

Ln 546 The number of pixels width is represented by [check number]

Ln 548 By subtracting the side whose fluorescence was enhanced by photoconversion from the opposite side of the spindle, it sounds as if all your values would be negative. Have you stated this backwards, or did I misunderstand?

Ln 563 This section raises the specifics of problems mentioned in Results above. The NA of the objective used here is high, meaning that its depth of field is comparatively narrow. The issue of how birefringent material that is at the edge of, or outside, the depth of field contributes to the retardation recorded is a difficult problem not addressed by Oldenberg and not considered here. The fact that in a spindle with the shape of a human metaphase, many MTs are running at high angles to the plane of focus adds more complexity. The fact that the fraction of MTs at high obliquity to the plane increases near the pole complicates the problem further. Thus, the use of polarizing optics to assess MT orientation is problematic, even with a first-class instrument, like the PolScope. MT orientations determined by electron tomography do not face these issues, albeit they have problems of their own. This means that the similarity of the false color images generated from ET and pol microscopy raise concerns about the validity of the ways in which the colors are generated and what they really mean.

Ln 588 The fact that the full 3D geometry of the spindle seen by ET was used in projection to generate the EM version of the MT orientation map, whereas there is no such 3D information in the polarization microscope images used for that map, yet the two maps look quite similar adds to my worries about the value of this presentation.

*Reviewer #3 (Recommendations for the authors):*

This is an excellent work and I only have minor suggestions to the authors that do not require additional experiments:

1) The authors are inferring about the origin of k-fibers in human cells by looking at metaphase spindles that are already at steady state. This is in my view the biggest limitation of the present work. Nevertheless, the comparisons drawn from both the electron tomography data and the investigation of spindle microtubule dynamics, supported by the biophysical modeling are powerful in supporting the proposed model of k-fiber self-assembly. Whether or not all kinetochore microtubules originate from kinetochores, remains debatable. For example, the present study does not take into account the possible contribution of Augmin-mediated microtubule-dependent amplification (see e.g. Almeida et al., BioRxiv, 2021.08.18.456780). In addition, the authors cannot formally exclude that a fraction of those 50% of kinetochore microtubules that end at spindle poles have a non-kinetochore origin, as predicted by their "hybrid" model. These limitations should be acknowledged/discussed appropriately.

2) On page 2, the authors confront the two main models of k-fiber origin: Microtubule capture vs. nucleation at kinetochores. Although they do acknowledge later on that nucleation from kinetochores would be qualitatively indistinguishable from capture of small microtubules near kinetochores, this reviewer is of the opinion that the term "nucleation" might be misleading and should be renamed in order to avoid incorrect interpretations by less familiar readers (e.g. nucleation with minus-ends anchored at kinetochores).

3) How do the authors explain the reduction of flux velocity near the poles? Do the biophysical models distil any prediction? The authors should discuss that this finding might actually explain different flux rates measured for the same human cell type by different laboratories, possibly due to where in the spindle the photoconversion line is initially positioned relative to the poles.

4) It wasn't totally clear to this reviewer how SNAP-Centrin was fluorescently labelled and why is it not depicted in the photoconversion experiments? For example, in Figure 2A, it is not clear whether the two rows correspond to different channels of the same cell? If they do, the images do not seem aligned.

[Editors' note: further revisions were suggested prior to acceptance, as described below.]

Thank you for resubmitting your work entitled "Self-organization of kinetochore-fibers in human mitotic spindles" for further consideration by *eLife*. Your revised article has been evaluated by Anna Akhmanova (Senior Editor) and a Reviewing Editor.

The manuscript has been improved but there are some remaining issues that need to be addressed, as outlined below:

1. Please clarify the conceptual language around the model where kMTs have slower motions at the poles. This part is challenging to understand and interpret in the current version of the manuscript. Is the following interpretation reasonable to explain the observations? KMTs by definition have one end on a kinetochore. Near the kinetochore itself, many of the KMTs are short. These grow comparatively fast, so the flux of tubulin in their walls is also fast. Longer KMTs reach closer to the poles. there are fewer of them, but for reasons not identified, they present more drag on the MT growth process, slowing down their polymerization. Thus, their tubulin is fluxing more slowly. Stated more simply, as KMTs elongate, the increase in drag on their poleward motion slows flux. Drag could be one explanation for the slower growth, but there could also be others. If this "slower growth" interpretation is correct, it would imply heterogeneity among kMTs. In this case, a long microtubule that fluxes from KT to pole slower than a shorter kMT should still contribute with slow flux when a mark is photoactivated closer to the chromosomes (where flux is faster?). Is this what was meant? This needs to be clarified in the manuscript.

Further, this interpretation makes a prediction: near the kinetochores, where there are both long and short MTs, a bar of induced fluorescence should spread more quickly than an identical bar of fluorescence generated near the pole. This prediction is based on the ET data in the companion paper that near the kinetochore there are both long and short KMTs, whereas near the pole, only long KMTs are found. The heterogeneity in KMT length near the kinetochores should lead to a faster spread of induced fluorescence. This would be an easy measurement to make, probably on existing data, and the results might either support or refute the author's interpretation. Although inclusion of such data is not essential, it would strengthen the manuscript.

2. Could the authors clarify whether they collected single-plane images or of z-stacks that were subsequently projected in 2D for the above analysis and explain how they were processed.

*Reviewer #1 (Recommendations for the authors):*

The authors have made some important changes to their manuscript, and the result is a much-improved paper. Experimentally, the use of two-photon activation for the study of microtubule movements in a metaphase spindle is a significant improvement. Although the results are similar to those previously obtained, the narrower field depth of photoactivation eliminates one plausible alternative explanation for the observations and raises confidence in the interpretations given. Well done.

The new material that explains more fully the relationship between 3D and 2D calculations for the active liquid crystal theory is again a substantive improvement in the paper, but I note that such material was more prevalent in the rebuttal than in the paper itself. The difficulties with the previous version were not due simply to succinctness. There were confusions and even inaccuracies, but I think the new version is acceptable.

The response to criticisms of the previous paper's description of KMT turnover is fine. The new text clarifies what the authors are talking about, again a substantive improvement. This reviewer still has trouble, though, with the descriptions of the changes in speed of KMT minus ends as a function of position in the spindle. The explanation offered in the rebuttal and in the discussion, that longer KMTs add subunits more slowly at the kinetochore, is plausible and helps the reader understand immediately what is probably going on to account for the observations. I encourage the authors to work words of that kind into their paper somewhere near the first statement of this observation, because otherwise, the result appears to be hard to understand. They talk about KMTs slowing as they approach the pole, but isn't clearer to say that long KMTs, whose minus ends are closer to the pole, add tubulin at their kinetochore-associated end more slowly than shorter KMTs. All in all, the paper is now a fine piece of work and a significant contribution to the field.

Specific comments

Abstract: An issue raised in initial review is that the words used to describe the speed of KMT movement toward the pole suggest that different parts of one MT move at different rates, which the authors clearly do not mean. Their sentence, "in metaphase, KMTs grow away from the kinetochores along.

well-defined trajectories, continually decreasing in speed as they approach the poles" perpetuates this problem. By definition, a KMT has its plus end associated with a kinetochore. If the minus ends of these MTs move toward the pole at different rates, then the rates of growth at the kinetochore must be inversely related to KMT length. Some statement to that effect would clarify the authors meaning. Moreover, the statement "KMTs predominately nucleate de novo at kinetochores" lacks the specification, made clear in the rebuttal document, that the authors mean during metaphase. Please be explicit about this. Otherwise, the statement is incorrect.

Introduction:

Ln 103 Again, specify metaphase.

Results:

Ln 125, the same.

Ln 241 "Due to our use of a low numerical aperture 242 condenser (NA=0.85)…" This sentence is an over-simplification. With this condenser NA, the cone of illumination still has an angle of about 50o. This is hardly a column through the spindle, as described in the rebuttal. The measurements made with this pair of objective and condenser have a depth of field of ~1 µm, so a lot of out-of-focus retardation is contributing to the brightness measured at each pixel. Given the broad cone of illumination, retardance measured at any point in the image plane includes retardance from parts of the spindle volume away from the optic axis at that point. The apparently impressive correspondence between Oldenbourg's single MT measurement and the prediction for average spindle brightness is consistent with my view that the pole-scope observations average over a range of X and Y as the retardation from different levels of Z are added to each position in the image plane. Thus, the observations lack the precision of a "projection", but the point is not worth fussing over. I agree that these polarization measurements have value, but I doubt that they have the precision claimed.

Ln 265 The second reference to Fürthauer et al. lacks a journal.

Ln 273 You persist in equating the Laplacian to 0 without mentioned that you are using a 2D approximation. I am puzzled by your resistance to accuracy.

Ln 476 Again, add metaphase.

Ln 480 and following. This statement is very nice and clear and fully understandable, when accompanied by the inference that the rate of polymerization of tubulin at the kinetochore is inversely related to the length of the MT. Why not add that here for completeness?

Ln 532 The word "speed" here is ambiguous, because you are talking both about MT turnover and speed of MT motions. Please clarify.

Ln 545 Again, by stating that MT minus ends slow down as they approach the pole without any mention of what's going on at the plus end of those same MTs, you suggest that the minus ends are moving at a rate that is different from the rest of the MT, which you don't intend.

Ln 938 Need a space after 0

*Reviewer #3 (Recommendations for the authors):*

The authors did an excellent job clarifying the required revision points. However, this reviewer is still not fully satisfied with the explanation and data about flux decrease near the poles. It is not so much the way the photoactivation mark was created (1- vs 2-photon, 3D vs 2D) but the way the movement of the mark was imaged and tracked. There is no information in the methods on whether the authors collected single-plane images or a collection of z-stacks that were subsequently projected in 2D. Either way, higher microtubule curvature near the poles would be masked in any 2D analysis, creating the illusion of flux slowing down near the poles. If one assumes that the photoactivation mark created on spindle microtubules near the equator does slow down when it reaches the poles, it implies discontinuity of the photoactivated microtubules, which is hard to conceive given that microtubules are transported as continuous units, even if only a fraction of the microtubule minus ends reach the poles. Could the authors prove me wrong on this, or am I missing something very obvious here? Recognizing potential technical limitations/caveats and making this point absolutely crystal clear in a simple language that a common cell biologist without extensive notions of physics can understand would certainly help the field and many of the potential readers of this article. As a minor note, the cited Almeida et al., preprint has now been peer-reviewed and published in Cell Reports. The authors may kindly update the citation of this work.

---

## [Author Response]

Essential Revisions (for the authors):1) The authors need to consider in their analyses the almost spherical shape of the HeLa spindle and the impact of the resulting MT curvatures in 3D on the way photoactivated fluorescence will appear to change with time as KMTs flux when observed in a plane that contains the spindle axis. Clarity is required on whether the spindle is considered a 2 or a 3 dimensional object. SDCF imaging is essentially planar; photoactivation of fluorescence is not. The slowing of flux near the poles might be a result of this feature of experimentation and the shape of a human spindle: the apparent slowing may simply be a result of observing the fluorescence of a 3D object in 2D. Similar 2D/3D issues emerge in the projections of ET data to make to 2D color map that is compared with Pol optics and theory. Though better explanations are required to understand how this was done, the impression is that the theory is in 2D, (Laplassian not = 0 in 3D).

We thank the reviewers for highlighting the confusion over a 2D vs 3D treatment of the spindle and apologize that our original presentation was too terse. We have expanded our explanation in the text to clarify that we are treating the spindle as a 3D object using the full 3D information available from the electron tomography (ET) data.

The reviewers raise the concern that the slowing down of the photoactivated tubulin near the poles might be an artifact of the spindle shape and the contribution of out of focus photoactivation in 3D. We previously found from our flux analysis and photoactivation simulations that the curvature of the KMTs alone could not explain the distribution of KMT minus ends from the ET reconstructions and the speed of a line of photoactivated tubulin in the spindle, but we neglected to include these results in the original manuscript. Still, to more rigorously check that the observed slowdown is not an imaging artifact, we repeated our original photoactivation experiments using a two-photon photoactivation system. The two-photon system produces a photoactivated line with a narrow one-micron width in the z-direction (newly included Figure 2S1). The contribution of out of focus photoactivated tubulin entering the imaging plane is therefore minimal when using the two-photon photoactivation system. We have explained this in the text (pg. 4):

“The two-photon photoactivation produced a narrow line in the z-direction perpendicular to the imaging plane (σ=1.0±0.1µm), so the contribution of out of focus photoactivated tubulin entering the imaging plane is minimal (Figure 2S1).”

We obtained very similar KMT tubulin speeds using the two-photon photoactivation system as we had in the original one-photon photoactivation experiments, indicating that the observed slowdown of tubulin near the poles was not an artifact of out of focus tubulin. The revised manuscript uses the new two-photon photoactivation results throughout (see updated Figures 2 and 3) and includes one-photon photoactivation as only a supplemental figure (Figure 3S1). We have noted this in the text (pg. 5):

“The speeds we observed with the two-photon photoactivation were very similar to the speeds we observed with a traditional one-photon photoactivation system (Figure 3S1). The measured KMT and non-KMT lifetimes were indistinguishable between the one- and two-photon activation systems (KMT Lifetime: One-Photon: 2.7±0.2min, Two-Photon: 2.8±0.2min, p=0.71; Non-KMT Lifetime: One-Photon: 0.29±0.02min, Two-Photon: 0.26±0.01min, p=0.10).”

To determine if the slowdown was the result of increased microtubule curvature near the poles in 2D, we analyzed the motion of a thin section of the photoactivated line near the center of the spindle where the microtubules are all very straight. We found that the speed of this narrow central region of the line was very similar to the speed of the entire line binned together, indicating that the slowdown was not solely the result of increased microtubule curvature near the poles (Figure 3S2). To determine whether a model with constant minus end speed could reproduce the observed slowdown, we performed an additional simulation with constant KMT minus end speed (Figure 7S5). We found that the constant speed model did not reproduce the photoactivation data, arguing that the observed slowdown is not a consequence of the spindle geometry. Thus, the observed slowdown near the poles must be the result of the tubulin in individual KMTs slowing down as KMT minus ends reach the poles. We have added a description of these experiments to the result section of the manuscript on pg. 5:

“To test if the observed tubulin slowdown near the poles was a consequence of the increased curvature of microtubules near the pole, we analyzed the motion of a thinner 2μm section of the photoactivation line near the spindle axis where the microtubules are relatively straight (Figure 3S2). We found that the motion of this central portion of the line moved at very similar speeds to the entire line binned together, suggesting that the observed slowdown was a not a consequence of increased curvature near the poles.”

And (pg. 10):

“We compared the predictions from a model where KMT minus ends move at constant velocity in the spindle bulk and found that this model did not agree well with the speeds from the photoactivation experiment (Figure 7S5).This suggests that the KMT tubulin slowdown near the pole is not merely the result of increased MT curvature.“

The reviewers raise the additional concern that the comparison between the radial projection of the ET data, a 2D theory, and the PolScope imaging is inappropriate because it compares 2D to 3D data, and that the Laplacian of the angle of the director is not equal to zero in 3D. We again apologize for the terse presentation in our original manuscript. For objects shaped like spindles, the solution of the 2D theory is an excellent approximation for both a 2D slice of the full 3D solution and a 2D projection of the full 3D solution: to demonstrate this point, we have added a comparison between the 2D approximate solution, a central slice of the 3D solution and the 2D projection of the 3D solution (Figure 5S3). We used the 2D Laplacian approximation in our original manuscript for simplicity (as had previously been done in Brugues and Needleman, 2014). We appreciate that this was confusing, so we have updated the original figures 4, 4S2 and 3 (now 4S1 and 2), 5 and 5S1 (now 5S4) to show the projection of the solution of the full 3D theory as would be observed in PolScope microscopy (see below). In the confocal imaging simulations, we used a central slice of the full 3D theory to most accurately represent the orientations of the microtubules imaged during the photoactivation experiment (pg. 9):

“We used a 2D simulated spindle to model KMT motion. Since the imaging depth from the photoactivation experiments was narrow (~1um), we used a central slice of the full 3D director field predicted by the active liquid crystal theory as a model for the KMT orientations. From these orientations, we generated a set of streamlines spaced 0.5 μm apart at the center of the spindle (Figure 7A).

We thank the reviewers for their careful reading and hope that this modification makes the text easier to understand.

2) The authors also need to give proper consideration to the impact of this same spherical geometry on the distribution of birefringence observed with a PolScope, which uses a high numerical aperture objective.

We again thank the reviewers for their close reading and for highlighting the confusion with the 2D vs 3D treatment of the spindle. As we explained in the response to point 1, for an ellipsoidal tactoidal geometry like the spindle, the predictions from a 2D theory, a central slice of a 3D theory and projecting the 3D theory are extremely similar, so we originally used the 2D approximate solution to predict the microtubule orientation from the active liquid crystal theory. To avoid any confusion regarding the treatment of the spindle geometry, we have modified Figures 4, 4S2/3, 5 and 5S4 so that projection of the 3D theory into 2D is always shown (pg 5):

“We projected the MTs into a 2D XY plane and calculated the average orientation, ⟨θ⟩ where tan⁡θ=nynx, in the spindle for both non-KMTs (Figure 4B) and KMTs (Figure 4C). For each spindle, we averaged the spindle over twenty π10 radian rotations to produce a uniform projection.”

The reviewers note that the PolScope images were acquired using a high numerical aperture objective, which would seem to imply that the imaging on the PolScope is well resolved in a tight imaging plane. We apologize for the confusing language surrounding this description. We used a high numerical aperture objective with a low numerical aperture condenser (NA=0.85) which produces light that remains columnated over the entire z-span of the spindle. Since the PolScope retardance and orientation measurements depend on the sum of the interaction of the light rays with the birefringent microtubules as they pass through the sample from the condenser to the objective, the final PolScope image is essentially an average of all the spindle z-planes, weighted by the density of microtubules at that position in each z-plane. To verify this, we computed the retardance area observed in the PolScope images of each of the HeLa cells (new Figure 5S1). We found a mean retardance area of 12,100 ± 900nm^2^ in the PolScope images. Given that the retardance area of a single microtubule is ~7.5 nm^2^ (Oldenbourg et al. 1998), the microtubule density measured in the ET reconstructions would predict a retardance area of 11,400 ± 800nm^2^ if summed over the entire z-direction of the spindle. The close agreement between the calculated retardance area from ET reconstructions with the retardance area measured from PolScope is consistent with the PolScope image being an average over the entire spindle depth in the z-direction, as is expect from the optics used in our experiments. This argues that it is most appropriate to compare the PolScope images with the 3D model projected into 2D, as we show in the updated Figure 5. We thank the reviewers for pointing out this confusion and hope that our updated figures will be clearer to readers.

3) The model for spindle structure based on "active liquid crystal theory", similar to work from this same group on spindles formed in frog egg extract has two major issues. (A) It does not seem to have the interpretive power attributed to it. (B) A more accessible description of the active liquid crystal theory work in Figures 4-5 is required. The authors summarize this in lines 220-223, and the logic there is not clear. This seems like an almost tautological conclusion the way it is presented.

We apologize for the confusing description of the active liquid crystal model. The interpretive power of the model comes from its ability to predict the orientation of microtubules in the spindle. If the system was not well described by a continuum active liquid crystal, then the model predictions would not agree with the microtubule orientation seen in ET and the PolScope data. To make this point more clearly, and to make the theory more accessible, we have edited the description of the theory to read (pg. 5-7):

“We next calculated the orientation field of MTs in Hela spindles by averaging together data from both non-KMTs and KMTs from all three EM reconstructions by rescaling each spindle to have the same pole-pole distance and radial width (Figure 5A). We sought to test if the resulting orientation field was representative by obtaining data on additional Hela spindles. Performing significantly more large-scale EM reconstructions is prohibitively time consuming, so we turned to an alternative technique: the LC-Polscope, a form of polarized light microscopy that can quantitively measure the optical slow axis (i.e. the average MT orientation) and the optical retardance (which is related to the integrated microtubule density over the image depth) with optical resolution (Oldenbourg et al. 1998). Due to our use of a low numerical aperture condenser (NA=0.85), the Polscope measurements corresponded to projections over the z-depth of the spindle. Consistent with this expectation, the measured retardance from the PolScope agrees with the predicted retardance from projecting the entire z-depth of the EM reconstructions onto one plane (Figure 5S1). We next averaged together live-cell LC-Polscope data from eleven Hela spindles and obtained an orientational field (Figure 5B) that looked remarkably similar to the projected orientations measured by EM (compare Figure 5A and 5B).

[…]

The observation that the active liquid crystal theory can accurately account for the orientation of MTs throughout Hela spindles (a prediction which, as noted above, does not depend on any parameters of the theory) provides support for the utility of the theory and the validity of its underlying assumptions. This, in turn, suggests that the orientation of MTs in Hela spindles are determined by their mutual, local interactions, which cause them to tend to grow and move along the direction set by the orientation field. The predicted trajectories of MT growth in the theory are streamlines that lie tangent to the MT orientation field.”

4) The authors need to clear up confusion in terminology and apparently in thinking, about the difference between MTs whose growth is initiated at the kinetochore vs. those whose dynamics are affected by tubulin addition at the kinetochore. This confusion is compounded by a presentation that conflates what goes on during spindle formation and what happens in a metaphase spindle, where one is looking at the dynamics and turnover of an already formed K-fiber, vs. the formation of such a fiber as the spindle develops. Ibn other words, the manuscript analyses KMT turnover in metaphase cells and uses this to address the question of where KMTs come from. However, the source of KMTs may change with time in mitosis, so the origin of KMTs during prometaphase may be different from that in metaphase. This is important because it is the early KMTs that initiate chromosome attachment and guide most of metakinesis. However,the observations in this manuscript pertain to KMT turnover, not initial KMT formation.

We thank the reviewers for noting their misunderstanding surrounding our discussion of KMT recruitment. The reviewers are mistaken: we do not discuss the behaviors of KMTs during prometaphase. We suspect that this confusion arises from our discussion of how the KMTs are recruited, and we apologize for the terse language in this section of the manuscript. We are referring to KMT recruitment in metaphase, not prometaphase. Progression from early prometaphase to metaphase takes roughly 20 minutes, and metaphase lasts 5-10 minutes in human tissue culture cells. Since KMT numbers remain roughly constant over much of this time (McEwen et al. 1997, McEwen et al. 1998), and the KMTs turnover roughly every 2.5 minutes in metaphase, new KMTs must continually be recruited to the kinetochore during metaphase. The early KMTs that initiate chromosome attachment in prometaphase will have long since detached from the kinetochore ~25 minutes later in metaphase. Our data argues that the metaphase-recruited KMTs must be nucleated at, or very near, the kinetochore rather than nucleated elsewhere in the spindle and then captured by the kinetochore. We make this determination from the results of a steady-steady analysis of the position of KMT minus ends in metaphase. In this analysis, we balance the incoming flux from KMTs that are newly recruited during metaphase with KMT motion and KMT detachment. The KMT lifecycle in other stages of mitosis (i.e. prometaphase, anaphase) could be quite different from metaphase and elucidating the behavior of prometaphase and anaphase KMTs will require further experiments.

To ensure that this point is made clearly in the text, we have edited our discussion of KMT recruitment in the paper introduction (pg. 2):

“The lifecycle of a metaphase KMT consists of its recruitment to the kinetochore, its subsequent motion, polymerization and depolymerization, and its eventual detachment from the kinetochore. In metaphase, KMTs turnover with a half-life of ~2.5 min, so the KMTs that originally attached during initial spindle assembly in early prometaphase have long since detached from the kinetochore and been replaced by freshly recruited KMTs over the ~25 minutes from nuclear envelope breakdown to anaphase. The number of KMTs remains relatively constant over the course of mitosis (McKewen et al.1997, McKewen et al. 1998), so new KMTs must be continually recruited to kinetochores throughout metaphase to replace the detaching KMTs. Prior experiments have established that kinetochores are capable of both nucleating microtubules *de-novo* and capturing exiting non-KMT microtubules (Telzer et al. 1975, Mitchinson and Kirschner 1985a, Mitchinson and Kirschner 1985b, Huitorel and Kirschner 1988, Heald and Khodjakov 2015, LaFountain and Oldenborug 2014, Petry 2016, Sikirzhyski et al. 2018, David et al. 2019, Renda and Khodjakov 2021). Either of these mechanisms could potentially be responsible for the KMT recruitment to kinetochores during metaphase.”

In the Results section surrounding the discussion of the steady-state minus end flux analysis (Pg. 8):

“This prediction requires specifying the rate of metaphase KMT detachment, which, based on our photoconversion measurements, we take to be *r* = 0.4 min^-1^ from the mean observed KMT lifetime in the spindle bulk (Figure 3). Presumably, the KMT minus end distribution n(s), the KMT minus end velocity v(s), the detachment rate *r* and the KMT recruitment rate j(s) all vary over the course of mitosis. Here, we only focus on metaphase when the spindle is in (approximate) steady-state.

We consider two possible models of recruitment of new KMTs to the kinetochore during metaphase: either that KMTs are nucleated at kinetochores or that KMTs arise from non-KMTs whose plus ends are captured by kinetochores. We base these two possibilities on prior experiments indicating that kinetochores are capable of both nucleating KMTs de-novo (Witt et al. 1980, Mitchinson and Kirschner 1985a,, Khodjakov et al. 2000, Khodjakov et al. 2003, Maiatio et al. 2004, Sikirzhytski et al. 2018) and capturing existing non-KMTs (Mitchinson and Kirschner 1984, 1985b, 1986, Huitorel and Kirschner 1988, Rider and Alexander 1990, Hayden et al. 1990, Kamasaki et al. 2013, David et al. 2019). It has not been clear which of these possibilities is responsible for recruiting metaphase KMTs to kinetochores.”

And the paper’s discussion (pg. 13):

“Since the observed KMT turnover (~2.5 minutes) is rapid and the number of KMTs remains roughly constant throughout the course of mitosis (~30 min) (McKewen et al.1997, McKewen et al. 1998), new KMTs must be recruited to kinetochores during metaphase. Prior work has shown that kinetochores are capable of both nucleating new KMTs de-novo and capturing existing KMTs from solution; however, it has been unclear which of these possible mechanisms is responsible for recruiting new KMTs to the kinetochore during metaphase. (Tezlzer et al. 1975, Mitchinson and Kirschner 1985a, Mitchinson and Kirschner 1985b, Huitorel and Kirschner 1988, Heald and Khodjakov 2015, LaFountain and Oldenborug 2014, Petry 2016, Sikirzhyski et al. 2018, David et al. 2019, Renda and Khodjakov 2021). We show that a model where metaphase KMTs nucleate at the kinetochore is consistent with the KMT ultrastructure observed in the tomography reconstructions and the tubulin dynamics observed in the photoconversion experiments.”

5) The authors show regional variation of flux in human spindles, whereby KMTs slow down near poles. The authors should discuss this finding in the context of existing models showing slowed speeds near poles, such as in Burbank et al. 2007 slide-and-cluster model. Dscussion on possible mechanisms for slower flux closer to poles in human spindles. should also be added.

We have added a discussion of the mechanism of KMT elongation to the Discussion section (pg 13).

“In *Xenopus* egg extract spindles, non-KMTs move more slowly near the spindle poles than near the spindle equator. Inhibiting dynein causes the non-KMT speed to become spatially uniform, suggesting that the non-KMT slowdown is the result of dynein-mediated microtubule clustering (Burbank 2007, Yang 2008). A similar mechanism might explain the slowdown of KMTs described in this work. Further experiments in dynein-inhibited HeLa spindles will be necessary to test this possibility.”

Reviewer #1 (Recommendations for the authors):Their model for spindle structure is based on the condensed matter physics of an "active liquid crystal". This model was initially developed in previous papers from this same group, but here it is revised to correspond to the structure of a human spindle and to interpret the structural and kinetic data mentioned above. In this paper the model is poorly described, so this part of the paper is hard to read with confidence. However, several aspects of the model appear to be misleading or incorrect. For example, the theory described resembles the familiar equilibrium Frank orientational elastic theory of nematic liquid crystals with prescribed boundary conditions (tangential anchoring at the spindle boundary) and with two point-defects, which the authors associate with the spindle poles. Although details are lacking, this model appears to be the same as that developed in Brugués and Needleman 2014, namely a one-constant approximation with equal splay and bend elastic constants, with the additional assumptions that the spindle is twist-free and axially symmetric. The authors claim that within this approximation the MT orientation field θ(r) in the spindle satisfies the Laplace equation in this case (∇^2 θ=0), but this is not correct. In cylindrical coordinates, the equilibrium MT orientation field θ(ρ,z) in the one-constant approximation is the solution of∇^2 θ(ρ,z)=1/ρ^2 sinθcosθsee, for example, M. Press and A. Arrott, "Elastic energies and director fields in liquid crystal droplets, I. Cylindrical symmetry", Journal de Physique Colloques 36, (C1), C1-177 – C1-184 (1975). Apparently, both here and in the earlier paper by Brugués and Needleman, the spindle was modeled as a two-dimensional object, in which case the director field is indeed the solution of the Laplace equation in the one-constant approximation. Two significant limitations of the current paper are: (1) that the discussion switches back and forth from 3D to 2D without adequate signals to the reader and

We are, of course, aware that the 2D solution and the 3D solution are not mathematically identical. As we discussed in our response to the essential revisions, it is well known that for shapes like the spindle, the 2D solution is an excellent approximation to both a central slice and the projection of the full 3D solution (as demonstrated in updated Figure 5S3). The 2D approximation was used in our previous publication and in our previous version of the manuscript. However, we appreciate that our terse description of this is confusing, particularly for readers who lack the relevant background. Thus, we have updated the figures and analysis to use the projection of the full 3D solution.

2) that details of the theory are not in the manuscript, for example details about how the boundary conditions were imposed.

We thank the reviewer for highlighting that the details of the implantation of the boundary conditions were not made sufficiently clear. We have added these details to the methods section on lines pg. 15:

“We imposed the boundary conditions by setting the microtubule orientation at the elliptical boundary to be tangent to the ellipse and radially outward within the aster radius at every finite difference method iteration.”

It is noteworthy that the ambiguity about two- or three-dimensional treatments is a persistent problem in several aspects of the paper. The tomographic reconstructions, which are being related to the data from light microscopy, are unquestionably 3D. The polarization optical data are apparently 2D, because the PoleScope used has a high numerical aperture objective, and there is no mention of collecting data from serial optical sections.

The reviewer is mistaken regarding the optics of the PolScope. Though we used a high NA objective, the NA of the condenser we used was quite low (NA=0.85), so the PolScope data is expected to be a sum of the projection of the microtubule orientation in 2D over every z-plane in the spindle. Consistent with this expectation, when we compared the observed retardance form the PolScope images with the predicted retardance from the EM reconstruction (new Figure 5S1), we found that they agreed well if the PolScope image was a sum over the entire spindle z-direction. We apologize for the confusion in the manuscript and have added clarifying language surrounding our description of the PolScope imaging.

Likewise, the images for analysis of fluorescence redistribution after photo-activation are 2D, since they were collected with a spinning disk confocal microscope (SDCM). In the paper's section where the tomographic data are being projected, it appears (the writing is unclear) that all spindle MTs are projected onto a plane and this projection is then compared with the images from polarization microscopy. If this is in fact what was done, the two structures being compared are not the same thing, and the apparent similarity in distribution of angle-revealing colors is a coincidence.Moreover, the activation of fluorescence was done by moving the photoconverting laser beam across the spindle, and this beam certainly activates fluorescence above and below the plane of observation. (The beam is presumably a diffraction limited spot, whose length along the optic axis is greater than its width in the plane of view. Moreover, although photo-converting intensity drops with the square of distance from the image plane, the time of irradiation by the moving beam increases linearly, so fluorescence at significant distances from the plane of focus will experience photo-conversion.) This leads to a key problem in data interpretation. The slowing of MT migration with proximity to the pole probably derives simply from the large number of MTs that are oblique to the optical section observed in the SDCM, particularly in the vicinity of the pole. From this point of view, KMTs flux away from kinetochores at a constant rate, but the apparent rate of KMT motion along the spindle axis changes as one nears the pole, simply because in that region most MTs are at high angles to that axis.

As discussed in our response to the essential revisions, we performed new experiments with two-photon photoactivation to address potential artifacts due to the geometry of the spindle. The two-photon photoactivation has a narrow width in the z-direction (roughly 1 micron; Figure 2S1), so the contribution of out of focus photoactivated tubulin should be negligible. The new two-photon photoactivation results recapitulate our previous one-photon activation results. Furthermore, we analyzed the motion of a thin section of the two-photon photoactivated line near the center of the spindle where the microtubules are all very straight and found that the speed of this narrow central region of the line was very similar to the speed of the entire line binned together, indicating that the slowdown is not primarily the result of increased microtubule curvature near the poles (Figure 3S2). In addition, simulation of the expected results of photoactivation experiments with constant KMT minus end speed are inconsistent with our data (Figure 7S5). Taken together, these results strongly argue that the observed slowdown near the poles is not a consequence of the spindle geometry, rather tubulin in individual KMTs slowing down as KMT minus ends reach the poles

This interpretation has the value that different parts of a given MT all move through space at the same rate, not different rates, as implied by the language of paper.

The reviewer is mistaken in claiming that the tubulin moves at different speeds within the same KMT in our model. In each k-Fiber, there are many KMTs with minus ends scattered between the kinetochore and the pole. In our model, each of these individual KMT minus ends move at different rates, meaning that different KMTs within the same k-Fiber move/elongate at different speeds. The speed of tubulin motion along an individual KMT is the same at any instant in time, though the speed of the KMT minus end slows down as it approaches the pole.

Another problem with the paper is imprecise and confusing use of language. For example, the name "theta" is given both to an axis and an angle; no distinction is made between them when the word is used in the text, making for very hard reading.

Since we now display the projection of the spindle in our figures, we have removed any mention of the theta axis which should resolve the highlighted confusion.

The word "initiation" is used to describe KMT behavior in the vicinity of the kinetochore, when really elongation (or polymerization) is what the data report. The paper cites previous work that examined the issue of KMTs arising from capture of MTs initiated at the pole or initiation of MTs at the kinetochore, but these papers were looking at KMT formation during spindle development, not at metaphase.

We cite this literature to demonstrate that there is evidence of KMTs being both nucleated at kinetochores and captured from the spindle. It has not previously been well understood which of these mechanisms is responsible for the recruitment of metaphase KMTs. Given the rapid turnover of KMTs and that the number of KMTs is roughly constant, new microtubules must be recruited to the kinetochore throughout metaphase. This “initiation” of new microtubules is distinct from the elongation/polymerization of KMTs that the reviewer describes.

Flux of KMTs is a result of kinetochore-proximal tubulin addition to existing MTs and does not relate to issues of KMT initiation. This whole section of the paper is written in such a way that it is difficult to understand what the authors are trying to say.

KMT initiation and elongation certainly occur via different physical mechanisms: elongation by the addition of tubulin to the KMT plus end and, we argue, initiation via nucleation of KMTs at the kinetochore. However, the fluxes associated with each of these processes, and with KMT detachment, must balance at stead-state. Thus, while KMT initiation and elongation are mechanistically distinct processes, they are kinematically coupled. KMT initiation via non-KMT capture would add KMT minus ends at various positions throughout the spindle, while KMT elongation moves the minus ends towards the pole. These fluxes, along with a detachment flux, must cancel each other out at steady state. We can therefore infer a unique minus end speed/KMT elongation rate in the spindle given a model of the flux from KMT initiation.

Although the model, the data from electron tomography, and structural information from high quality polarization microscopy all appear to agree well in their descriptions of MT orientations, limitations in the paper's consideration of the full three-dimensional structure of the spindle and issues of depth of field cloud the interpretations given. The possible role of the spindle's 3D shape in offering a simple explanation for the slowing of fluorescence as it approaches the pole is not discussed, so the more complex model that is discussed does not have the intellectual force one might hope for. In sum, this paper does not contribute very much to our understanding of metaphase structure or spindle MT dynamics.

As discussed above, we believe that our terse explanations likely led to the reviewer’s confusion regarding numerous key points. We hope that our improved presentation and new result make our work easier to understand. We thank the reviewer for their time and effort, which we believe has resulted in our manuscript being much improved.

Detailed commentsLn 18 The word "many" is used here and at several other places in the paper to describe the number of KMTs that do not extend all the way to the metaphase spindle pole. The numbers, both in this paper and in Kiewisz et al. show that half go that far, half do not. Since you have the data, why not state it?

We have edited the sentence to state that roughly half of the KMTs reach the pole.

Ln 56 The most important paper about initial capture of pole-initiated MTs by prometaphase kinetochores is work from the Rieder lab, which should be cited.

We have added the Rieder citation (Rieder and Alexander 1990).

Ln 62 This long list of observations includes many that do not pertain to the statement made. However, the initial observations that support the statement (Data from Mitchison's observations with injected biotinylated tubulin, then with photo-activatable tubulin) are not mentioned.

We have added the Mitchinson citation (Mitchinson and Kirchner 1985b; Mitchinson 1989).

Ln 85 suggest adding "here" at sentence beginning to mark the transition from discussion of other people's work to the description of this paper.

Done.

Ln 93 Nucleate seems to be the wrong word. This is a big issue as the paper goes forward. To me, and I think most people in the field, the initiation of a MT is the event that starts its growth, whether this involves a seed, like the γ tubulin complex, or simply the coming together of enough tubulin oligomers to get a MT started. This is distinct from MT growth (elongation by tubulin polymerization at one end or the other). KMT turnover could be accomplished by dissociation of KMT plus ends and then true MT initiation at the kinetochores or by the mechanism sketched here (spontaneous initiation near the kinetochore, followed by kinetochore binding of the MT plus end) or by kinetochore binding of the plus end of a fully formed non-KMT initiated elsewhere. These distinctions are not clearly made in this paper, and the relevance of MT initiation to KMT flux is not spelled out.

We hope that our improved presentation has helped with the reviewer’s confusion: we are clearly distinguishing KMT growth from “initiation”. We agree that these are mechanistically distinct processes. However, at (approximate) steady-state there must be a kinematic balance between KMT growth, initiation, and detachment. Thus, even though these are three mechanistically distinct processes, knowing two (i.e. initiation and detachment) allows the third to be inferred (i.e. growth).

Ln 624 Legend to Figure 1: the minus ends are not "shown" in black, they are marked with a black circle.

Corrected.

Also, the term Pole is used here without specific definition. For these measurements, I presume you are using the middle of the mother centriole, or perhaps the center of a Gaussian spot from a fluorescent label, but you should be specific.

We mean the middle of the mother centriole and have clarified in the text (pg 3).

“We defined the location of the pole as the center of the mother centriole.”

Figure 1C. In the PDF, there is marked aliasing in each curve representing the KMTs. Find a better mode of display? Also, the representation of the metaphase kinetochore in the upper right doesn't look like any HeLa metaphase kinetochore that I have seen. Note also that the fiber shown in this diagram does not show any of the decrease in diameter toward the pole that is described in Kiewisz et al. Is this one really representative? Moreover, word usage in the insert is very cryptic. I presume that the probability you are talking about is that of finding a KMT plus end at that distance from the pole whose minus end lies within 1 μm of the Pole (to be defined). Please clarify. In D, the term, "captured from spindle bulk" again uses undefined terms. I expect you mean the capture of a non-KMT, turning it into a KMT. Please clarify.

We believe that a red dot in the upper right is a reasonable abstraction for the kinetochore, particularly when we do not intend to show any structural or mechanistic details. The reviewer is mistaken about the probability displayed in the inset. We have clarified that the probability we refer to in the inset is the probability of a given k-Fiber having X fraction of minus ends within 1.7 µm of the pole. There is no reference in the figure to the position of the KMT plus ends. We have clarified that in the figure caption for D that we do refer to the capture of a non-KMT, turning it into a KMT.

Ln 128 Images in Figure 2. Why does the polar staining with the marker for CENP-A increase so dramatically as soon as your photo-converting irradiation is done? Has the camera gain been increased to accommodate bleaching? Note also that the "line" of photo-activated tubulin appears to spread rather markedly throughout the half-spindle within 5 sec. This is not so marked in the traces of fluorescent intensity as it is in the images. Have the display parameters for the images been altered between the 0s and 5s images? The spreading is even more marked at 25s suggesting this might be the case. If so, more realistic images are called for.

We have replaced the images in question with images from the two-photon photoactivation experiment which has a narrow point spread function in the imaging plane. We have also reset the contrast on the images so that the maximum contract is the same in every image rather than adjusted to the max of the individual images. We believe this should resolve the concerns about the spreading and display parameters of the images.

Ln 633 You have a very compact way of stating the wavelength for excitation and emission in your fluorescence observations. The first time you use it, I suggest stating explicitly what each number means, so the reader is not obliged to infer it from your words.

We have added this description to the methods section (pg. 14):

“Two fluorescence channels were acquired every 5s with either 500ms exposure, 488nm excitation, 514/30 emission for the photoactivated PA-GFP channel or 300ms exposure 647nnm excitation, 647 LP emission. The PA-GFP was photoconverted using an Insight X3 femtosecond pulsed laser tuned to 750nm (Spectra Physics) and a PI-XYZnano piezo (P-545 PInano XYZ; Physik Instrumente) to draw the photoconverted line by moving a diffraction limited spot across the spindle. The line was moved at a speed of 5um/s with a laser power of 3mW (measured at the objective).”

Ln 635 You refer to the light that stimulates fluorescence as 40 nm. Typo?

We thank the reviewer for their careful attention to detail. Yes, this is a typo. 405nm light was used. We have corrected the text.

Ln 636 You say, "Line profile pulled from the dotted box shown in B." Are the graphs based on a scan simply down the line shown, or are you using a slit oriented perpendicular to the direction of scanning (the line) whose length is the vertical dimension of the box. Please be more specific.

We averaged using a slit orientated perpendicular to the direction of scanning and have clarified in the text to state (pg. 14)

“Line profile generated by averaging the intensity in 15 pixels on either side of the spindle axis in the dotted box shown in A.”

This legend lacks periods at the ends of each unit of information.

Corrected.

Ln 171 This appears to be an intriguing observation, and to my knowledge novel. Differences in MT flux over time in mitosis are well characterized, but difference at one mitotic stage, depending on position could be very interesting. After a close look at Figure 3, however, I am concerned about both the robustness of the data supporting this observation and their interpretation. Finding the position of the fluorescence peaks looks non-trivial, because the shapes of the traces change so much, both with position and time. Since this phenomenon is really intriguing, your analysis of the raw data needs better support. Some supplementary figures here, showing exactly how you determined the position of the peak and the uncertainty in those positions is needed to make the claim convincing.

As stated in the methods section, we determined the position of the center of the peak by fitting Gaussian profiles to the top of the peak. The two-photon photoactivated profiles are much thinner than the one-photon profiles and do not appear to change much over the course of imaging, which should avoid the fitting artifacts the reviewer describes.

Ln 179 Here, I'm confused. You mention projecting all MTs on "a" spindle-axis, radial-axis plane. In Figure 4s3 we are supposed to learn what you mean by this. It is easy to see that there is one spindle-axis. However, given the rough cylindrical symmetry of the spindle about this axis, one must either think of one radial axis and define a single plane that contains both these axes or consider a large number of radial axes at different orientations (all perpendicular to the spindle axis), in which case one is going to project the MTs onto many different planes. The latter would preserve the shape of curving MTs quite well, the former very badly. It seems as if you projected the 3D data from ET onto a single plane and from here on think of the spindle as a 2D object. However, the text is not sufficiently explicit about this projection to know what you have done. If your average orientations are based on projecting all MTs onto a single plane, I can't see that the parameter has any value, because you have distorted the spindle fibers so dramatically in projection. Moreover, this planar projection now contains a large number of MTs that are not imaged by a single focal plane image in the PolScope. Thus, if these colored displays of MT orientation look the same, it demonstrates that they have no value.

We thank the reviewer for noting their confusion. We hope the additional information we have added is helpful. As we discussed in the essential revisions response, for shapes like the spindle, a 2D approximation, 3D central slice, and 3D projection are all very similar (Figure 5S4). However, to avoid confusion, we have updated the relevant figures to show the projection rather than the radial average of the EM data. As we noted earlier in the response, the reviewer is mistaken about the PolScope providing single-focal plane data. Due to the low NA condensor we use, and as verified by comparison to the EM reconstructions, the PolScope data corresponds to a projection averaged over the full spindle. Thus, we have updated the manuscript with the appropriate comparison: between the PolScope data, the projection of the EM reconstruction, and the projection of the full 3D theoretical solution.

We also note that the graphs looking similar certainly does not demonstrate that they have no value, in fact agreement between two different measurement techniques (and between measurements and theory) provides strong evidence for their value

Figure 4, legend. Your text (ln 681) theta is not defined until Figure 4s3. Moreover, when you say, "averaged over all theta", where theta is the name you have given to both an axis and an angle. This needs to be clarified. Moreover, the quantities in question obviously vary with z and ρ, so this doesn't make much sense.

We have edited the figure to show a projection, which should resolve the confusion surrounding the theta, z and rho axis.

Figure 4D you give no values for the scalar nematic order parameter. Please be more specific. Most of these comments also pertain to Figure 4s2. In addition, note that the color bar that relates image hue to angle is so small it is hard to read, and it is missing from all these supplementary figures.

We have made the color bar larger and added it to the supplementary figures. Figure 4D describes the average angle of the microtubules, not the nematic order parameter. We have added the values for the mean nematic alignment in the spindle to Figure 4G.

Ln 181 There is where θ is first introduced as a MT orientation parameter, but it is never defined.

We have added a clarification that tan(θ)=ny/nx.

Ln 188 In this statement of the equation for S you use θ for the angle. Previously you have called a standard Cartesian axis theta. Suggest renaming to avoid confusion. To understand the use of this equation in the context of the spindle, we need to know how you are measuring <θ>. Is <θ> determined at each position in the spindle volume and each specific θ compared with that? The writing doesn't spell out what you have done.

We have eliminated the theta axis figure, so there should no longer be a confusion about what the angle theta refers to.

Ln 200 How does the LC-Polscope cope with birefringence that is outside the depth of field of the lens being used? I worry that the pleasing similarities between the EM data and the polescope images are not as informative as they appear, due to the ways you have projected the EM data and observed birefringence in the LM. (This point is expanded in comments on Methods.)

As discussed previously, the PolScope averages birefringence form the entire spindle, meaning that the EM projection we performed is appropriate.

Ln 704 The words used here to describe the model are not sufficient to spell out what it is. Clarify?

We have added that this refers to the active liquid crystal model discussed in the text.

Ln 204 These two paragraphs contain a brief description of the theory used to model the orientation field of MTs in Hela spindles. The same theory was applied to *Xenopus* spindles in earlier studies (Brugués and Needleman 2014, Oriola et al. 2020). As mentioned above, the development here lacks both specifics and clarity. It also reflects a general problem of this paper, the issue of whether the problem at hand is being considered in two dimensions or three. This issue must be clarified throughout the paper.

As previously discussed, the full 3D theory and the 2D approximation produce very similar results. In our revision, we have used the full 3D theory for clarity.

Ln 211 The model requires "a boundary" and two "defects". As stated, these are undefined. The relationship between the defects and the centrosomes is mentioned, but the nature of the boundary is left unspecified. As such, it sounds as if the model is set up to mimic spindle structure but without physically meaningful relations to the biological reality in the cell.

We have added a specification that we fit the edge of the spindle to an elliptical boundary and impose tangential anchoring, meaning that the local orientation of microtubules is tangential to the ellipse at the boundary. The “boundary” is simply the edge of the spindle: i.e. where the density of microtubules rapidly falls off. Defects are discontinuities in the direct field: here we use two +1 defects with radial anchoring. This is reminiscent of the structure of an aster. The model is the result of applying well established physical principles to describe biological reality in the cell.

Ln 242 This paragraph describes phenomenology of MT behavior in the language you have introduced (streamlines), but each phenomenon mentioned is equally well described by a more conventional view of the spindle, based only on MTs whose plus ends may bind and dissociate from kinetochores, whose walls may be bridged to adjacent MTs though motors and non-motor MAPs, and whose plus-end growth may promote flux of the MT's center of mass toward the spindle pole. Your streamlines are beautifully represented by the trajectories of spindles MTs in the electron tomograms, so the value of the condensed matter terminology seems limited.

The reviewer’s description of the “more conventional view of the spindle” are the same basic assumptions needed to motivate the development of an active liquid crystal theory: microtubules locally align due to interactions with motors and other microtubules. The theory describes the predicted microtubule orientation assuming tangential anchoring at an elliptical boundary and two +1 point defects (i.e. asters) located at the poles. Comparing the model predictions with the EM and PolScope data strongly suggests that the model description is appropriate since the theory predictions and the data agree quite well. In our view, “streamlines are beautifully represented by the trajectories of spindles MTs in the electron tomograms” is strong evidence favoring that the predictions made by the active liquid crystal model are correct. We also note that the theory provides quantitative predictions, not merely verbal descriptions, and the streamlines are a fundamental part of these calculations: The streamlines are derived from active liquid crystal theory and then used in the resulting flux-balance analysis and simulations.

Ln 257 I think there is a fundamental problem with this formulation. The test of the model will be the way in which it fits the distributions of spindle fluorescence as a function of time after excitation of fluorescence. The model very reasonably talks about j(s) as the rate at which non-KMTs become KMTs as a result of kinetochore binding by their plus ends. However, the fluorescence of non-KMTs changes much faster than that of KMTs, because this is the population that displays fast fluorescence loss. Therefore, when a non-KMT binds to a kinetochore, it does not bring to the KMT population a fluorescence commensurate with a normal KMT. It has already exchanged many subunits and become less fluorescent. This is, of course, the phenomenon you used to calculate exchange rates. How is this aspect of the phenomenology accounted for in the analysis?

We believe that there is a fundamental misunderstanding: Equation 1 is an equation that describes the behaviors of KMT minus ends, not (fluorescent) tubulin in a KMT. When analyzing photoactivation experiments, we assume that all of the fluorescence decay observed in the long-decay exponential is the result of KMTs that were already bound to the kinetochore. Since the KMT decay rate (2.5min) is much longer than the short non-KMT decay rate (20 sec), we presume that virtually none of the activated non-KMT tubulin would be incorporated into KMTs if they were captured by the kinetochore because only roughly 10% of the KMTs will have exchanged before the non-KMTs that were activated deplolymerize. We can therefore safely ignore any potential contribution to the slow decay rate from the non-KMT population. We do present simulations of the “spindle fluorescence as a function of time after excitation of fluorescence” which show excellent agreement with measurements (Figure 8F), but these simulations use the full inferred behavior of KMTs as described in the text (not merely Equation 1).

Ln 266 Here you state that the rate of KMC detachment based on photoconversion experiments is r = 0.4 min-1. Is it obvious how this number is obtained? I don't recall seeing this explained.

We obtained the r=0.4min^-1^ number from the average KMT lifetime that we observed in the spindle bulk. We have added this detail to the text (pg. 8):

“This prediction requires specifying the rate of metaphase KMT detachment, which, based on our photoconversion measurements, we take to be *r* = 0.4 min^-1^from the mean observed KMT lifetime in the spindle bulk (Figure 3).”

Ln 272 "If all KMTs were nucleated at kinetochores…" is a statement that reveals a complexity not considered in this development. All data come from metaphase spindles and all consideration is directed to such structures. However, every metaphase spindle has a history; it formed during prometaphase through the action of many processes and during that formation, it displayed shapes that are quite different from the spindle at metaphase. Moreover, spindles can form normal metaphase structures by a variety of pathways, depending in part on the degree of separation of the centrosomes at the time of nuclear envelope breakdown and rapid MT growth. MTs that become KMTs early may be initiated in ways that differ from MTs that become KMTs later in mitosis. Thus, the language in this paragraph must be changed to focus simply on the metaphase spindle.

Our model only describes the lifecycle of KMTs during metaphase. Based on the photoconversion experiments, a KMT remains attached to a kinetochore for, on average, 2.5 minutes. Therefore, the KMTs that initially attached to the kinetochore during early prometaphase will have long since detached ~30 minutes later when the spindles reach metaphase. We therefore presume that the KMTs that we observed during metaphase were nucleated during metaphase (or perhaps very late in prometaphase). To clarify this, we have adjusted the language to read “if all metaphase KMTs…”

Ln 280 I don't understand how the spatial distribution can be a proxy for the non-KMT nucleation rate. Do you mean the spatial variation of that rate? This, of course, assumes that minus ends of KMTs are always due to MT initiation at the kinetochore, rather than initiation in the body of the spindle. This point is a major weakness of the paper. Considerable data in the literature demonstrates that there is Augmin-dependent initiation of MTs, often in association with KMTs, and this process adds many minus ends to the distribution seen by electron tomography. This complexity is never mentioned in the current work.

The reviewer is incorrect: our argument does not assume that the minus ends of KMTs are always due to MT initiation at the kinetochore. In this work, we also explicitly consider models in which KMTs initiate in the body of the spindle as non-KMTs that become captured by the kinetochore. We take the spatial distribution of non-KMTs as a proxy for the spatial variation of the non-KMT nucleation rate. We argue that because the non-KMTs turnover very quickly (~20 seconds) compared to their velocity (at most ~1 micron/min), they do not move substantially from where they are nucleated (~0.3 microns). Therefore, the position of the non-KMT minus ends in the reconstructions is a reasonable proxy for where the non-KMTs originally nucleated. This claim is true, independent of the mechanism for how these non-KMTs were nucleated. Our analysis and assumptions are fully consistent with the known behavior of Augmin. We have added a sentence to the manuscript noting that non-KMT nucleation in the spindle bulk may be mediated by the Augmin complex (pg. 8).

“These non-KMTs could be nucleated by a variety of mechanisms in the spindle bulk, including by the Augmin complex (Goshima et al. 2008; David et al. 2019).”

Ln 727 Isn't the statement "whose plus ends were upstream from that position" unnecessary, because the way polarity is assigned to an end is by the fact that it is proximal to the pole.

We mean that when we normalize the distribution we only include microtubules whose plus ends are upstream of this position (i.e. we do not include KMTs whose associated kinetochores are downstream of the position)

Ln 283 The statement here and the graphs in Figure 6E are dependent on the model developed in the appendix, and this should be noted. As it is, they appear to come out of thin air.

We have added a note to direct readers to the supplement here.

Ln 735 I don't see how the kinetochore nucleation model leads the speed of KMT flux to decrease toward the spindle pole.

Presuming a no-flux pole boundary condition, the KMT minus end speed must be zero at the pole. If the KMTs were moving at a constant speed, then the distribution of KMT minus ends would decrease exponentially moving from the kinetochore to the pole (due to the measured constant detachment rate). Instead, we see that the KMT minus end distribution is flat in the spindle bulk and spikes near the pole. In the kinetochore nucleation model, the only source of KMTs comes from the kinetochore, so a deviation from the constant speed KMT minus end distribution must be the result of a change in the KMT minus end velocity, implying that KMT minus ends must slow down near the pole.

Ln 371 There is a persistent terminology problem in this paper. It is the difference between KMTs being nucleated at the kinetochore, vs. growing at the kinetochore by addition of tubulin to the KMT's plus-end. Can't MTs be captured by the kinetochore and then grow at the kinetochore, even if they were not nucleated there? In that scenario, I don't see how the models are so clearly discriminated as the paper suggests.

The reviewer is mistaken. We do distinguish between KMT growth and “initiation”. We also do explicitly consider a model in which MTs are “captured by the kinetochore and then grow at the kinetochore, even if they were not nucleated there”, that is our “capture from spindle model”. This model leads to predictions that are inconsistent with our data. In contrast, a model in which KMTs directly nucleate at the kinetochore, our “nucleate at kinetochore”, is consistent with our data. These two models have different j(s) flux in functions, and therefore predict different speeds of the KMT minus ends (i.e. different KMT growth speeds). These different predictions result even though KMT growth and initiation are mechanistically distinct: at steady-state the fluxes of KMT growth, initiation, and detachment must balance, so knowing two (i.e. initiation and detachment) allows the third to be inferred (i.e. growth).

Ln 376 This line states quite clearly the enigma in this paper's data: KMTs are said to polymerize at their plus ends, and once their minus end has moved far enough poleward, it is captured by polar material. But if the KMT is continuous, how can different parts of the same MT move at different rates? Isn't it likely that the observed slowing is simply a result of the curved trajectory of most KMTs, so a constant speed along the stream lines leads to a perceived slowing of the fluorescence when viewed relative to a plane that contains the pole-to-pole axis?

The reviewer is mistaken about our claim and our model. In our model, tubulin in each individual KMT moves at the same rate throughout the KMT. Tubulin in different KMTs moves at different speeds dependent on the position of the KMT minus end in the spindle. As we have previously discussed, the curved trajectories cannot account for the observed slowing down of tubulin (Figure 7S5).

Ln 383 Why do you call the observed speed an inferred speed? Is this because you are trying to relate the observations to the geometry of the K-fibers?

The inferred speed v(s) is the speed of the KMT minus ends that we calculated from the minus end flux analysis (inferred because we calculated it from the flux analysis rather than via direct observation). We have added a clarification that this speed comes from the minus end flux analysis.

Ln 483 The novel observation in this paper is the slowing of the bar of fluorescence as it moves toward the pole. As suggested above, this result can be explained by spindle geometry with no reference to streamlines, nematic liquid crystals on anything but the shape of the spindle MTs as determined by ET. The apparent speed of the bar moving along the spindle axis will vary with position because most KMTs and others are in roughly elliptical arcs. The motion of a zone of activated fluorescence along those curving fibers will appear to slow, because the component of its velocity vector that is parallel to the spindle axis is less and less as the MTs get closer to the poles. I think this simple idea explains everything without any reference to liquid crystals.

The reviewer is mistaken. As previously discussed, the curvature of the MTs is not sufficient to explain the slowdown of the bar as it nears the poles (Figure 7S5). Furthermore, the active liquid crystal theory quantitatively accounts for the orientation of MTs throughout the spindle, which holds irrespective of our photoactivation experiments.

Ln 457 This statement seems incorrect for several reasons. There is no evidence that non-KMTs in HeLa cells flux toward the pole in these cells. They turn over too fast for this motion to be easily seen. Thus, only KMT are being followed in the fluorescence observation, and these, by definition, have their plus ends on the kinetochores. Shorter KMTs simply have their minus end farther from the pole. These KMTs of different lengths cannot move at different speeds, because their speed of movement is defined by the rate of tubulin addition at the kinetochore.

The reviewer’s assertion that “KMTs of different lengths cannot move at different speeds” is mere conjecture that is not supported by any data that we are aware of. There is no *a-priori* reason that KMTs of different lengths cannot incorporate tubulin at different rates at their plus ends. Furthermore, it is unclear if the motion of KMT minus ends results from either (1) direct transport of KMT minus ends by motors towards the pole or (2) addition of tubulin at the plus end pushing KMT minus ends backwards. If motion results from the direct transport of KMT minus ends, the ability of motors to transport minus ends could certainly be impaired near the pole due to minus end clustering, drag force from longer KMTs, etc. This minus end motion would then result in the addition of tubulin at the KMT plus end. Alternatively, if motion results from the addition of tubulin at the plus end, longer KMTs could polymerize more slowly due to, for example, increased drag on the microtubule.

Ln 463 The Onsager estimate for the I-N transition volume fraction (~0.04) depends on the mean aspect ratio of MTs. What aspect ratio was assumed in making this estimate?

We used an aspect ratio of R=(1.9μm/25nm)=76 based on the mean length of non-KMTs measured in the EM reconstructions (Figure A2). We have added this clarification to the text.

Ln 529 There is a key point missing from this analysis of fluorescence intensity: the shape of the spindle in 3D. Since observations were made with spinning disk confocal optics, the fluorescence recorded will have been largely from a plane containing the spindle axis and perpendicular to the optic axis, including material from some half-micrometer of thickness. The line of photoactivation, on the other hand, probably includes tubulin at different heights along the optic axis. The actual method of photo-activation is not well spelled out, but from the use of the Physika photo-positioner, I infer that the 405 nm light was passed through the same objective used for imaging, forming a diffraction-limited spot. This was then moved perpendicular to the spindle axis to generate a line, perhaps with successive pulses from the laser. Although the high NA will have made the region of maximum intensity pretty short along the optic axis, the multiple sites of activation, pulsed or not, will mean that some MTs below and above the plane of focus will have been activated. Given the spindle's geometry, these MTs will curve in toward the plane of observation as one approaches the pole, so the fluorescence observed along the spindle axis will contain fluorescence from MTs that were out of the plane of view near the equator. If these regions of fluorescence move along the arcing KMTs, they will arrive at the poles later than the fluorescence that moved on straight KMTs. The same will of course be true for MTs curving within the plan of observation. This is probably a major cause of the observed smearing of the photo-activated line, and it is probably the explanation for the observed slowing of the fluorescent bar as it nears the pole. Note the very high curvature of most MTs in the vicinity of the pole. These geometrical issues are not mentioned, and they are probably the reason for the most interesting observation presented. In spindles like those formed in PtK cells, which are much flatter, so the MTs are far straighter in each half spindle, this curvature would be much less of an issue. This difference may explain why previous studies of spindle flux have not observed a pole-proximal slowing of flux during metaphase, even though flux rates were shown to vary with time in mitosis.

As we discussed in our response to the essential revisions, our two-photon photoactivation system minimally activates tubulin outside of the focal plane, so the concerns about the contribution of out of focus tubulin should be minimal. We observe that the central portion of the photoactivated line, where the microtubules are essentially straight, slows down near the pole, indicating that the observed slowdown is not the result of increase MT curvature or out of focus tubulin. Furthermore, simulations in which KMTs move at constant speed are inconsistent with our measurements (Figure 7S5).

Ln 529 It is not obvious why the photoconverting line is moved at this rate. Do you mean that the spot of the photo-activating laser was moved to make a line? Please clarify.

We mean that the spot of the photo-activating laser was moved to make a line and have clarified in the text.

Ln 546 The number of pixels width is represented by [check number]

Corrected.

Ln 548 By subtracting the side whose fluorescence was enhanced by photoconversion from the opposite side of the spindle, it sounds as if all your values would be negative. Have you stated this backwards, or did I misunderstand?

Yes, this is stated backwards. We have corrected it in the text.

Ln 563 This section raises the specifics of problems mentioned in Results above. The NA of the objective used here is high, meaning that its depth of field is comparatively narrow. The issue of how birefringent material that is at the edge of, or outside, the depth of field contributes to the retardation recorded is a difficult problem not addressed by Oldenberg and not considered here. The fact that in a spindle with the shape of a human metaphase, many MTs are running at high angles to the plane of focus adds more complexity. The fact that the fraction of MTs at high obliquity to the plane increases near the pole complicates the problem further. Thus, the use of polarizing optics to assess MT orientation is problematic, even with a first-class instrument, like the PolScope. MT orientations determined by electron tomography do not face these issues, albeit they have problems of their own. This means that the similarity of the false color images generated from ET and pol microscopy raise concerns about the validity of the ways in which the colors are generated and what they really mean.

As we discussed in the response to the essential revisions, there is no substantial portion of the spindle outside of the PolScope’s depth of field. Our analysis comparing the 2D approximation, a central slice of the 3D theory and a projection of the full z-depth of the 3D theory show very similar predictions for the MT orientations. It is therefore appropriate to compare the PolScope images to the projection we performed.

Ln 588 The fact that the full 3D geometry of the spindle seen by ET was used in projection to generate the EM version of the MT orientation map, whereas there is no such 3D information in the polarization microscope images used for that map, yet the two maps look quite similar adds to my worries about the value of this presentation.

Again, the 2D approximation and the 3D projection produce very similar microtubule orientations. The PolScope averages the entire z-depth of the spindle together so comparing it to a projection of the ET data and the theory is appropriate.

Reviewer #3 (Recommendations for the authors):This is an excellent work and I only have minor suggestions to the authors that do not require additional experiments:1) The authors are inferring about the origin of k-fibers in human cells by looking at metaphase spindles that are already at steady state. This is in my view the biggest limitation of the present work. Nevertheless, the comparisons drawn from both the electron tomography data and the investigation of spindle microtubule dynamics, supported by the biophysical modeling are powerful in supporting the proposed model of k-fiber self-assembly. Whether or not all kinetochore microtubules originate from kinetochores, remains debatable. For example, the present study does not take into account the possible contribution of Augmin-mediated microtubule-dependent amplification (see e.g. Almeida et al., BioRxiv, 2021.08.18.456780). In addition, the authors cannot formally exclude that a fraction of those 50% of kinetochore microtubules that end at spindle poles have a non-kinetochore origin, as predicted by their "hybrid" model. These limitations should be acknowledged/discussed appropriately.

As discussed in the response to the essential revisions, our experiments address the lifecycle of KMTs in metaphase. Since KMTs turnover roughly every 2.5 minutes in metaphase HeLa cells, the KMTs that originally attached to kinetochores in early prometaphase will no longer be attached to the kinetochore in metaphase. Since the number of KMTs is roughly constant at these times (McEwen et al. 1997; McEwen et al. 1998), this implies that the metaphase KMTs are in (approximate) steady-state. Developing a model of the lifecycle of KMTs during initial attachment and spindle assembly in early prometaphase would require tomography reconstructions and photoconversion experiments of prometaphase cells.

In our framework, the “capture form spindle” model encompasses any mechanism of non-KMT production in the spindle. These microtubules could originate from nucleation at the centrosome, Augmin-mediate microtubule-dependent amplification, or a number of other possible nucleation mechanisms. We have added a sentence to clarify this (pg. 8). Since the non-KMTs turnover very rapidly (~20s), the non-KMT minus ends do not move far (<0.5μm) from where they nucleate. We therefore assume that the positions of the non-KMT minus ends in the reconstruction are a good proxy for where the non-KMTs nucleate. We use all of the non-KMT minus ends in this estimate, including the non-KMT minus ends at the poles.

Concerning the origin of the KMT minus ends at the poles, we include these KMTs in our flux analysis to predict the speed that the KMT minus ends grow towards the pole. If there were, for example, fewer KMTs at the pole, then the flux analysis would predict that the KMT minus ends grow towards the pole more slowly. While we cannot exclude that some portion of the KMTs are captured previously existing non-KMTs, we estimate that this portion of microtubules does not exceed ~20% of the total based on the Bayesian parameter probability comparison (Figure 7C).

2) On page 2, the authors confront the two main models of k-fiber origin: Microtubule capture vs. nucleation at kinetochores. Although they do acknowledge later on that nucleation from kinetochores would be qualitatively indistinguishable from capture of small microtubules near kinetochores, this reviewer is of the opinion that the term "nucleation" might be misleading and should be renamed in order to avoid incorrect interpretations by less familiar readers (e.g. nucleation with minus-ends anchored at kinetochores).

We thank the reviewer for highlighting this lack of clarity in our discussion of “nucleation at kinetochores”. We have added the caveats regarding the capture of small microtubules near the kinetochore to our initial discussion of the nucleate at kinetochore model in the introduction (pg 2).

“The *de-novo* kinetochore nucleated microtubules may in fact be nucleated in the vicinity of the kinetochore and then attach while they are still near zero length, though this process would be distinct from indiscriminate capture of non-KMTs of varied lengths from the spindle (Sikirzhyski et al. 2018).”

And in our initial description of the behaviour of KMTs accompanying Figure 1 (pg 3):

“In our model, we take *de-novo* nucleation of KMTs at the kinetochore and the capture of very short microtubules nucleated in the vicinity of the kinetochore to be equivalent and refer to both as “kinetochore nucleated” KMTs.”

And in the Discussion section (pg 13):

“Our results would also be consistent with a model in which specifically MTs nucleate very near the kinetochore and are rapidly captured while they are still near zero length (Sikirzhyski et al. 2018).”

3) How do the authors explain the reduction of flux velocity near the poles? Do the biophysical models distil any prediction? The authors should discuss that this finding might actually explain different flux rates measured for the same human cell type by different laboratories, possibly due to where in the spindle the photoconversion line is initially positioned relative to the poles.

As discussed in the essential revisions response, we have added speculation on the possible mechanism of KMT elongation to the Discussion (pg.12).

4) It wasn't totally clear to this reviewer how SNAP-Centrin was fluorescently labelled and why is it not depicted in the photoconversion experiments? For example, in Figure 2A, it is not clear whether the two rows correspond to different channels of the same cell? If they do, the images do not seem aligned.

We thank the reviewer for pointing out this confusion. We have swapped the images for our two-photon photoactivation experiment and included a pole marker channel in the updated figure. We stained the SNAP-Centrin with 500nM of SNAP-Sir for 30 minutes and have noted this in the methods section (pg. 14):

“Cells were stained with 500nM SNAP-SIR (New England Biolabs) in standard DMEM media for 30 minutes and then recovered in standard DMEM media for at least 4 hours.”

[Editors' note: further revisions were suggested prior to acceptance, as described below.]

The manuscript has been improved but there are some remaining issues that need to be addressed, as outlined below:1. Please clarify the conceptual language around the model where kMTs have slower motions at the poles. This part is challenging to understand and interpret in the current version of the manuscript. Is the following interpretation reasonable to explain the observations? KMTs by definition have one end on a kinetochore. Near the kinetochore itself, many of the KMTs are short. These grow comparatively fast, so the flux of tubulin in their walls is also fast. Longer KMTs reach closer to the poles. there are fewer of them, but for reasons not identified, they present more drag on the MT growth process, slowing down their polymerization. Thus, their tubulin is fluxing more slowly. Stated more simply, as KMTs elongate, the increase in drag on their poleward motion slows flux. Drag could be one explanation for the slower growth, but there could also be others. If this "slower growth" interpretation is correct, it would imply heterogeneity among kMTs. In this case, a long microtubule that fluxes from KT to pole slower than a shorter kMT should still contribute with slow flux when a mark is photoactivated closer to the chromosomes (where flux is faster?). Is this what was meant? This needs to be clarified in the manuscript.

The interpretation presented here is essentially correct. Our flux-balance analysis (Figure 6) indicates that KMTs whose minus ends are closer to the poles grow more slowly than KMTs whose minus ends are further from the pole. Since the speed of tubulin motion is constant throughout an individual KMT, this implies that longer KMTs that have reached all the way to the pole grow more slowly than shorter KMTs with minus ends still near the kinetochore. Our interpretation is certainly that there is significantly heterogeneity among KMTs, with longer KMTs with their minus ends near the pole fluxing more slowly than shorter KMTs with their minus ends far from the pole, though the speed of tubulin flux within each individual KMT is constant throughout the KMT at any given point in time.

We do not, however, believe that our data can fully support the claim that “as

KMTs elongate, the increase in drag on their poleward motion slows flux” because, as the reviewers note, there could be many explanations for why the KMT minus ends slow down as they approach the pole Our preferred explanation is that the KMT minus end slowdown is the result of dynein-mediated clustering near the pole; however, our current data cannot distinguish between various mechanisms modifying the plus or end minus dynamics of the KMTs.

To clarify this in the manuscript, we have edited our summary descriptions of the model to clarify that the minus ends slow down as they approach the pole, which reduces the flux throughout the KMT. We have edited the abstract (pg. 1):

“Our results indicate that in metaphase, KMTs grow away from the kinetochores along well-defined trajectories, with the speed of the KMT minus ends continually decreasing as the minus ends approach the pole, implying that longer KMTs grow more slowly than shorter KMTs.”

The introduction (pg. 3):

“Taken together, these results lead us to construct a model in which metaphase KMTs nucleate at the kinetochore and grow towards the spindle pole along defined trajectories. The KMT minus ends slow down as they approach the pole. Since the flux of tubulin is constant throughout a single KMT at any given moment in time, the minus end slowdown is coupled to a decrease in the polymerization rate at the KMT plus end. KMTs detach from the kinetochore at a constant rate, independent of the minus end position.”

The description of the spatially dependence photoactivated line experiments (pg. 5):

“We typically track the photoconverted line for ~2.5 minutes, so the line speed we measure is primarily the result of motion of tubulin in KMTs. The faster line speed further from the pole implies that tubulin in short KMTs, whose minus ends are near the kinetochore, move more quickly than tubulin in long KMTs that reach all the way from the kinetochore to the pole.”

The description of the motion of KMT minus ends in introducing the model (pg. 8):

“KMT minus ends move towards the pole with a speed, v(s), that may vary with position along the streamline. The speed of the KMT minus ends is coupled to the plus end polymerization speed because the flux of tubulin is constant throughout a single microtubule at a given point in time. Assuming that KMTs do not deviate from a single streamline trajectory, the minus end speed along the streamlines is equal to the plus end polymerization rate (in the absence of treadmilling).”

The summary description of the model accompanying the final model and experiment comparison in figure 8 (pg. 11):

“We therefore propose a model where metaphase KMTs nucleate at kinetochores and grow along streamlines (Figure 8A). As the KMTs grow and the KMT minus ends approach the pole, the KMT minus ends slow down. This decrease in the KMT minus end speed is coupled to a decrease in the KMT polymerization rate at the plus end. As a result of this minus end slow down, longer KMTs that reach all the way to the pole grow more slowly than short KMTs with minus ends near the kinetochore. When the KMT minus ends reach the pole, minus end depolymerization causes tubulin to treadmill through the KMT.”

The description of the model in the discussion (pg. 13):

“The KMT minus end speed decreases as the KMTs approach the pole. Because the flux of tubulin is constant throughout a single KMT at any given point in time, the decrease in the minus end speed must be accompanied by a decrease in the plus end polymerization rate (in the absence of treadmilling).”

And our description of the mechanism of minus end transport in the discussion (pg. 13):

“We observed that the speed of motion the photoconverted line was slower for lines drawn near the pole than in the center of the spindle (Figure 3), and our analysis of the position of the KMT minus ends indicates that the KMT minus ends themselves slow down as they approach the pole (Figure 6). The mechanism of KMT minus end transport to the pole is not clear. Tubulin polymerization at the kinetochore might push the KMTs backwards towards the pole. Alternatively, the KMT minus ends could be transported to the pole by dynein (Elting et al. 2014, Sikirzhytski et al. 2014). Either way, since the flux of tubulin is constant throughout an individual KMT at any given point in time, the speed of the KMT minus end must be coupled to the speed of tubulin polymerization at the KMT plus end. The mechanism responsible for KMT minus ends slowing down as they approach the pole is also unclear. Longer KMTs are presumable subject to more drag, and more friction with the surrounding network of non-KMTs, which might lead to a reduction in the KMT polymerization rate and hence a slowdown of KMT minus ends. In *Xenopus* egg extract spindles, non-KMTs move more slowly near the spindle poles than near the spindle equator. Inhibiting dynein causes the non-KMT speed to become spatially uniform, suggesting that the non-KMT slowdown is the result of dynein-mediated MT clustering (Burbank 2007, Yang 2008). A similar mechanism might explain the slowdown of KMTs described in this work. Further experiments in dynein-inhibited HeLa spindles will be necessary to test this possibility.”

Further, this interpretation makes a prediction: near the kinetochores, where there are both long and short MTs, a bar of induced fluorescence should spread more quickly than an identical bar of fluorescence generated near the pole. This prediction is based on the ET data in the companion paper that near the kinetochore there are both long and short KMTs, whereas near the pole, only long KMTs are found. The heterogeneity in KMT length near the kinetochores should lead to a faster spread of induced fluorescence. This would be an easy measurement to make, probably on existing data, and the results might either support or refute the author's interpretation. Although inclusion of such data is not essential, it would strengthen the manuscript.

We thank the reviewers for the suggestion. We analyzed the change in the width of the peak over time. We compared this analysis to the predictions of how quickly the photoactivated line spread in the photoactivation simulations and found that the model predictions agreed relatively well with the observed peak spreading from the photoconversion experiments. The model predicts the correct line spreading speed, but the effect from the different populations of KMTs is relatively subtle, in both the experiments and simulations. We have included this analysis as a new figure 8S1 and noted in the text (pg 12):

“The model also accurately predicted a slight spread in the peak width over time (Figure 8S1).”

2. Could the authors clarify whether they collected single-plane images or of z-stacks that were subsequently projected in 2D for the above analysis and explain how they were processed.

We are collecting a single z plane at the center of the photoactivated line. We suspect that this confusion may have arisen from the inclusion of a control experiment where we collected z-stacks to determine the width of the photoactivated line in the z-direction perpendicular to the imaging plane (Figure 2S1) To clarify this issue, we have edited the methods section of the manuscript to read (pg. 15):

“Two separate fluorescence channels were acquired every 5s with either 500ms exposure, 488nm excitation, 514/30 emission for the photoactivated PA-GFP channel or 300ms exposure 647nnm excitation, 647 longpass emission in a single z plane for both channels.”

Reviewer #1 (Recommendations for the authors):The authors have made some important changes to their manuscript, and the result is a much-improved paper. Experimentally, the use of two-photon activation for the study of microtubule movements in a metaphase spindle is a significant improvement. Although the results are similar to those previously obtained, the narrower field depth of photoactivation eliminates one plausible alternative explanation for the observations and raises confidence in the interpretations given. Well done.The new material that explains more fully the relationship between 3D and 2D calculations for the active liquid crystal theory is again a substantive improvement in the paper, but I note that such material was more prevalent in the rebuttal than in the paper itself. The difficulties with the previous version were not due simply to succinctness. There were confusions and even inaccuracies, but I think the new version is acceptable.The response to criticisms of the previous paper's description of KMT turnover is fine. The new text clarifies what the authors are talking about, again a substantive improvement. This reviewer still has trouble, though, with the descriptions of the changes in speed of KMT minus ends as a function of position in the spindle. The explanation offered in the rebuttal and in the discussion, that longer KMTs add subunits more slowly at the kinetochore, is plausible and helps the reader understand immediately what is probably going on to account for the observations. I encourage the authors to work words of that kind into their paper somewhere near the first statement of this observation, because otherwise, the result appears to be hard to understand. They talk about KMTs slowing as they approach the pole, but isn't clearer to say that long KMTs, whose minus ends are closer to the pole, add tubulin at their kinetochore-associated end more slowly than shorter KMTs. All in all, the paper is now a fine piece of work and a significant contribution to the field.

We thank the reviewer for their careful attention to detail and their suggestions to improve the explanations in the manuscript. We have edited the manuscript to add this reference to long and short KMTs in our description of the spatial dependence of the photoactivation experiment (pg. 5):

“We typically track the photoconverted line for ~2.5 minutes, so the line speed we measure is primarily the result of motion of tubulin in KMTs, implying that tubulin in short KMTs, whose minus ends are near the kinetochore, move more quickly than tubulin in long KMTs that reach all the way from the kinetochore to the pole.”

We have also edited the description when we introduce the KMT minus end speed v(s) to read (pg. 8):

“KMT minus ends move towards the pole with a speed, v(s), that may vary with position along the streamline. The speed of the KMT minus ends is coupled to the plus end polymerization speed because the flux of tubulin is constant throughout a single microtubule at a given moment in time. Assuming that KMTs do not deviate from a single streamline trajectory, the minus end speed along the streamlines is equal to the plus end polymerization rate (in the absence of treadmilling).”

Specific commentsAbstract: An issue raised in initial review is that the words used to describe the speed of KMT movement toward the pole suggest that different parts of one MT move at different rates, which the authors clearly do not mean. Their sentence, "in metaphase, KMTs grow away from the kinetochores along.well-defined trajectories, continually decreasing in speed as they approach the poles" perpetuates this problem. By definition, a KMT has its plus end associated with a kinetochore. If the minus ends of these MTs move toward the pole at different rates, then the rates of growth at the kinetochore must be inversely related to KMT length. Some statement to that effect would clarify the authors meaning.

We have edited the abstract to read (pg 1):

“Our results indicate that in metaphase, KMTs grow away from the kinetochores along well-defined trajectories, with the speed of the KMT minus ends continually decreasing as the minus ends approach the pole, implying that longer KMTs grow more slowly than shorter KMTs.”

Moreover, the statement "KMTs predominately nucleate de novo at kinetochores" lacks the specification, made clear in the rebuttal document, that the authors mean during metaphase. Please be explicit about this. Otherwise, the statement is incorrect.

Done

Introduction:Ln 103 Again, specify metaphase.

Done

Results:Ln 125, the same.

Done

Ln 241 "Due to our use of a low numerical aperture 242 condenser (NA=0.85)…" This sentence is an over-simplification. With this condenser NA, the cone of illumination still has an angle of about 50o. This is hardly a column through the spindle, as described in the rebuttal. The measurements made with this pair of objective and condenser have a depth of field of ~1 µm, so a lot of out-of-focus retardation is contributing to the brightness measured at each pixel. Given the broad cone of illumination, retardance measured at any point in the image plane includes retardance from parts of the spindle volume away from the optic axis at that point. The apparently impressive correspondence between Oldenbourg's single MT measurement and the prediction for average spindle brightness is consistent with my view that the pole-scope observations average over a range of X and Y as the retardation from different levels of Z are added to each position in the image plane. Thus, the observations lack the precision of a "projection", but the point is not worth fussing over. I agree that these polarization measurements have value, but I doubt that they have the precision claimed.

We have edited this line of the text to read:

“Due to our use of a low numerical aperture condenser (NA=0.85), it is reasonable to approximate the Polscope measurements as projections over the z-depth of the spindle.”

Ln 265 The second reference to Fürthauer et al. lacks a journal.

Fixed.

Ln 273 You persist in equating the Laplacian to 0 without mentioned that you are using a 2D approximation. I am puzzled by your resistance to accuracy.

This statement is correct as written and does not result from using a 2D approximation. The reviewer’s confusion is presumable due to the fact that the Laplacian of the angle describing the orientation of microtubules is zero in 2D but not in 3D. However, the Laplacian of the nematic director field (i.e. the field of unit vectors denoting the orientation of microtubules) is zero in both 2D and 3D.

Ln 476 Again, add metaphase.

Done

Ln 480 and following. This statement is very nice and clear and fully understandable, when accompanied by the inference that the rate of polymerization of tubulin at the kinetochore is inversely related to the length of the MT. Why not add that here for completeness?

We have edited description of figure 8 to note (pg 11):

“We therefore propose a model where metaphase KMTs nucleate at kinetochores and grow along streamlines (Figure 8A). As the KMTs grow and the KMT minus ends approach the pole, the KMT minus ends slow down. This decrease in the KMT minus end speed is coupled to a decrease in the KMT polymerization rate at the plus end. As a result of this minus end slow down, longer KMTs that reach all the way to the pole grow more slowly than short KMTs with minus ends near the kinetochore. When the KMT minus ends reach the pole, minus end depolymerization causes tubulin to treadmill through the KMT. The KMTs detach from the kinetochore at a constant rate, independent of their position in the spindle.”

Ln 532 The word "speed" here is ambiguous, because you are talking both about MT turnover and speed of MT motions. Please clarify.

We have edited the text to read:

“The photoconversion experiments also showed that tubulin in KMTs moves more slowly near the poles and that KMT turnover was uniform throughout the spindle (Figure 2 and 3)”

Ln 545 Again, by stating that MT minus ends slow down as they approach the pole without any mention of what's going on at the plus end of those same MTs, you suggest that the minus ends are moving at a rate that is different from the rest of the MT, which you don't intend.

We have edited the text here to read:

“The KMT minus end speed decreases as the KMTs approach the pole. This decrease in the KMT minus end speed must be accompanied by a decrease in the plus end polymerization rate because the flux of tubulin is constant throughout a single KMT at any given moment in time (in the absence of treadmilling).”

Ln 938 Need a space after 0

Done

Reviewer #3 (Recommendations for the authors):The authors did an excellent job clarifying the required revision points. However, this reviewer is still not fully satisfied with the explanation and data about flux decrease near the poles. It is not so much the way the photoactivation mark was created (1- vs 2-photon, 3D vs 2D) but the way the movement of the mark was imaged and tracked. There is no information in the methods on whether the authors collected single-plane images or a collection of z-stacks that were subsequently projected in 2D.

As discussed above, we take a single z-plane during the photoactivation experiments. We have edited the methods section to clarify this (pg 15):

“Two separate fluorescence channels were acquired every 5s with either 500ms exposure, 488nm excitation, 514/30 emission for the photoactivated PA-GFP channel or 300ms exposure 647nnm excitation, 647 longpass emission in a single z plane for both channels.”

Either way, higher microtubule curvature near the poles would be masked in any 2D analysis, creating the illusion of flux slowing down near the poles.

We directly assessed this possibility in Figure 7s5, where we show that a photoactivation simulation without KMT minus ends slowing down near the poles is not consistent with the observed increased in the speed of the photoactivated line further from the pole.

If one assumes that the photoactivation mark created on spindle microtubules near the equator does slow down when it reaches the poles, it implies discontinuity of the photoactivated microtubules, which is hard to conceive given that microtubules are transported as continuous units, even if only a fraction of the microtubule minus ends reach the poles. Could the authors prove me wrong on this, or am I missing something very obvious here? Recognizing potential technical limitations/caveats and making this point absolutely crystal clear in a simple language that a common cell biologist without extensive notions of physics can understand would certainly help the field and many of the potential readers of this article.

We hope that the edits we have made to the text have clarified these issues. We do not believe that the slowing mark implies a discontinuity of photoactivated microtubules. There are a mix of long KMTs which reach all the way to the pole and short KMTs near the kinetochore (along with a spread of intermediate KMTs with minus ends throughout the spindle). We claim that the shorter microtubules near the kinetochore grow more quickly than the longer KMTs that reach the pole. As a result, tubulin in the KMTs near the kinetochore moves towards the pole more quickly than tubulin in KMTs near the pole. We have clarified this point in the text as noted in the response to point 1 in the essential revisions.